# Mitochondrial calcium uniporter stabilization preserves energetic homeostasis during Complex I impairment

Enrique Balderas [1], David R. Eberhardt[1], Sandra Lee[1], John M. Pleinis[2], Salah Sommakia[1], Anthony M. Balynas[1], Xue Yin[1], Mitchell C. Parker[3], Colin T. Maguire [4], Scott Cho [1], Marta W. Szulik [1], Anna Bakhtina[1], Ryan D. Bia[1], Marisa W. Friederich[5,6], Timothy M. Locke [7], Johan L. K. Van Hove[5], Stavros G. Drakos [1,8], Yasemin Sancak [7], Martin Tristani-Firouzi [1,9], Sarah Franklin[1,8], Aylin R. Rodan [2,10] & Dipayan Chaudhuri [1,8,11✉]

Calcium entering mitochondria potently stimulates ATP synthesis. Increases in calcium preserve energy synthesis in cardiomyopathies caused by mitochondrial dysfunction, and occur due to enhanced activity of the mitochondrial calcium uniporter channel. The signaling mechanism that mediates this compensatory increase remains unknown. Here, we find that increases in the uniporter are due to impairment in Complex I of the electron transport chain. In normal physiology, Complex I promotes uniporter degradation via an interaction with the uniporter pore-forming subunit, a process we term Complex I-induced protein turnover. When Complex I dysfunction ensues, contact with the uniporter is inhibited, preventing degradation, and leading to a build-up in functional channels. Preventing uniporter activity leads to early demise in Complex I-deficient animals. Conversely, enhancing uniporter stability rescues survival and function in Complex I deficiency. Taken together, our data identify a fundamental pathway producing compensatory increases in calcium influx during Complex I impairment.

[1] Nora Eccles Harrison Cardiovascular Research and Training Institute, University of Utah, Salt Lake City, UT, USA. [2] Molecular Medicine Program, University of Utah, Salt Lake City, UT, USA. [3] University of Nevada Reno School of Medicine, Reno, NV, USA. [4] Clinical & Translational Science Institute, University of Utah, Salt Lake City, UT, USA. [5] Section of Clinical Genetics and Metabolism, Department of Pediatrics, University of Colorado, Aurora, CO, USA. [6] Department of Pathology and Laboratory Medicine, Children's Hospital Colorado, Aurora, CO, USA. [7] Department of Pharmacology, University of Washington, Seattle, WA, USA. [8] Division of Cardiovascular Medicine, Department of Internal Medicine, University of Utah, Salt Lake City, UT, USA. [9] Division of Pediatric Cardiology, University of Utah School of Medicine, Salt Lake City, UT, USA. [10] Division of Nephrology and Hypertension, Department of Internal Medicine, University of Utah, Salt Lake City, UT, USA. [11] Department of Biochemistry, Department of Biomedical Engineering, University of Utah, Salt Lake City, UT, USA. ✉email: dipayan.chaudhuri@hsc.utah.edu

Calcium ($Ca^{2+}$) is a potent regulator of metabolism, acting on multiple enzymes in mitochondria. Within the matrix, moderate $Ca^{2+}$ elevations double ATP synthesis rates, helping match energetic supply to demand[1,2]. Pathological failure to meet demand is a common feature across in cardiomyopathies. In fact, energetic failure can be a primary cause of cardiomyopathy in mitochondrial diseases. Such diseases involve deficient oxidative phosphorylation, and arise from mutations in mitochondrial proteins encoded by either the nuclear or the mitochondrial genome (mtDNA), with Complex I of the electron transport chain (ETC) most affected[3,4]. Despite often severe pathology, children with mitochondrial cardiomyopathies may survive prolonged periods, suggesting mechanisms exist to compensate for ETC dysfunction. Identifying such pathways offers new opportunities for broad therapeutic intervention, as ETC impairment is a fundamental feature of many common cardiac and neurological diseases.

There are limited prior investigations of mitochondrial $Ca^{2+}$ signaling during ETC deficiency[5–11]. The typical finding is reduced or unchanged $Ca^{2+}$ uptake in the presence of a diminished membrane voltage gradient ($\Delta\Psi$), where the diminished gradient correlated with severity of ETC deficiency. Because $Ca^{2+}$ influx through the uniporter is driven by this voltage gradient, a change to either $\Delta\Psi$ or uniporter activity can alter the size of $Ca^{2+}$ influx. In ETC-deficient mitochondria, precisely such diminished $\Delta\Psi$ may mask any compensatory increases in uniporter activity using typical in vitro imaging assays. In fact, we noted an interesting phenotype in mice with a cardiac-specific deletion of the transcription factor for mtDNA, *Tfam (*transcription factor A, mitochondrial*)*. This is a well-established model for mitochondrial cardiomyopathies, featuring a dilated cardiomyopathy caused by impaired transcription of core mtDNA-encoded ETC subunits, with Complex I function most severely affected[12,13]. Notably, as cardiomyopathy develops in the *Tfam* knockout, cardiac mitochondria become extremely $Ca^{2+}$ avid. These increases in mitochondrial $Ca^{2+}$ help rescue respiration and ATP synthesis to near wild-type levels[12]. The increase in mitochondrial $Ca^{2+}$ is mediated by a multi-subunit channel known as the mitochondrial $Ca^{2+}$ uniporter[14–20]. This channel resides in the inner membrane of mitochondria, is activated by cytoplasmic $Ca^{2+}$, and is the main portal for $Ca^{2+}$ entry into the matrix. In animal models, cardiac impairment of this channel leads to energetic supply-demand mismatch, leaving open the question of whether increased uniporter activity is an essential compensatory mechanism during ETC impairment.

Here, we unravel the mechanism for the enhancement in uniporter activity during ETC dysfunction and show that it is essential for survival. We find that this phenomenon depends on impairment in Complex I and is widespread, occurring in a variety of cell types, and across species. Under normal Complex I activity, uniporter turnover is accelerated by an interaction between the N-terminal domain (NTD) of the pore-forming subunit of the uniporter (MCU) and Complex I, a mechanism we term Complex I-induced protein turnover (CLIPT). When Complex I becomes dysfunctional, CLIPT is abrogated, leading to slower MCU turnover and a buildup of uniporter channels. This mechanism is evident in hearts from *Tfam* knockout mice, and its disruption leads to their faster demise. Similarly, in *Drosophila*, $Ca^{2+}$ signaling through the uniporter is essential for survival during Complex I dysfunction. Enhancing this mechanism, by overexpression of MCU or its NTD, in Complex I deficient flies rescues both functional impairments and survival.

## Results

### Complex I dysfunction leads to enhanced mitochondrial calcium uniporter levels.
Measuring $Ca^{2+}$ uptake using typical $Ca^{2+}$ fluorescence assays poorly captures uniporter activity during ETC dysfunction, because such dysfunction also alters other parameters controlling $Ca^{2+}$ transport, including $\Delta\Psi$, pH, morphology, and $Ca^{2+}$ buffering[8,21,22]. To more precisely examine uniporter-mediated $Ca^{2+}$ uptake, we used whole-mitoplast voltage-clamp electrophysiology[17]. In this assay, micropipettes are attached to individual spherical mitoplasts (mitochondria stripped of their outer membranes) to record ionic currents through a voltage-clamp feedback electrode. This allows full control over $\Delta\Psi$, matrix and external solutions, eliminating uncontrolled variation in the factors listed, and allowing direct measurement of uniporter $Ca^{2+}$ currents ($I_{MiCa}$).

The complexes with *Tfam*-dependent subunits (Complexes I, III, and IV) may integrate into a supercomplex known as the respirasome. Therefore, we first tested whether enhanced $I_{MiCa}$ depended on disrupting the respirasome or a specific ETC complex. We examined HEK293T cells because long-term culture with drugs is possible, gene-edited lines exist for both uniporter[23] and Complex I analysis[24], and uniporter currents are robust[15]. Individual ETC Complexes were inhibited pharmacologically for 2–3 days in HEK293T cells (Fig. S1). To maintain viability, cells were cultured with 0.4 mM uridine to maintain pyrimidine biosynthesis and 2 mM pyruvate to regenerate $NAD^+$ [25,26]. Whereas chronic inhibition with Complex III or IV antagonists (1 μM antimycin A, 200 μM sodium azide) produced no change in $I_{MiCa}$ (Fig. S2), disruption of Complex I with rotenone produced a dose-dependent increase in $I_{MiCa}$ (Fig. 1a, b). Rotenone did not alter channel kinetics, and failed to increase $I_{MiCa}$ when added acutely during mitoplast recordings (Fig. S3). This suggests that rotenone did not enhance $I_{MiCa}$ directly, but through its effect on Complex I, inhibiting electron transfer to ubiquinone, leading to substantial reductions in Complex I activity and levels (Figs. S1, S4). Similar to the effect in *Tfam* knockout hearts[12], protein levels for uniporter subunits MCU and EMRE were increased (Figs. 1c, S5), though gene expression was not (relative mRNA expression MCU, $0.89 \pm 0.18$; EMRE, $1.37 \pm 0.4$; MICU1, $1.03 \pm 0.15$, $n = 3$).

Having established that the increase in uniporter currents observed in *Tfam* knockout cardiomyocytes could be reproduced by isolated Complex I inhibition, we tested the robustness of this phenomenon. To confirm the increase in $I_{MiCa}$ was due to Complex I and not an off-target rotenone effect, we measured $I_{MiCa}$ in HEK293T cells featuring deletion of the Complex I accessory subunit NDUFB10 or late assembly factor FOXRED1 ($NDUFB10^{KO}$, $FOXRED1^{KO}$, Fig. S6)[24,27]. As with rotenone-treated cells, genetic inhibition of Complex I depolarized $\Delta\Psi$ (Fig. S7). In these, $I_{MiCa}$ was also substantially increased over controls (Fig. 1d, e). We saw a similar increase in $I_{MiCa}$ after disruption of accessory subunit NDUFS4 as well (Figs. S6, S8). To see if $I_{MiCa}$ enhancement was prevalent in Complex I-mediated disease, we obtained fibroblasts from an infant with fatal lactic acidosis and cardiomyopathy due to NDUFB10 deficiency ($NDUFB10^{-/C107S}$)[28]. Induced pluripotent stem cells (IPSCs) derived from these fibroblasts retained the compound heterozygous mutations and had essentially absent NDUFB10 expression (Fig. S9), validating their use for gauging uniporter activity in Complex I-deficient disease. $I_{MiCa}$ in these cells was also increased, and this effect was rescued by re-expression of wild-type NDUFB10 (Fig. 1f). Next, to see if this effect was evolutionarily conserved, we examined *Drosophila* with Complex I dysfunction in flight muscle, generated by the expression of short hairpin RNA (shRNA) targeting NDUFB10 via *MHC*-Gal4 ($NDUFB10^{RNAi}$)[29]. Here too Complex I dysfunction was associated with an increase in $I_{MiCa}$ (Fig. 1g). The increase in $I_{MiCa}$ corresponded to increases in mitochondrial $Ca^{2+}$ uptake measured by imaging (Fig. S10). Matrix free $[Ca^{2+}]$ levels were

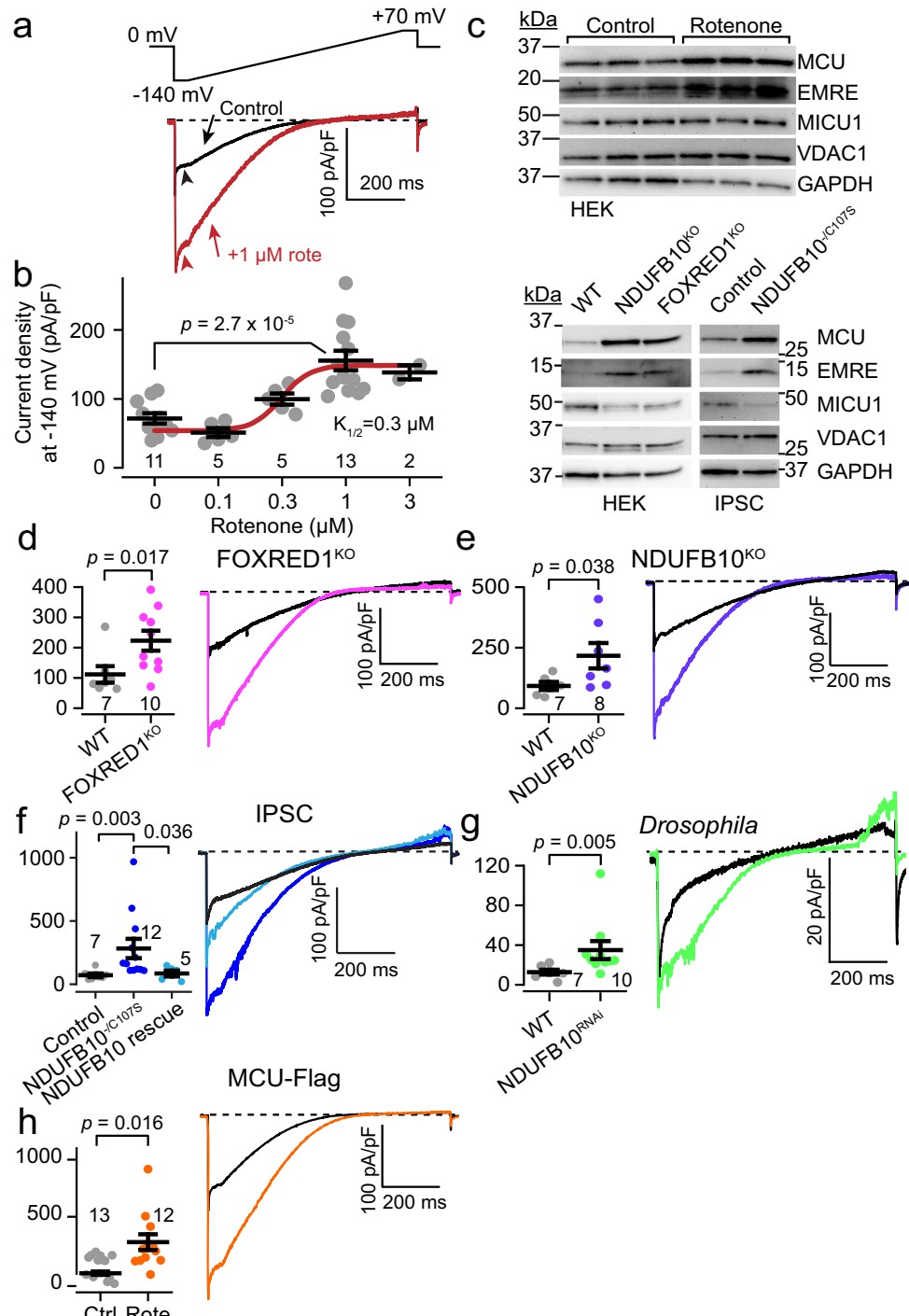

**Fig. 1 Complex I dysfunction increases $I_{MiCa}$. a** Top, voltage ramp protocol. Bottom, Exemplar $I_{MiCa}$ traces are larger after chronic 1 μM rotenone treatment in HEK293T cells. **b** Summary rotenone dose-response curve. **c** Immunoblotting reveals increased uniporter subunit proteins after Complex I impairment in HEK293T, NDUFB10$^{KO}$, FOXRED1$^{KO}$, and patient-derived IPSCs (NDUFB10$^{−/C107S}$) compared to controls. VDAC1 and GAPDH are loading controls. Here and throughout, representative blots from at least 3 replicates are shown. kDa, kilodaltons. **d–g** Each panel contains the summary (left) and exemplar (right) for peak inward $I_{MiCa}$ in FOXRED1$^{KO}$ (**d**), NDUFB10$^{KO}$ (**e**), patient-derived IPSCs (**f**), and *Drosophila* NDUFB10$^{RNAi}$ (**g**), compared to controls (black). Each point in the summary graphs in panels 1b, 1d-h represents the current density measured at −140 mV for an individual mitoplast (see arrowheads in (**a**)). **h** $I_{MiCa}$ is increased in cells expressing MCU-Flag after Complex I impairment. In summary graphs, data are presented as mean values ± SEM. *N* values are listed in the summary graph here and throughout. Statistics: (**a**) 1-way ANOVA followed by Bonferroni corrected means comparisons; (**f**) Kruskal–Wallis followed by Dunn's test with Bonferroni correction; all others, two-sided Mann–Whitney *U* tests. Source data are provided as a Source Data file.

marginally higher in the Complex I deficient cells (Fig. S11), though there was substantial overlap with controls. One downstream effect of mitochondrial $Ca^{2+}$ is to stimulate pyruvate dehydrogenase phosphatases that reduce PDH phosphorylation and increase its activity. Consistent with enhanced $Ca^{2+}$ uptake, we found reduced PDH phosphorylation in Complex I inhibited cells (Fig. S12). Thus, one main purpose of enhancing $I_{MiCa}$ may be to maintain mitochondrial $Ca^{2+}$ levels when dysfunctional Complex I leads to a depolarized mitochondrial membrane potential and blunted $Ca^{2+}$ uptake through individual channels. Finally, we confirmed that the increase in current was specific to the uniporter, as the chloride current carried by the ubiquitous inner membrane anion channel showed no change after rotenone incubation in HEK293T cells (Fig. S13). In summary, we show that disrupting Complex I leads to an approximately 2–3 fold increase in $I_{MiCa}$. This effect can be triggered by a variety of insults to Complex I (*Tfam* knockout, rotenone, gene editing, shRNA, and congenital mutations), and occurs across species (mouse, *Drosophila*, human), cell types (cardiomyocytes, flight muscle, cultured lines, and IPSCs), and in human disease.

As with the *Tfam* knockout mice[12,30] and the rotenone-treated HEK293T cells, NDUFB10[KO], FOXRED1[KO], and NDUFB10[−/C107S] cells also exhibited increased MCU protein, suggesting that the enhancement in $I_{MiCa}$ reflects an increase in channels (Fig. 1c). MCU levels increased in these different lines despite the absence of a corresponding mRNA upregulation (MCU mRNA expression versus control: 0.95, NDUFB10[KO]; 0.68, FOXRED1[KO]; 0.88, NDUFB10[−/C107S], $n = 2$). The discordance between MCU protein and mRNA levels may be explained by a post-transcriptional mechanism. To further verify this, we expressed carboxy-terminal Flag-tagged MCU in MCU[KO] HEK293T cells from a plasmid lacking the native promoter, introns, and other untranslated regions, disrupting endogenous transcriptional regulation[23]. Here too we found that $I_{MiCa}$ increased after rotenone incubation (Fig. 1h). Thus, our results suggest Complex I regulation of the uniporter is post-transcriptional.

**Aberrant $NAD^+$/NADH oxidation and reactive oxygen species (ROS) production drive uniporter enhancement.** Next, we investigated the signal that leads to $I_{MiCa}$ enhancement. When Complex I becomes dysfunctional, its NADH:ubiquinone oxidoreductase activity is impaired, leading to an increased $NADH:NAD^+$ ratio and greater superoxide production, which persist until excess ROS leads to self-inactivation and Complex I disassembly (Fig. 2a)[31,32]. We found evidence for increases in ROS levels using the mitochondrially-targeted ROS sensor MitoSOX[33], and increased $NADH:NAD^+$ ratio using a mitochondrially-targeted version of the genetically-encoded SoNar sensor (Fig. 2b, c)[34]. Conversely, $NADH:NAD^+$ ratio was reduced in MCU[KO] cells (Fig. S14). To test if these were key signals for $I_{MiCa}$ enhancement, we turned again to whole-mitoplast electrophysiology. To blunt changes in the $NADH:NAD^+$ ratio, cell lines were engineered to stably express a mitochondrially-targeted water-forming NADH oxidase from *Lactobacillus brevis* (*Lb*NOX, Fig. S15a). This approach was previously shown to reduce $NADH/NAD^+$ ratios during Complex I inhibition[25]. $I_{MiCa}$ magnitudes from *Lb*NOX-expressing cells were no longer increased following rotenone treatment, suggesting that abnormal NADH oxidation contributes to uniporter enhancement (Fig. 2d).

Employing a similar strategy to blunt the increase in superoxide, we generated cells stably overexpressing *mitochondrial superoxide dismutase 2* (SOD2, Fig. S15b). In these cells, baseline and rotenone-induced superoxide production was blunted (Fig. S16a). $I_{MiCa}$ also failed to increase following rotenone

treatment (Fig. 2e). We then tested the reciprocal hypothesis, that producing pathological ROS would be sufficient to induce $I_{MiCa}$ enhancement. A cell line was created stably expressing a mitochondria-targeted version of mini-singlet oxygen generator (mt-miniSOG, Fig. S15c). This fluorescent flavoprotein generates both singlet oxygen and superoxide when excited by blue light, and substantially increased production of ROS (Fig. S16b)[35,36]. In cells exposed to blue light 2–3 days before recording, $I_{MiCa}$ size was ~3x greater than in unexposed cells (Fig. 2f).

Mito-*Lb*NOX has been used to rescue cell survival by preserving $NADH:NAD^+$ homeostasis during dysfunction induced by the Complex I inhibitor piericidin[25]. Thus, to determine whether maintaining $NADH:NAD^+$ homeostasis during Complex I impairment required the uniporter, we expressed mito-*Lb*NOX in control and MCU[KO] HEK293T and exposed them to 500 nM piericidin. In control cells, mito-*Lb*NOX expression did not alter cell survival and rescued the cell proliferation defect caused by piericidin, as expected (Fig. S17). In contrast, expressing mito-*Lb*NOX in MCU[KO] cells both impaired baseline cell survival and failed to rescue cell death after piericidin treatment, suggesting that the uniporter maintenance of NADH production is required for survival during Complex I dysfunction. Taken together, these data indicate that aberrant NADH oxidation and ROS generation are critical signals for uniporter enhancement during Complex I dysfunction.

**The MCU N-terminal domain is necessary for Complex-I mediated enhancement.** In a prior report, MCU was shown to be sensitive to matrix redox status and oxidative stress via S-glutathionylation at a conserved cysteine residue in its N-terminal domain (NTD)[37]. Though such regulation was not specific to Complex I, it offered a potential mechanism to explain $I_{MiCa}$ enhancement. Therefore, we expressed an MCU construct where all five cysteines were mutated (Cysteine-free MCU, CF-MCU) in MCU[KO] cells. As in the prior report, CF-MCU conducted $I_{MiCa}$, with no obvious effect on basal channel function. Unexpectedly, rotenone-mediated $I_{MiCa}$ enhancement persisted in these cells (Fig. S18). These data indicate that redox sensation by MCU cysteines fails to explain uniporter enhancement observed during Complex I dysfunction.

Nevertheless, the MCU NTD itself remained an interesting target to examine further. This structure is evolutionarily conserved, forming an independent domain within the matrix[38–40]. The NTD has been implicated in channel dimerization, though it is not essential for channel activity[38,39]. Given its highly-conserved structure, we hypothesized that the NTD may be responsible for Complex I-mediated enhancement. MCU lacking the NTD (ΔNTD-MCU) was stably expressed in MCU[KO] cells. Consistent with our hypothesis, $I_{MiCa}$ in these cells failed to increase in response to rotenone (Fig. 2g). Thus, an intact MCU NTD is necessary for the enhancement in current seen during Complex I dysfunction.

**Uniporter enhancement depends on an interaction between MCU and Complex I.** The importance of ROS in $I_{MiCa}$ enhancement revealed an intriguing discrepancy. Complex III inhibition with antimycin A produces abundant ROS within the matrix and intermembrane space[41], yet it did not lead to $I_{MiCa}$ enhancement. We surmised that aberrant ROS may be primarily disrupting Complex I, since it is much larger than the uniporter and extends further into the matrix, with Complex I dysfunction subsequently altering uniporter behavior. This predicts close proximity between MCU and Complex I. Evidence for such an interaction was found incidentally in a proteomic screen for Complex I assembly factors, where MCU (annotated as

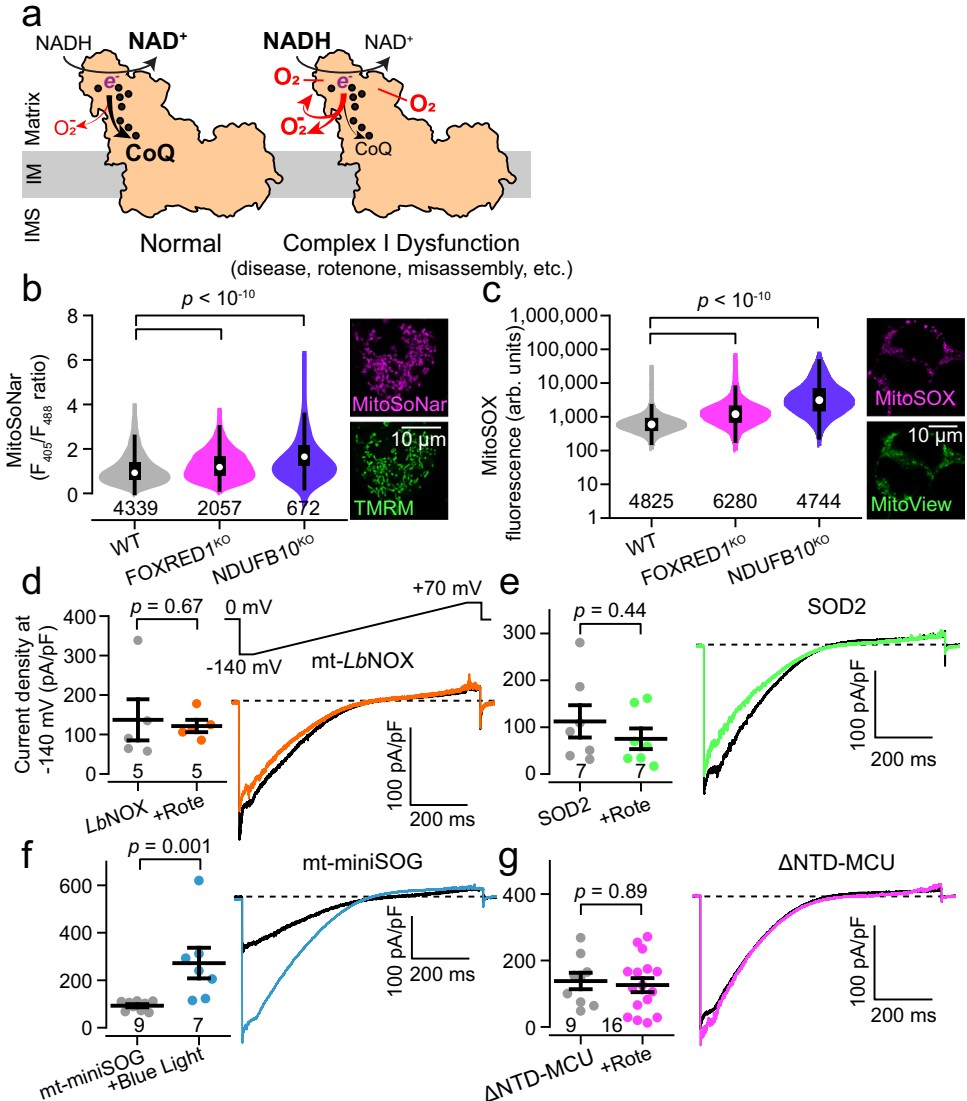

**Fig. 2 Reactive oxygen species signal $I_{MiCa}$ enhancement. a** Complex I cartoon depicting minimal (left) and excessive (right) electron ($e^-$) transfer from NADH to superoxide compared to ubiquinone (CoQ). Excessive $O_2^-$ can escape from or self-inactivate Complex I. Black dots, Fe-S clusters. IMS, intermembrane space; IM, inner membrane. **b, c** Violin plots of MitoSoNar fluorescence ratio (B, increased $F_{405}/F_{488}$ value corresponds to increased NADH:NAD$^+$ ratio) and mitochondrial ROS sensor MitoSOX fluorescence (**c**) measured via flow cytometry. Average MitoSoNar ratio: 1.1 (WT), 1.3 (FOXRED1$^{KO}$), 1.8 (NDUFB10$^{KO}$). Average MitoSOX: 849 (WT), 1950 (FOXRED1$^{KO}$), 5704 (NDUFB10$^{KO}$). Insets show mitochondrial targeting of the corresponding sensor. **d–g** Summary (left) and exemplar (right) $I_{MiCa}$ in HEK293T cells expressing mito-$Lb$NOX (**d**), SOD2 (**e**), mito-miniSOG (**f**), and ΔNTD-MCU (**g**). Arb. units, arbitrary units. Summary data are presented as mean values ± SEM. Violin plot insets display Tukey boxplots. Statistics: (**b–c**) 1-way ANOVA followed by Bonferroni-corrected means comparisons; (**d–g**) two-sided Mann–Whitney $U$ tests. Source data are provided as a Source Data file.

CCDC109A) bound Complex I without affecting its assembly[42]. Similarly, a more recent compendium of mitochondrial protein-protein interactions revealed close proximity between MCU and several subunits of Complex I[43]. We therefore tested for a Complex I-MCU interaction. Immunoprecipitation of Flag-tagged MCU in 1% digitonin was followed by mass spectro-metric analysis of co-precipitating proteins, identifying NDUFA3, NDUFA8, and NDUFA13 as potential interactors (Fig. S19). All three of these are closely apposed on the Complex I structure. We confirmed this interaction by co-immunoprecipitating NDUFA13 with MCU-Flag but not Flag-tagged succinate dehydrogenase complex subunit B (SDHB-Flag) (Figs. 3a, S20).

To confirm the interaction within live cells with intact mitochondria, we turned to Förster energy resonance transfer (FRET) assays. For the mVenus-mCerulean pair used here, the

Förster radius for 50% FRET efficiency is ~5 nm[44]. We took advantage of prior studies that showed that several Complex I subunits, typically those that had carboxy-termini exposed at the Complex I surface, are unaffected by carboxy-terminal fusion with fluorescent proteins[45]. MCU can similarly be linked at its C-terminus with fluorescent proteins[37]. To sample various portions of Complex I, we tagged eight NDUF subunits with mVenus, while MCU was fused to mCerulean (Figs. 3b, S21). FRET was detected from co-transfected constructs in HEK293T cells using flow cytometry, which allows us to measure interactions over a wide range of protein expression levels[46]. NDUFA2 and NDUFA5 failed to target mitochondria when tagged with mVenus, and reassuringly showed no FRET with MCU-mCerulean. Expressing mitochondria-targeted mVenus with MCU-mCerulean revealed that some FRET was detected

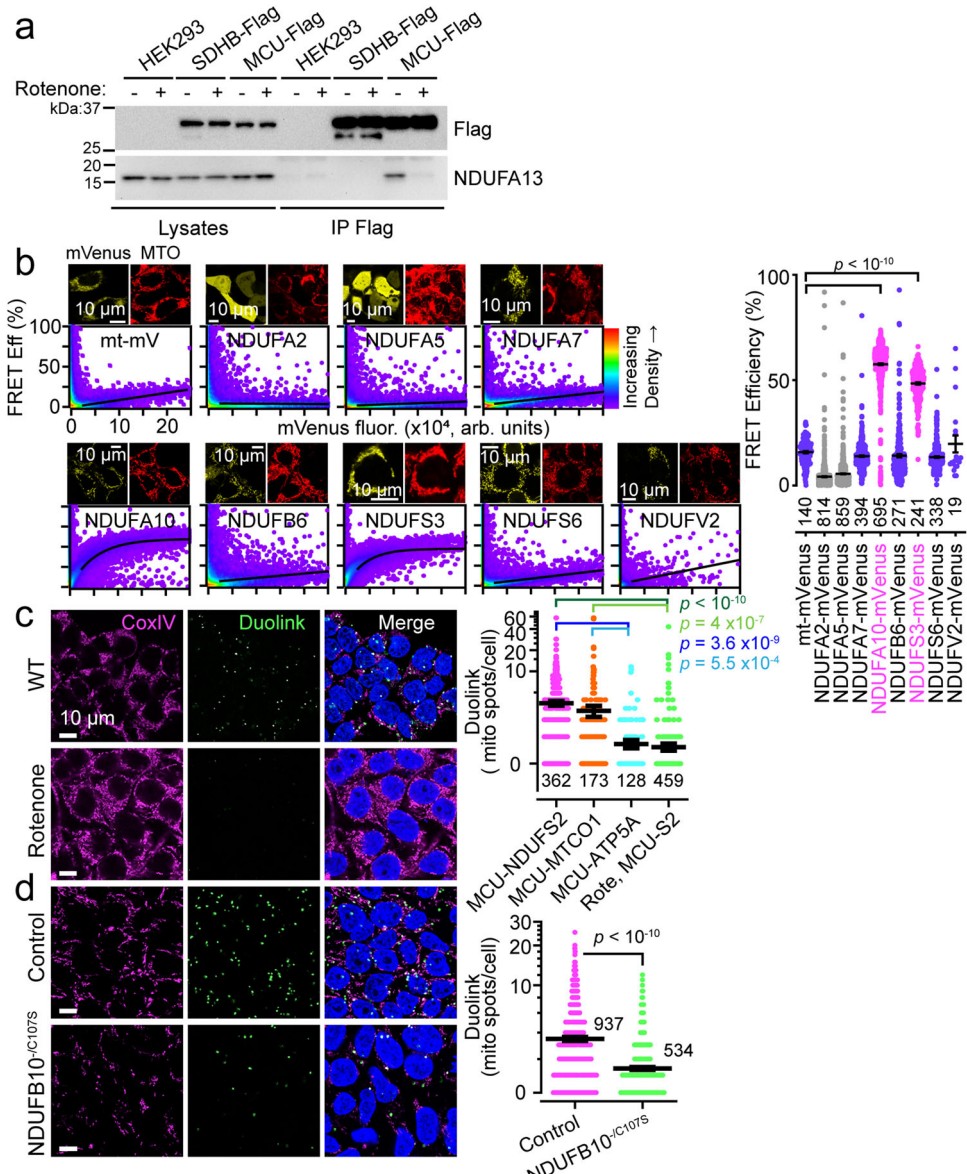

**Fig. 3 A direct interaction between MCU and Complex I alters uniporter stability. a** NDUFA13 co-immunoprecipitates with MCU-Flag. Images are representative of 3 separate trials. **b** Left, mVenus-tagged Complex I subunits surveyed for FRET with MCU-mCerulean via flow cytometry. Images above each graph show mVenus-tagged constructs and MitoTracker Orange (MTO). Within graphs, each point in the density plot is an individual cell expressing MCU-mCerulean and the corresponding mVenus-tagged construct. For each cell, the FRET efficiency between its mCerulean and mVenus is displayed in the y axis, while the degree of expression of the mVenus-tagged construct is revealed by the mVenus fluorescence in the x axis. Right, summary of FRET efficiency between MCU-mCerulean and the corresponding mVenus-tagged constructs at moderate mVenus expression. **c** Left, MCU-NDUFS2 Duolink colocalization occurs in mitochondria (CoxIV) and is more prevalent at baseline (WT) than after Complex I inhibition (Rotenone). Right, Duolink summary. Note that 74% of MCU-ATP5A and 85% of Rotenone-treated (MCU-S2) cells had zero Duolink spots, compared to 27% of MCU-NDUFS2 and 37% of MCU-MTCO1 cells. **d** Left, MCU-NDUFS2 Duolink greater in control than NDUFB10$^{-/C107S}$ IPSCs. Right, Summary. 45% of NDUFB10$^{-/C107S}$ IPSCs had zero Duolink spots, compared to 19% for control. Summary data are presented as mean values ± SEM. Statistics: (**b**, **c**) 1-way ANOVA followed by Bonferroni-corrected means comparisons; (**d**), two-sided Student's t test. Source data are provided as a Source Data file.

at high mVenus concentrations. Such concentration-dependent effects likely reflect the small volume of the matrix relative to cytoplasm. This FRET level served as the bound for spurious interactions, and a similar value was seen for NDUFA7, NDUFB6, NDUFS6, and NDUFV2, suggesting that these either were distant from MCU or had fluorophore orientations that minimized FRET. NDUFA10 and NDUFS3, however, demonstrated robust interaction with MCU-mCerulean, confirmed via acceptor photobleaching FRET assays as well (Fig. S22), implying close physical proximity.

Although our biochemical and FRET assays showed MCU-Complex I interaction, these required heterologous expression of tagged proteins. To confirm such interaction between endogenous molecules, we used the Duolink proximity ligation system, which stochastically produces bright fluorescent spots when target proteins less than ~40 nm apart are co-immunolabeled[47]. We used anti-MCU and anti-NDUFS2 monoclonal antibodies to label the uniporter and Complex I respectively in HEK293T cells and IPSCs. Controls with either antibody alone, or in MCU$^{KO}$ cells, displayed no Duolink spots (Fig. S23). When both antibodies were used, we

saw robust Duolink labeling ($3.6 \pm 0.3$ spots/cell, Fig. 3c). To confirm the interaction, we took advantage of the more lenient distance threshold detected by the Duolink system and used an antibody targeting Complex IV (MTCO1), since this is also part of the respirasome. This antibody also produced substantial, though less robust, Duolink signal, possibly because it is further from the putative MCU interaction site ($2.8 \pm 0.5$ spots/cell, Figs. 3c, S24). To show that the interaction was specific, we also performed the Duolink assay targeting Complex V (ATP5A), which is not part of the respirasome, and found substantially reduced labeling ($0.7 \pm 0.2$ spots/cell).

Having established that endogenous Complex I interacts with MCU, we investigated changes produced by Complex I dysfunction. Treating HEK293T cells with rotenone eliminated NDUFA13-MCU co-immunoprecipitation and markedly reduced Duolink targeting ($0.6 \pm 0.1$ spots/cell, Fig. 3a, c). Similarly, the NDUFB10$^{-/C107S}$ IPSCs had diminished labeling compared to control (Control: $3.5 \pm 0.1$ spots/cell, NDUFB10$^{-/C107S}$: $1.4 \pm 0.1$ spots/cell), despite preserved NDUFS2 and MCU (Figs. 3d, S25). In sum, based on immunoprecipitation, FRET, and proximity ligation, in both HEK293T cells and IPSCs, we find that MCU interacts with Complex I, but becomes decoupled when Complex I dysfunction ensues. Interestingly, an unexpected result from these assays was that the MCU-interacting NDUF subunits all clustered on the lateral surface of Complex I (Fig. S26). This is consistent with the architecture of the respirasome, since MCU would not be hindered by the Complex III dimer nor Complex IV, which reside on the opposite sides. Taken together, we establish an interaction between MCU and Complex I that is disrupted during Complex I dysfunction.

**Complex I-dependent protein turnover (CLIPT) controls MCU degradation.** A common finding in the multiple systems examined was an increase in uniporter protein consistent with enhanced $I_{MiCa}$ (Fig. 1c). Such increases could be mediated by either enhanced synthesis or diminished degradation. The absence of mRNA upregulation and the ability to enhance $I_{MiCa}$ in heterologously-expressed channels suggested the effect was likely not from greater synthesis. To evaluate MCU degradation, initial experiments with short-term cycloheximide treatment suggested a subtle change in MCU, but not MICU1 or EMRE lifetimes (Fig. S27). For longer term assessment cycloheximide is toxic, so we designed a tetracycline-repressible MCU-Flag construct and expressed it in MCU$^{KO}$ cells[48]. Cells were grown in 3% fetal bovine serum-supplemented media, to minimize proliferation (Fig. S28). Addition of 1 μg/mL doxycycline had no effect on respiration (Fig. S29), but suppressed MCU-Flag transcription, leading to depletion within two days in control cells. In contrast, upon treatment with rotenone, MCU-Flag expression persisted for the four-day experimental timeline (Figs. 4a, S30), confirming that reduced degradation of the uniporter was the primary mechanism for this effect. Inhibition of Complex III or IV produced no such effect on MCU. To confirm that stabilization was specific to MCU, we looked at another mitochondrial transmembrane protein regulated by ROS, ROMO1, and found that its lifetime failed to enhance after Complex I inhibition (Fig. S31).

At this stage, we considered two potential hypotheses linking ROS production in Complex I to stabilization of MCU (Fig. S32). The first, analogous to the cysteine mechanism described previously[37], would involve post-translational modification (PTM) of a specific NTD residue that increases MCU stability (PTM hypothesis). Although simple, concerns about this hypothesis include the promiscuous modifications ROS can induce on target peptides. Many of these are irreversible, and tend to damage rather than enhance protein activity. Moreover, while

ROS leaks from Complex I under physiological conditions, excess ROS produced during Complex I dysfunction induces self-inactivation[31,32]. Excess ROS produced by mito-miniSOG is also more likely to non-specifically damage Complex I, which has large matrix components and is quite sensitive to indiscriminate ROS production, than produce a specific modification on MCU. Finally, impaired Complex I can no longer bind MCU, and it is unclear how a Complex I ROS signal would modify channels no longer bound. Thus, we considered an alternate hypothesis, termed Complex I induced protein turnover (CLIPT). Here, under normal conditions MCU, via its NTD, interacts with Complex I, being turned over by quality-control proteases as oxidative damage from basal ROS leakage impairs MCU function. When Complex I becomes dysfunctional from excess ROS, MCU can no longer interact and buffer ROS, becoming more stable.

To distinguish the PTM and CLIPT mechanisms, we first assayed how depleting quality-control proteases would affect MCU stability. Under the PTM hypothesis, this should not affect MCU stability, as there should be little ROS-induced damage. Conversely, if MCU turnover depends on Complex I ROS, impairing quality control will stabilize MCU. We depleted several quality-control proteases using shRNA, and found that LONP1 depletion led to an increase in MCU stability (Fig. S33; % knockdown by qPCR: 88%, AFG3L2; 89%, CLPP; 93%, LONP1; 88%, SPG7, $n = 2$). LONP1 performs quality control of matrix proteins, and this effect is consistent with modifications of matrix-resident NTD leading to quality control, rather than transmembrane domain quality-control proteases AFG3L2 and SPG7. Moreover, a close interaction between LONP1 and MCU was also detected in the recently published mitochondrial protein-protein compendium[43]. Next, we examined how the NTD affected stability. Expressing the NTD fragment by itself should have no effect on MCU stability under basal conditions for the PTM hypothesis, whereas excess NTD should disrupt the MCU-Complex I interaction and stabilize the channel during CLIPT. We expressed an HA-tagged NTD fragment in cells also containing doxycycline-repressible MCU-Flag, and found that MCU stability increased, consistent with CLIPT (Fig. 4b).

To further confirm CLIPT, we designed the drug-induced dimerization experiment outlined in Fig. 4c. The FK506-binding protein (FKBP) binds the FKBP-rapamycin-binding domain (FRB) of MTOR only in the presence of rapamycin[49] and using this system allows us to determine if MCU-Complex I interaction is the key determinant of MCU stability, without having to create Complex I dysfunction. The MCU NTD was replaced with the similarly-sized FRB fragment, to generate a FRB-MCU fusion construct (Fig. 4d). For Complex I, we fused FKBP to NDUFA10 (NDUFA10-FKBP), the Complex I subunit showing strong FRET with MCU (Fig. 3b). As a control, we created a mitochondrially-targeted FKBP (mito-FKBP). First, we established that the system was functional. In MCU$^{KO}$ cells, FRB-MCU was able to confer $Ca^{2+}$ uptake (Fig. S34), showing this construct formed functional channels. In the absence of rapamycin, FRB-MCU and the FKBP constructs failed to interact (Fig. 4e), confirming the importance of the NTD in MCU-Complex I binding, whereas adding 100 nM rapamycin induced robust co-immunoprecipitation.

Having established functional rapamycin-induced interaction, we used the tetracycline-repressible system to examine if MCU turnover depended on its Complex I interaction (Fig. 4f). 100 nM rapamycin was added to cell culture dishes one day prior to adding doxycycline to repress FRB-MCU transcription. When co-expressed with mito-FKBP, FRB-MCU showed minimal degradation in the presence or absence of rapamycin. Similarly, when co-expressed with NDUFA10-FKBP, FRB-MCU protein was stable in the absence of rapamycin. Remarkably, however, when rapamycin was added in this condition, FRB-MCU was

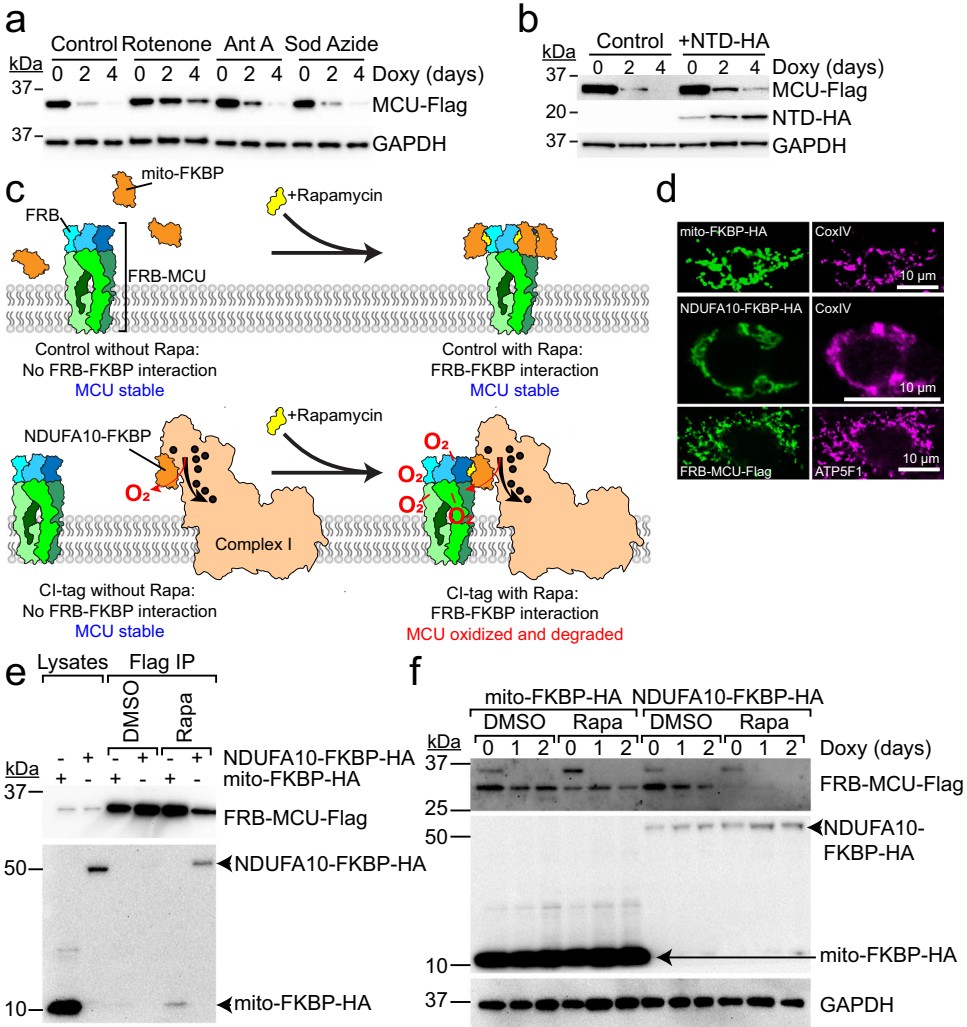

**Fig. 4 CLIPT controls MCU degradation. a** Doxycycline (doxy) treatment represses transcription of MCU-Flag. MCU-Flag persists after Complex I inhibition (rotenone) but not Complex III (antimycin A) or IV (sodium azide). **b** MCU-Flag stabilization induced by NTD peptide. **c** Design of rapamycin (Rapa)-induced dimerization experiment to test if MCU-Complex I interactions dictate MCU degradation. **d** Immunocytochemistry reveals co-localization of FRB-MCU, NDUFA10-FKBP, and mito-FKBP with mitochondrial markers CoxIV or ATP5F1. **e** With replacement of NTD with FRB, MCU only binds FKBP-tagged NDUFA10 in the presence of 100 nM rapamycin. Mito-FKBP-HA is a control. **f** FRB-MCU is stable in the absence or presence of rapamycin when cells co-express mito-FKBP-HA control, but rapidly degraded when rapamycin induces Complex I-binding in NDUFA10-FKBP-HA expressing cells. Images are representative of 3 (**a**, **b**, **f**) or 2 (**d**, **e**) separate trials. Source data are provided as a Source Data file.

rapidly degraded. Four implications arise from these results. First, the changes in MCU turnover are due to its interaction with Complex I, and not an off-target rotenone effect. Second, whereas a large fraction of full-length MCU degraded over 2 days under control conditions (Fig. 4a), removing the NTD conferred stability on the channel (Fig. 4f). Third, MCU turnover due to Complex I is not dependent on a specific modification of any particular NTD residue, since we could alter turnover in channels with the entire NTD replaced. Finally, MCU turnover appears to be tunable. Whereas enhancing the MCU-Complex I interaction with the FKBP-FRB-rapamycin system led to rapid MCU degradation, overexpressing the NTD alone stabilized it. Taken together, our results identify a mechanism, Complex I induced protein turnover (CLIPT), critical for controlling uniporter levels and activity.

**Uniporter stabilization during ETC impairment prolongs survival in mitochondrial cardiomyopathies.** Multiple studies over two decades have revealed that cardiac *Tfam* deletion in

mice produces many of the same clinical, biochemical, and ultrastructural features found in human mitochondrial cardiomyopathies[50–53]. Therefore, to examine whether uniporter stability is enhanced in vivo during disease, we turned to this model. For cardiac-specific deletion, we use the *Myh6-Cre* recombinase driver, which begins expressing embryonically[54]. When crossed with *Tfam*[loxP/loxP] animals, loss of myocardial TFAM leads to a 75% reduction in Complex I activity, with 30–35% inhibition of Complex III and IV, and early death between 3 and 6 weeks of age[12].

By crossing *Myh6-Cre* with *Mcu*[loxP/loxP] mice (*Mcu* KO), we could disrupt transcription of the *Mcu* gene, and assess its stability in *Mcu* KO, compared to mice with both *Mcu* and *Tfam* deleted (*Myh6-Cre*; *Tfam*[loxP/loxP]; *Mcu*[loxP/loxP] [*Tfam-Mcu* DKO]). By assaying for MCU protein persistence subsequent to Cre-mediated disruption of the *Mcu* gene, we could perform an in vivo experiment analogous to the transcriptional repression assays performed in cells (Fig. 4a). 10–14 day old mouse hearts were processed by Western blotting. Notably, despite embryonic initiation of *Mcu* deletion, we could detect substantial levels of

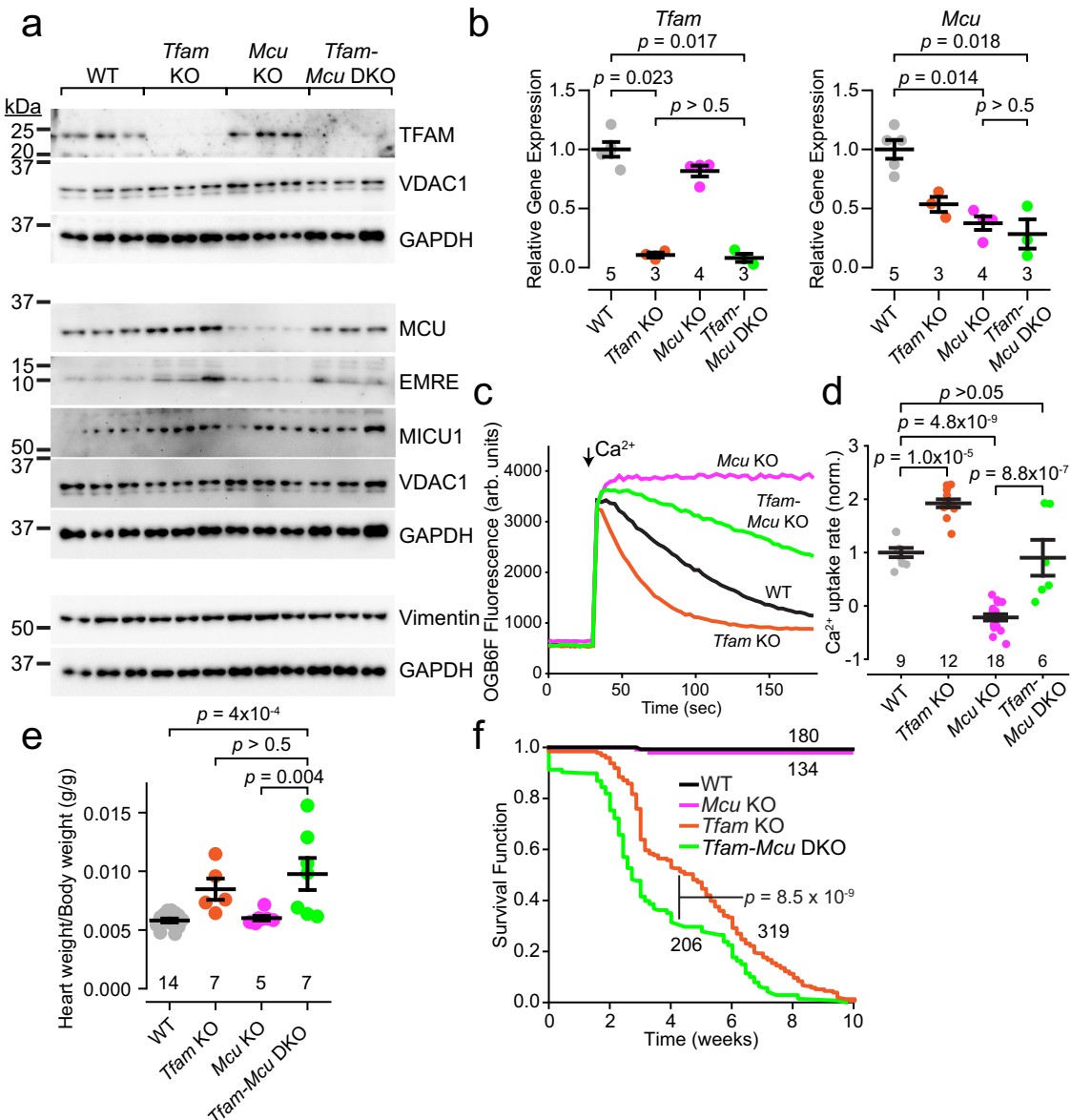

**Fig. 5 Diminished MCU degradation is responsible for enhanced uniporter activity in *Tfam* KO hearts, and prolongs survival. a** Immunoblotting for the specified proteins in P10-P14 mouse heart lysates. Samples from 3 mice shown per genotype. **b** Analysis of *Tfam* and *Mcu* transcripts in mouse hearts. **c** $Ca^{2+}$ uptake in isolated cardiac mitochondria incubated in Oregon Green BAPTA 6F (OGB6F). Arrow indicates 10 µM $Ca^{2+}$ pulse. **d** Summary of the normalized rate of $Ca^{2+}$ uptake after a 10 µM $Ca^{2+}$ pulse. **e** Heart weight to body weight ratios of P10-P14 mice of the indicated genotypes. **f** Kaplan–Meier survival analysis of the *Tfam* KO mice compared to *Tfam-Mcu* DKO mice. Comparison via a log rank test. Summary data are presented as mean values ± SEM. Statistics: (**b**) Kruskal–Wallis followed by Dunn's test with Bonferroni correction; (**d**, **e**) 1-way ANOVA followed by Bonferroni-corrected means comparisons; (**f**) Log-rank test. Source data are provided as a Source Data file.

uniporter subunit proteins in *Tfam-Mcu* DKO animals, approaching those seen in wild-type animals, compared with much lower levels in *Mcu* KO hearts (Figs. 5a, S35). This result suggests that, although its transcription has been disrupted, the MCU protein already present becomes much more stable in the ETC-deficient animals, compared to those with a functional ETC. This was not due to deficient Cre activity in the *Tfam-Mcu* DKO, as protein levels of TFAM were equally reduced between *Tfam* KO and *Tfam-Mcu* DKO hearts (Fig. 5a), *Tfam* mRNA transcripts were equally reduced between *Tfam* KO and *Tfam-Mcu* DKO hearts, and *Mcu* mRNA transcripts were equally reduced between *Mcu* KO and *Tfam-Mcu* DKO hearts (Fig. 5b). Moreover, this was also not due to MCU protein from excess infiltration of fibroblasts or other non-cardiac cells into the

myocardium. Non-cardiomyocyte cell quantities are lowest in juvenile hearts, and their mitochondrial mass per cell is far smaller than in cardiomyocytes, demonstrated by the trivial amount of TFAM left after its deletion in cardiomyocytes (Fig. 5a). A fibroblast marker, vimentin, was no different across the different genotypes (Fig. 5a). We also directly quantified cell amounts in histological slices. CellProfiler software was used to count the number of cellular nuclei in tissue slices from Masson's trichrome-stained hearts. *Mcu* KO and *Tfam-Mcu* DKO hearts had similar numbers of nuclei per mm[2] of tissue, these tended to be lower compared to wild-type animals, and no obvious excess non-cardiomyocyte infiltrates were noted (Fig. S36). Therefore, as in cultured cells, disruption of the ETC in vivo in mouse myocardium led to increased stability of the uniporter.

To determine if these persistent uniporter channels were functional, we measured $Ca^{2+}$ uptake in mitochondrial fractions isolated from mouse hearts. We used fluorescent $Ca^{2+}$ imaging, as this allows an integrative assessment of $Ca^{2+}$ uptake in the context of TFAM deletion. Decreasing Oregon Green BAPTA 6F fluorescence following a $Ca^{2+}$ pulse indicated mitochondrial $Ca^{2+}$ uptake (Fig. 5c, d). Wild-type mice took up the $Ca^{2+}$ pulse rapidly, whereas *Mcu* KO mice were unable to take up $Ca^{2+}$, consistent with loss of functional uniporter. Most of the *Tfam-Mcu* DKO mice, however, had persistent $Ca^{2+}$ uptake, revealing preservation of functional uniporter channels, despite deletion of the *Mcu* gene.

Next, we assessed if loss of MCU impaired the health of *Tfam-Mcu* DKO mice. Of note, though MCU levels and activity persisted in the juvenile mice, as noted above, these were not at wild-type levels (Fig. 5a, d). Moreover, during the second postnatal week there appeared to be a reduction in the persistent uniporter channels, as a subset of *Tfam-Mcu* DKO mice were no longer capable of cardiac mitochondrial $Ca^{2+}$ uptake (lower points in Fig. 5d). Thus, we were still able to assess how partial depletion of MCU altered cardiac status. The *Tfam-Mcu* DKO developed a cardiomyopathy similar to *Tfam* KO mice. They had enlarged hearts (Fig. 5e), with reduced contractile function, dilated ventricles, and thinned walls on echocardiography (Fig. S37). Remarkably, even partial loss of uniporter channels proved fatal, as a steep decline in survival occurred during the second postnatal week (Fig. 5f), with *Tfam-Mcu* DKO mice dying 1.5 weeks earlier than *Tfam* KO mice (survival time in weeks, average [95% confidence interval]: *Tfam-Mcu* DKO, 3.2 [2.9–3.5]; *Tfam* KO, 4.8 [4.5–5.0]). In summary, in a disease model of cardiac ETC dysfunction, loss of the uniporter further impaired survival.

**NTD overexpression improves survival and function in Complex I-impaired *Drosophila*.** Complex I deficiency is the most common cause of monogenic mitochondrial disorders, and is frequently implicated in neurological and cardiac disease[55]. To further explore the physiological relevance of the MCU-Complex I interaction. *Drosophila* is an ideal system, as models for Complex I disease exist[29], the mitochondrial $Ca^{2+}$ uptake machinery is closely conserved[56], and crosses can be rapidly generated. For these analyses, Complex I was inhibited in *Drosophila* flight muscle, which possesses sarcomeric organization and mitochondrial $Ca^{2+}$ uptake that mimics mammalian cardiomyocytes[57]. Complex I dysfunction in NDUFB10[RNAi] led to mild developmental lethality and weak flies (Fig. 6a, b). To test flight muscle, the time it takes *Drosophila* to fly off a platform sitting in water is measured (Supplementary Videos 1–9). On this island assay, NDUFB10[RNAi] flies took longer to escape compared to controls. We also analyzed the recently-described *Drosophila* whole-body MCU knockout (MCU[1])[56]. These flies have no ruthenium-red sensitive $I_{MiCa}$ (Fig. S38), and show neither developmental lethality nor flight weakness[56]. When crossed with NDUFB10[RNAi], however, there was a clear genetic interaction. Double-mutant flies suffered severe developmental lethality and flight muscle weakness (Fig. 6a, b). Notably, these impairments were entirely rescued, reverting NDUFB10[RNAi] flies to near wild-type function, by re-expressing a full-length MCU[56]. To test the importance of the NTD, we created a *Drosophila* ΔNTD-MCU transgene, which targeted mitochondria and was functional in muscle (Fig. S39). This construct, however, entirely failed at rescue, reinforcing the importance of the MCU-Complex I interaction. To confirm the importance of $Ca^{2+}$ uptake through the uniporter, we also tested mutant flies that express a flight muscle-restricted pore mutant of MCU (MCU[DQEQ]) that inhibits $Ca^{2+}$ transport in a dominant-

negative fashion[58]. These *Drosophila* developed in expected numbers and had no flight impairment compared to controls. Here too, a clear genetic interaction was noted between MCU[DQEQ] and NDUFB10[RNAi], with double-mutant *Drosophila* having substantial developmental lethality and the survivors being impaired in flight (Fig. 6c, d). This genetic interaction was not specific to NDUFB10[RNAi] nor was there a threshold for Complex I impairment necessary, as we saw a similar, milder pattern with NDU-FA13[RNAi], which produces a much weaker level of Complex I deficiency (Fig. S40)[29]. Therefore, in *Drosophila* as in *Tfam* KO mice, uniporter activity was necessary to preserve cellular function when Complex I is impaired.

The NTD is critical for the functional, biochemical, and genetic interaction between MCU and Complex I. Moreover, expressing the NTD fragment alone stabilized the uniporter (Fig. 4b), which appears necessary for maintaining homeostasis during Complex I impairment. Reasoning that this strategy may alleviate the phenotype caused by Complex I dysfunction, we generated a construct encoding the isolated *Drosophila* NTD fragment, which also targeted mitochondria (Fig. S39). When expressed in NDUFB10[RNAi] flies, the NTD improved both survival (male flies) and flight (Fig. 6e, f), though the rescue was not as complete as expressing full-length MCU. Taken together, these results imply that targeting the MCU NTD may be a novel strategy for treating Complex I impairment.

## Discussion

In this report, we identify a functional, biochemical, and genetic interaction between Complex I and the uniporter, that helps maintain bioenergetic homeostasis when Complex I becomes impaired. In deciphering how Complex I dysfunction enhances uniporter levels, we identified CLIPT as a mechanism for MCU protein turnover (Fig. 7), and show that it may be exploited to preserve organismal function during Complex I deficiency, a pathology common to varied diseases.

In deciphering how Complex I dysfunction enhances uniporter levels, we revealed the Complex I interaction is the key determinant for MCU protein turnover, which we term CLIPT. In this process, MCU binding to Complex I leads to uniporter degradation, most likely via oxidative damage from physiological ROS generated in Complex I. Unlike other forms of post-translational MCU regulation, there does not appear to be a specific residue modification driving CLIPT. Rather, as long as MCU and Complex I interact, CLIPT occurs. Aberrant ROS production during Complex I dysfunction may either prevent the MCU-Complex I interaction by modifying the binding site, or lead to Complex I inactivation and disassembly. In either case, enhancement of uniporter levels occurs due to reduced MCU degradation. The fraction of MCU binding Complex I likely varies between tissues and cell types based on their abundances, localization, activity of quality-control proteases, and possibly other factors. In cardiomyocytes, approximately 1/3 of MCU protein is located within cristae, suggesting this fraction may be interacting with Complex I. It remains to be seen if CLIPT controls the turnover of other mitochondrial proteins as well.

Prior studies had hinted at a relationship between the uniporter and Complex I, with biochemical interaction found in several proteomic compendia, and evidence of reduced Complex I levels in *Mcu* KO hearts[42,43,59]. We find the MCU-Complex I relationship is evident in *Drosophila*, mouse, and human mitochondria. Notably, such strong evolutionary conservation is reinforced when comparing the species distribution of the uniporter and Complex I, which substantially overlap[60]. Both the uniporter and Complex I are present in animals, plants, and trypanosomes, but absent in yeast.

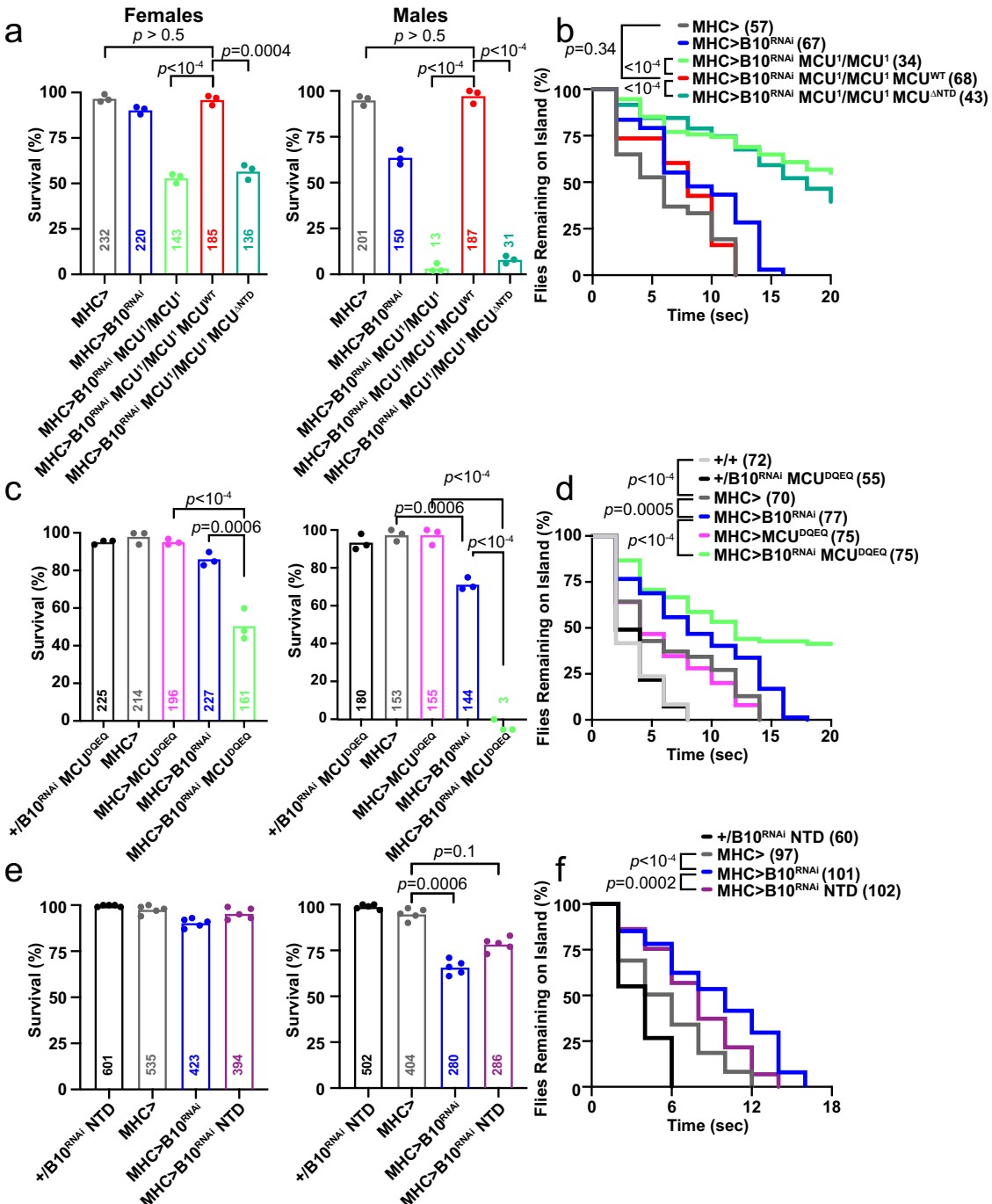

**Fig. 6 Genetic interaction between MCU and Complex I in *Drosophila*. a, c, e** *Drosophila* survival for the indicated genotypes and sex. **b, d, f** Quantification of dwell time in the island assay for indicated genotypes for female flies. **a, b** The muscle specific *MHC*-GAL4 was used to drive expression of *NDUFB10*[RNAi], *MCU*[WT], or *MCU*[ΔNTD] in wild-type or *MCU*[1]/*MCU*[1] mutant flies, as indicated. **c, d** As in (**a, b**), except the dominant-negative pore mutant *MCU*[DQEQ] was expressed with *MHC*-GAL4. **e, f** As in (**a, b**), except the isolated NTD fragment was expressed with *MHC*-GAL4. *p < 0.05, **p < 0.01, ***p < 0.001. Statistics: (**a, c, e**) Fischer's Exact Test with Bonferroni correction; (**b, d, f**) Log-rank (Mantel-Cox) test with Bonferroni correction. Source data are provided as a Source Data file.

For the uniporter to undergo CLIPT an intact MCU N-terminal domain is required. The NTD has previously been implicated in binding matrix cations to inactivate the channel[40], and protein–protein interactions[38,61,62]. We found that NTD deletion fully abolished the enhancement in Ca$^{2+}$ current seen after Complex I inhibition, whereas overexpression of the NTD by itself made endogenous uniporter channels more degradation-resistant. This ability to enhance endogenous uniporter stability led us to examine the benefits of isolated NTD expression. In proof-of-concept studies, we show that expression of the isolated NTD in models of Complex I impairment can improve muscular function and survival. Targeting this highly-conserved relationship may be a potential therapy in disorders that feature Complex I dysfunction.

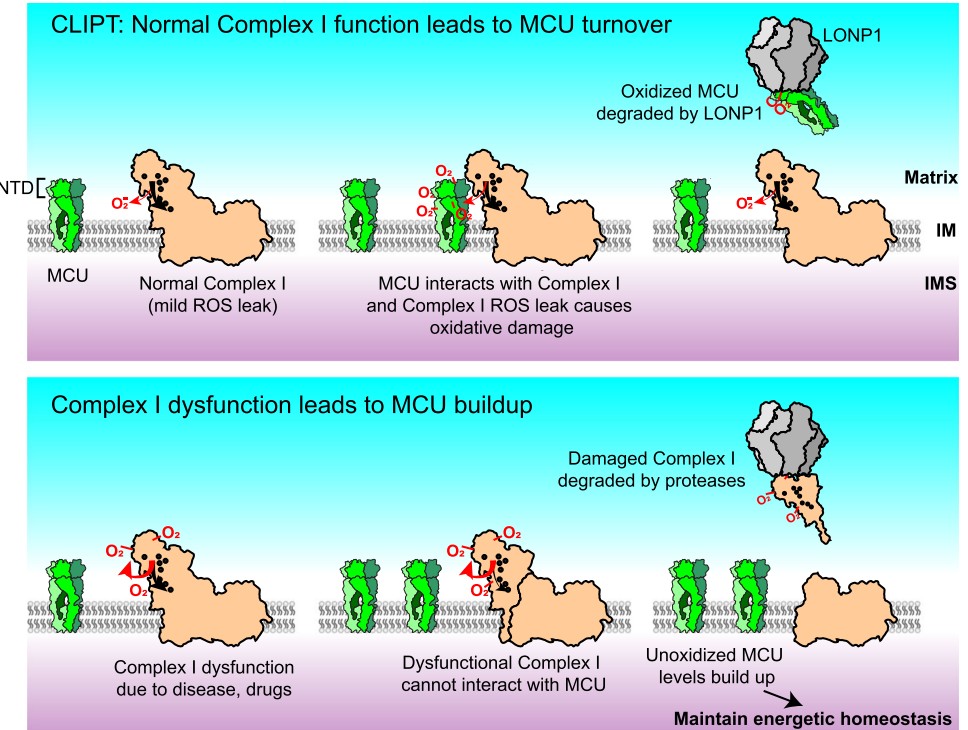

**Fig. 7 Current hypothetical model for Complex I-induced protein turnover of MCU.** Top, under physiological conditions, MCU interacts with Complex I and is oxidized by the mild ROS leak produced by Complex I. Such oxidized MCU becomes damaged and degraded by LONP1 or other quality-control proteases, leaving Complex I available to interact with additional channels. We term this process Complex I-induced protein turnover (CLIPT). Bottom, when Complex I becomes impaired or misassembled, it produces excessive ROS and self-inactivates. Such dysfunctional Complex I can no longer interact with MCU, nor damage it with basal ROS leak, and is cleared by housekeeping proteases CLPP and LONP1. Thus, functional MCU levels build up, and additional Ca$^{2+}$ influx through these channels maintains energetic homeostasis.

## Methods

**Ethics.** The research complies with all relevant ethical regulations. All animal procedures have been reviewed and approved by the Institutional Animal Care and Use Committee at the University of Utah under protocol 19-01004.

**Antibodies.** ATP5F1 (ab117991, Abcam, 1:1000), β-Actin (ab8224, Abcam, 1:1000), COX IV-Alexa Fluor 488 (4853 S, Cell Signaling Tech [CST], 1:200), EMRE (A300-BL19208, Bethyl, 1:1000), FLAG HRP (A8592, Sigma, 1:1000), FLAG magnetic beads (M8823, Sigma), FOXRED1 (sc-377264, Santa Cruz, 1:1000), GAPDH (2118 S, CST, 1:1000), GFP (ab290, Abcam, BN-PAGE: 1:1000), goat anti-mouse Alexa Fluor 555 (A21422, ThermoFisher, 1:100), goat anti-rabbit Alexa Fluor 488 (A32731, ThermoFisher, 1:100), HA (3724 S, CST, 1:800), HA HRP (12013819001, Sigma, 1:1000), MCU (14997 S, CST, WB 1:1000, ICC 1:100, DuoLink 1:200), MICU1 (12524 S, CST, 1:1000), MICU1 (HPA037480, Sigma, 1:1000), MTCO1 (ab14705, Abcam, 1:100), NDUFA13 (ab110240, Abcam, 1:1000), NDUFB10 (ab196019, Abcam, 1:1000), NDUFS2 (ab110249, Abcam, WB 1:1000, ICC 1:100, DuoLink 1:200), NDUFS3 (ab177471, ICC 1:200), NDUFS4 (ab137064, WB 1:1000), NDUFS4 (PA5-21677, ThermoFisher, BN-PAGE: 1:300), Oct4 (ab19857, Abcam, 1:200), ROMO1 (TA505580, Origene, 1:1000), Sox2 (5024, CST, 1:200), TOM20 (42406 S, CST, BN-PAGE: 1:800), VDAC1 (ab14734, Abcam, 1:1000), Vimentin (ab92547, Abcam, 1:1000), ZO-1 (33-9100, ThermoFisher, 1:200).

**Plasmids for stable expression, transient transfection, and short hairpin RNA.** Newly-derived plasmids were created using the NEB HiFi DNA Assembly Cloning Kit (Ipswich, MA), and will be deposited at Addgene unless restricted by Material Transfer Agreements. Plasmids were verified by Sanger sequencing.

pUC57-mito-*Lb*NOX (Addgene #74448), pLYS5-SDHB-Flag (Addgene #50055), pLYS1-MCU-Flag, pLYS1-ΔNTD-MCU-Flag (removing amino acids 58–186), and pLYS1-Cys-Free-MCU-Flag were gifts from Vamsi Mootha[23,25,63,64]. pGP-CMV-GCaMP6m was a gift from Douglas Kim & GENIE Project (Addgene #40754). MiniSOG-Mito-7 (Addgene #57773) was a gift from Michael Davidson[35]. pCW57.1-MAT2A (Addgene #100521) was a gift from David Sabatini[48]. CFP-FKBP (Addgene #20160) and YFP-FRB (Addgene #20148) were gifts from Tobias Meyer[65]. pUASg.attB was a gift from Johannes Bischof and Konrad Basler. SoNar was a gift from Yi Yang[34].

Mito-*Lb*NOX was expanded from pUC57-mitoLbNOX and cloned into pLenti-CMV-puro for lentiviral generation. GCaMP6 was cloned into pLenti-CMV-puro with an N-terminal Cox8 mitochondrial targeting sequence and a C-terminal mCherry tag. Human SOD2 was expanded from a HEK293T cDNA library and cloned into pLenti-CMV-puro with a C-terminal HA tag. The mitochondrial targeting sequence from ABCB10 (aa 1–193) was expanded from a HEK293T cDNA library and cloned in-frame upstream of SoNar to create mitochondrially-targeted MitoSoNar. MCU-Flag was shuttled from pLYS1-MCU-Flag into pCW57.1-MAT2A to create pCW57.1-MCU-Flag. The MCU NTD (aa 1–185) was expanded from pLYS1-MCU and cloned into pLenti with an HA-tag at its C-terminus. FRB-MCU was generated by incorporating the FRB fragment immediately downstream of the mitochondrial targeting sequence within ΔNTD-MCU-Flag, and placing this in the pCW57.1-MAT2A plasmid. For mito-FKBP, the FKBP fragment was placed between a 4-fold repeat of the COX8A mitochondrially targeting sequence and an HA tag in pLenti-CMV-puro. NDUFA10-FKBP was created similarly, except the full NDUFA10 sequence was used instead of the mitochondrial targeting sequence. *Drosophila* MCU lacking amino acids 56–182 was cloned (UniProt Q8IQ70), along with a C-terminal HA tag, from pAC5.1[62] into pUASg.attB to create ΔNTD-MCU-pUASg.attB. A similar strategy was used to incorporate the *Drosophila* MCU NTD fragment (amino acids 1–182), creating NTD-pUASg.attB.

For FRET experiments, mitochondria-targeted mVenus (mt-mVenus) and mCerulean (mt-mCerulean) were created by adding four copies of the human COX8 mitochondrial targeting sequence to the N-terminal of mVenus in the pLenti-CMV-puro backbone. The sequence for MCU was expanded from pLYS1-MCU-Flag and cloned into the mCerulean-pLenti vector in place of the mitochondrial targeting sequence. Sequences for NDUFA2, NDUFA5, NDUFA7, NDUFA10, NDUFS3, NDUFS6, NDUFB6, and NDUFV2 were expanded from a HEK293T cDNA library via polymerase chain reaction and cloned N-terminal to mVenus in a pEGFP vector. For FRET calibration, we also cloned mitochondrially-targeted mVenus-mCerulean dimers separated by linkers of 5, 43, and 236 amino acids into pLenti-CMV-puro to create mito-C5V, mito-C43V, and mito-CTV.

Short hairpin RNA lentiviral constructs were obtained from The RNAi Consortium (Sigma). TRC IDs for shRNA plasmids are TRCN0000412514 (AFG3L2), TRCN0000046859 (CLPP), TRCN0000310154 (LONP1), TRCN0000046793 (LONP1), TRCN0000063825 (SPG7). The control was shGFP (SHC005).

**Cell culture and generation of stable lines.** HEK293T cells were grown in Dulbecco's Modified Eagle Medium (DMEM) (ThermoFisher) with 10% fetal

bovine serum, penicillin (100 I.U./ml), streptomycin (100 µg/ml). Cells that had genetic or pharmacological inhibition of Complex I (and their respective controls) were also supplemented with 0.4 mM uridine and 2 mM pyruvate. For pharmacological inhibition, media was supplemented with 0.1–3 µM rotenone (Complex I), 1 µM Antimycin A (Complex III), or 200 µM sodium azide (Complex IV), as indicated, 48–72 h prior to use. For generation of stable lines, cells were transduced with lentivirus and selected with 1.5 µg/mL puromycin or 10 µg/mL blasticidin.

NDUB10 and NDUFS4 knockout HEK293T cells were a kind gift of Michael Ryan[24]. HEK293T cells were a kind gift from David Clapham.

HEK293T genetic fingerprinting validation is via short tandem repeat analysis in September 2019, performed at the University of Utah DNA Sequencing Core.

**FOXRED1 knockout line generation.** HEK293T cells were electroporated (Neon Transfection System, MPK5000, ThermoFisher) with Alt-R Sp HiFi Cas9 nuclease v3, tracrRNA, and FOXRED1 crRNA (Hs.Cas9.FOXRED1.1.AA and Hs.Cas9.-FOXRED1.1.AC), all from Integrated DNA Technologies (Coralville, IA) per the manufacturer's instructions. Single clones were expanded and plated in both DMEM + 10% FBS media containing 0.4 mM uridine, 2 mM pyruvate and either 25 mM glucose or 10 mM galactose. Clones that failed to grow in galactose had genomic DNA Sanger sequenced for evidence of editing, and clones were subject to Western blot to confirm FOXRED1 deletion, and one of these that appeared to grow well was further studied.

**Patient-derived induced pluripotent stem cells.** Fibroblasts were obtained postmortem from an infant with NDUFB10 deficiency (NDUFB10$^{-/C107S}$) and studied in accordance with Colorado Multiple Institutional Review Board protocol 16–0146 and with parental approval[28]. No monetary compensation was provided. Ethical approval for induced pluripotent stem cell studies is through the University of Utah Institutional Review Board IRB_00088011. Fibroblasts were reprogrammed into IPSCs using CytoTune-iPS Sendai Reprogramming vector (A16517, ThermoFisher), and cultured in Essential 8 stem cell media (A1517001, ThermoFisher) supplemented with 2 mM pyruvate, 0.4 mM uridine, and 1 mM N-acetylcysteine. At around Day 20 post-transduction, newly formed IPSC colonies were transferred to new plates and expanded for another month. For quality control of IPSCs we performed four standard assays to determine (1) pluripotent status, (2) genetic fingerprinting, (3) karyotype and (4) absence of bacterial or mycoplasma contamination. Pluripotent status was determined by immunofluorescence against Oct4, Sox2, and ZO-1, and counterstained with Hoechst dye prior to imaging on a fluorescent EVOS Cell Imaging System (ThermoFisher). IPSC genetic fingerprinting is via short tandem repeat analysis. Molecular karyotyping of genomic DNA is via an nCounter Human Karyotype Assay (Nanostring Technologies Inc.). Contamination testing is via an e-Myco PLUS Mycoplasma PCR Detection Kit (iNtRON Biotechnology) and an PCR Bacteria Test Kit (PromoCell GmbH).

Control and NDUFB10$^{-/C107S}$ IPSCs were cultured in 6-well culture plates coated with vitronectin (A14700, ThermoFisher) in StemFlex medium (A3349401, ThermoFisher) medium containing 10 µM Y-27632 ROCK inhibitor (S1049, SelleckChem, Houston, TX), 2 mM pyruvate, and 0.4 mM uridine, changed daily.

**Mouse strains and survival.** All animal procedures have been reviewed and approved by the Institutional Animal Care and Use Committee at the University of Utah under protocol 19-01004. Tfam$^{loxP/loxP}$ mice were developed by Nils-Göran Larsson[13] and obtained from Ronald Kahn[66]. Mcu$^{loxP/loxP}$ mice were obtained from John Elrod[18]. Myh6-Cre transgenic mice were obtained from the Jackson Laboratory (Bar Harbor, ME, stock # 011038)[67]. Animals were kept on a C57BL/6 J background. Animals were housed under standard conditions and allowed free access to food and water. Animals used for experiments were 10–14 days old. Heart and body weights were recorded at time of euthanasia. For survival analysis, a Kaplan-Meier curve was constructed, censoring animals used for experiments.

**Fly Stocks and maintenance.** Studies on transgenic Drosophila are approved by the University of Utah Institutional Biosafety Committee. Fly rearing was done on standard cornmeal/yeast/molasses food prepared in a central kitchen at the University of Utah. The following Drosophila melanogaster strains were used: w Berlin control strain (obtained from Adrian Rothenfluh, University of Texas Southwestern Medical Center, Dallas, TX); w; MHC-GAL4/TM6C,Sb, w$^{1118}$; da-GAL4, yw; UAS-NDUFB10$^{RNAi}$, and yw; UAS-NDUFA13$^{RNAi}$ (obtained from Edward Owusu-Ansah, Columbia University; Bloomington Drosophila Stock Center numbers 55133, 29592, and 43279)[29]; w; UAS-MCU$^{D206Q,E263Q}$ (obtained from Ronald Davis, The Scripps Research Institute Florida; Jupiter, FL)[58]; w; MCU$^1$/ MCU$^1$ and w; UAS-MCU$^{WT}$ (obtained from Alex Whitworth, University of Cambridge, Cambridge, UK)[56]. PhiC31 integrase-mediated transgenic flies were generated by BestGene (Chino Hills, CA) from NTD-pUASg.attB. Integration site was at attP40, as with UAS-MCU$^{WT}$. Genomic PCR confirmation of insertion and subsequent balancing were performed by BestGene. Flies were reared at 25 °C in a 12:12 light/dark cycle. Fly genotypes are listed in Supplementary Table 1.

**Mitochondrial and mitoplast isolation.** Mitochondria were isolated from cultured cells or mouse hearts by differential centrifugation, and mitoplasts prepared by using a French Press (Sim-Aminco) to disrupt outer membranes[12,15]. For

Drosophila mitochondrial isolation, lines were raised at 20 °C. Adult flies >3 days of age after eclosion were used. Mitochondria were prepared from thoraces (devoid of legs and wings) of >50 flies. Thoraces were homogenized with a Potter-Elvehjem grinder set to 250 rpm, and the remainder of the protocol was as described above[15].

**Electron transport chain assays.** These were performed using established protocols[12]. Briefly, citrate synthase activity in isolated mitochondria was measured using the Citrate Synthase Assay Kit (CS0720, Sigma) following the manufacturer's instructions. Samples of equivalent citrate synthase activity were used for subsequent ETC assays. We measured Complex I activity by tracking changes in NADH absorbance at 340 nm in a Cytation 5 microplate reader (Biotek, Winooski, VT). Reactions were performed in a solution containing 50 mM potassium phosphate buffer (pH to 7.5), 3 g/L fatty-acid free BSA, 300 µM KCN, 500 µM NADH, and ~5 µg of mitochondria (exact amount adjusted based on citrate synthase activity). Mitochondria were subjected to 3 freeze-thaw cycles before assay. 60 µM ubiquinone was added to initiate the reaction, and complex I activity was calculated by measuring the difference in absorbance slopes in the presence or absence of 10 µM rotenone. We measured complex III activity by tracking changes in cytochrome c absorbance at 550 nm. Reactions were performed in a solution containing 25 mM potassium phosphate buffer, 75 µM oxidized cytochrome c, 500 µM KCN, 100 µM EDTA, 0.025% (w/v) Tween-20, and ~10 µg of mitochondria (exact amount adjusted based on citrate synthase activity). 100 µM of decylubiquinol was added to initiate the reaction and complex III activity was calculated by measuring the difference in absorbance slopes in the presence or absence of 18 µM antimycin A. We measured complex IV activity by tracking changes in cytochrome c absorbance at 550 nm. Reactions were performed in a solution containing 50 mM potassium phosphate buffer (pH 6.8), 100 µM reduced cytochrome c, and 0.5% Tween-20. ~10 µg of mitochondria (exact amount adjusted based on citrate synthase activity) was added to initiate the reaction and complex IV activity was calculated by measuring the difference in absorbance slopes in the presence or absence of 300 µM KCN.

**Electrophysiology.** Whole-mitoplast electrophysiology was performed using established protocols[15]. For Drosophila, we used MHC-Gal4 driven expression to measure changes in flight muscle I$_{MiCa}$. To confirm with electrophysiology that transgenes were expressing, we used the global Da-Gal4 driver for ΔNTD-MCU I$_{MiCa}$ in MCU[1] flies, and isolated MCU NTD in wild-type flies. Borosilicate glass pipettes with a resistance of 15–20 MΩ were used. Whole-mitoplasts currents were acquired at 5 kHz and filtered at 1 kHz using an Axopatch 200B amplifier (Molecular Devices, San Jose, CA). Mitoplasts had a capacitance 0.2–3.7 pF. Pipette solution was composed of (mM): Na-Gluconate 150, HEPES 10, EDTA 1, EGTA 1, pH 7.2, brought up to 320–340 mOsm with D-Mannitol. Bath solution for inner membrane anion channel (mM): 150 KCl, 10 HEPES, 1 EGTA. Bath solution for mitochondrial calcium uniporter (mM): Na-Gluconate 150, HEPES 10, 5 CaCl$_2$, pH 7.2. A ~12 mV junction potential was cancelled on switching between solutions. Ruthenium red (1 µM) was added to block uniporter-specific currents. For display purposes, capacitance transients caused by changing levels of solutions in the bath have been removed and the Simplify filter from Adobe Illustrator has been used to reduce the number of points, without altering the shape of the traces. Analysis was performed using pClamp v10 (Molecular Devices).

**Quantitative reverse transcription-polymerase chain reaction (qPCR) expression analysis.** Quantitative PCR from IPSC, HEK293T cells, and mouse hearts was performed using Power SYBR Green PCR Master Mix (ThermoFisher)[12]. SuperScript VILO (ThermoFisher) was used to prepare cDNA. Quantification of gene expression was performed on a 96-well BioRad CFX Connect Real-Time PCR Detection System (BioRad, CA, USA). Analysis was performed by using the 2$^{-\Delta\Delta Ct}$ method using GAPDH as reference. For HEK293T cell culture qPCR, N refers to technical replicates. Primers were obtained via NCBI Primer-BLAST, Primerbank, or prior reports and are listed in Supplementary Table 2[68,69].

**ΔΨ imaging.** HEK-293T cells grown on poly-lysine-coated glass coverslips. DMEM containing 10 mM galactose was added one hour prior to experiments. Cells were loaded with 20 nM tetramethylrhodamine methyl ester (TMRM, ThermoFisher) for 30 min at 37 °C. The media was changed to phosphate-buffered saline or a solution containing (in mM): 125 KCL, 20 HEPES, 5 K$_2$HPO$_4$, 1 MgCl$_2$ and cells were imaged as described in the section on confocal imaging at 548 nm excitation/574 nm emission. Analysis was in ImageJ.

**Western blots.** Cells or tissue were lysed in RIPA buffer containing Halt protease/ phosphatase inhibitors (78440, ThermoFisher). Protein concentration was determined by BCA assay (23227, ThermoFisher). 5–20 µg of protein from cell lysates were loaded on polyacrylamide gels and immunoblotted with the indicated antibodies[62]. Images were collected using ImageLab 6.1 (Bio-Rad). ImageJ was used for band quantification.

**Blue-native PAGE.** Methods were adapted from established protocols using the NativePAGE kit (BN2008, ThermoFisher)[23]. After mitochondrial isolation from

HEK-293T cells, 10–25 ug of protein was solubilized with 1% digitonin and 1x NativePAGE sample buffer. After centrifuging the sample for 30 min at 16,000 g, 4 °C NativePAGE G-250 Sample Additive was added to a final concentration of 0.1%. Gel electrophoresis was performed according to the manufacturer's protocol for the Invitrogen NativePAGE Novex Bis-Tris Gel System (BN1004BOX, ThermoFisher), and proteins were transferred to PVDF membranes using a TransBlot Semi-Dry system (Bio-Rad). Membranes were fixed in 8% acetic acid, washed, and then air-dried. After rehydration with ethanol or methanol, membranes were blocked and immunoblotted.

**Blue light assay for miniSOG.** HEK293T cells expressing mito-miniSOG at >70% confluence were exposed to blue light from an LED array for 10 min at RT. Longer periods of blue light exposure would often lead to substantial cell death. Mitoplast isolation as described above was performed 48–72 h later. Cells with the same construct without exposure to blue light were used as control.

**Flow cytometric measurements.** Data was collected at the Flow Cytometry Core Facility at the University of Utah. For mitochondrial ROS measurements, cells were incubated with 5 µM MitoSOX (M36008, ThermoFisher) at 37 °C for 15–30 min. For mitochondrial NADH/NAD$^+$ measurement, cells were transfected 24–48 h prior to analysis with the MitoSoNar construct. Cells were analyzed with a BD FACSCanto Analyzer running FACSDiva 6 software (both BD Biosciences, San Jose, CA). For mitochondrial Ca$^{2+}$, mito-GCaMP6m was measured using 488 nm excitation laser and 530/30 nm emission filters and mCherry was measured using a 561 nm laser and 585/15 nm emission filter. To obtain maximum and minimum fluorescence ratios, wild-type cells expressing the construct were analyzed after incubating in 10 µM ionomycin and either 5 mM Ca$^{2+}$ (maximum ratio) or 5 mM EGTA (minimum ratio). Ratios were transformed into Ca$^{2+}$ concentrations using published methods, assuming a $K_D$ of 167 nM[70,71]. For MitoSOX, we used a 488 nm excitation laser and 585/15 nm emission filter. For MitoSoNar, we used the ratio of 525/50 nm emissions when excited by either 405 nm or 488 nm laser lines after background correction in untransfected cells. For analysis, live, single cells were selected using forward and side-scatter parameters. Gating strategy exemplar is shown in Fig S41. Analysis was performed on FlowJo (v10.6, BD Life Sciences, Ashland, OR) and OriginPro (OriginLab).

***Lb*NOX cell proliferation assays.** One hundred thousand HEK293T WT, MCU KO, mito-*Lb*NOX, or MCU KO mito-*Lb*NOX cells were seeded in a 6-well dish containing 2 mL DMEM (ThermoFisher) with 10% FBS (Life Technologies), penicillin (100 I.U./ml), streptomycin (100 µg/ml), 1x Glutamax (ThermoFisher), 0.4 mM uridine (Fisher Scientific), and either 0 or 500 nM piericidin (Cayman Chemical). Medium was exchanged every twenty-four hours. After 2, 4, and 7 days, cells in each well were trypsinized and counted using a Z1 Coulter Counter (Beckman Coulter).

**Confocal imaging and immunocytochemistry.** To determine targeting of mCerulean- and mVenus-tagged constructs we used live cell confocal imaging. Cells were washed with phosphate buffered saline and incubated with 100 nM MitoTracker Orange CMTMRos (M7510, ThermoFisher) or MitoView Green (70054, Biotium, Fremont, CA).

For immunocytochemistry, cells were grown on poly-L-lysine-coated glass coverslips. After phosphate buffered saline washes, cells were fixed in 10% neutral buffered formalin (VWR), permeabilized with 1% Triton X-100, and blocked with fetal goat serum. Primary antibodies were incubated at the indicated dilutions, and when needed secondary antibodies were goat anti-mouse Alexa Fluor 555 or goat anti-rabbit Alexa Fluor 488.

For *Drosophila* imaging, flies were embedded in Tissue-Tex O.C.T. Compound (Sakura), sliced to 7 µm thickness and mounted on glass coverslips. The sections were fixed using Formalin (Fisher Healthcare), permeabilized using 0.5% Triton X-100, and blocked using goat serum. Staining with primary antibodies occurred overnight in the presence of *Drosophila* larvae homogenate in 0.2% Triton X-100. Secondary antibodies were goat anti-mouse Alexa Fluor 555 or goat anti-rabbit Alexa Fluor 488.

Image acquisition was at room temperature. Imaging was performed on a Leica TCS SPE confocal microscope (Buffalo Grove, IL) using Leica Application Suite X v3.5 software. Acquisition was for 1024 × 1024 pixels at an 8-bit depth. Images have been enhanced for contrast uniformly, without alteration of gamma, and pseudocolored from grayscale.

**Affinity purification and Co-immunoprecipitation.** HEK293T cells stably expressing MCU-Flag were grown to confluency, while wild-type cells and cells stably expressing SDHB-Flag served as controls. After washing in phosphate-buffered saline, cells were lysed in 1% digitonin, 50 mM Tris, 150 mM NaCl and Halt protease and phosphatase inhibitors. Lysates were centrifuged at 4 °C for 10 min at 16000 g. 200–500 µg of cleared lysates were incubated with 10 µl of FLAG antibody conjugated beads overnight at 4 °C. For co-immunoprecipitation studies, after extensive washing, the beads were heated to 80 °C in 50 µl of sample loading buffer and then used for Western blotting as described above.

For proteomic analysis, cells were processed as above but eluted in 2% SDS in mass spectrometry-grade water. We analyzed six independent MCU-Flag samples and four control non-transduced samples.

**Liquid chromatography-tandem mass spectrometry.** Sample preparation for mass spectrometry was performed using established protocols[72,73]. Briefly, samples were diluted with 8 M urea buffer, reduced, alkylated and proteins were digested with trypsin overnight at 37 °C on a filter, then acidified using 1% formic acid. Peptides were analyzed with an Orbitrap Velos Pro mass spectrometer (Thermo-Fisher) interfaced with an EASY nLC-1000 UPLC outfitted with a PicoFrit reversed phase c18 column (15 cm × 75 µm inner diameter, 3 µm particle size, 120 Å pore diameter, New Objective). Spectra were acquired in a data-dependent mode with dynamic exclusion enabled. The top 20 MS1 peaks were fragmented using CID. Samples were run in duplicate.

The resulting spectra were analyzed using MaxQuant 1.6 against the UniprotKB human database. Database search engine parameters were as follows: trypsin digestion, two missed cleavages, precursor mass tolerance of 20 ppm, fragment mass tolerance of 0.5 Da, and dynamic acetyl (Protein N-term), and oxidation (M) modifications, The false discovery rate (FDR) was 1% and modified peptides had a minimum Andromeda score of 40. The proteins identified were further filtered to only include those identified in at least 50% of all FLAG pull downs as confident interactors. For relative quantification, peptide abundance was log2 transformed and missing values were imputed from a normal distribution with Perseus 1.6. Normalized log2 intensities of peptides were used in statistical comparisons of groups with Student's two sample *t* test. Volcano plots were generated with an FDR of 0.1. Statistical analyses were performed in Perseus 1.6. The mass spectrometry proteomics data have been deposited to the ProteomeXchange Consortium via the PRIDE partner repository with the dataset identifier PXD018742.

**Förster resonance energy transfer.** For flow cytometric measurements, data was collected at the Flow Cytometry Core Facility at the University of Utah. We followed established protocols to quantify FRET using flow cytometry[46]. Cells in 6-well dishes were transfected with 5 µg plasmid DNA using Lipofectamine 2000 (11668027, ThermoFisher) 1–3 days prior to analysis. On the day of cytometry, cells were incubated with 100 µM cycloheximide for ~2 h prior to analysis, to prevent artifacts from incomplete protein synthesis[46]. Flow cytometry was performed on a BD FACSCanto Analyzer. Signals were recorded using (1) 408 nm laser excitation and 450/50 nm emission (mCerulean), (2) 408 nm excitation and 525/50 emission (FRET), and (3) 488 nm excitation and 530/30 nm emission (mVenus). In all cases, we subset populations for live fractions of single cells using forward and side-scatter parameters. We obtained signals from untransfected cells for background correction. Background-corrected signals are designated $S_{Ven}$, $S_{Cer}$, and $S_{FRET}$ for the excitation/emission pairs 488/530, 408/450, and 408/525, respectively. Signals from mt-mVenus transfected alone were used to calculate the cross-talk ratio between $S_{Ven}$ and $S_{FRET}$ ($R_{A1}$), while signals from mt-mCerulean transfected alone were used to calculate the cross-talk ratios between $S_{Cer}$ and $S_{FRET}$ ($R_{D1}$) and between $S_{Cer}$ and $S_{Ven}$ ($R_{D2}$). These ratios were then used to correct for cross-talk signals within each analyzed cell co-transfected with MCU-mCerulean and either mt-mVenus or each of the mVenus-tagged NDUF constructs (NDUFA2, NDUFA5, NDUFA7, NDUFA10, NDUFS3, NDUFS6, NDUFB6, and NDUFV2). The correction calculations are:

$$\text{Cer}_{\text{direct}} = R_{D1} \times S_{Cer} \tag{1}$$

$$\text{Ven}_{\text{direct}} = R_{A1} \left( S_{Ven} - R_{D2} \times S_{Cer} \right) \tag{2}$$

$$\text{Ven}_{\text{FRET}} = S_{FRET} - \text{Cer}_{\text{direct}} - \text{Ven}_{\text{direct}} \tag{3}$$

where (3) is the FRET signal corrected for cross-talk from fluorescence due directly to mVenus and mCerulean, calculated using the prior ratios and Eqs. (1)–(2). Next, to calibrate the FRET signal, we make use of the mitochondrial-targeted mVenus-mCerulean dimers with varying linkers. Because of the one-to-one relationship between FRET donor (mCerulean) and acceptor (mVenus) in these constructs, the following equation holds true

$$\frac{\text{Ven}_{\text{FRET}}}{\text{Cer}_{\text{direct}}} = \frac{g_{\text{Cer}}}{g_{\text{Ven}}} \times \frac{\text{Ven}_{\text{direct}}}{\text{Cer}_{\text{direct}}} - \frac{f_{\text{Ven}}}{f_{\text{Cer}}} \tag{4}$$

where instrument- and fluorophore-related constants for excitation and emission are captured in the $g$ and $f$ terms, respectively. Because this is a linear equation, by plotting $\frac{\text{Ven}_{\text{FRET}}}{\text{Cer}_{\text{direct}}}$ against $\frac{\text{Ven}_{\text{direct}}}{\text{Cer}_{\text{direct}}}$ for the three mVenus-mCerulean pairs we can obtain the intercept $\frac{f_{\text{Ven}}}{f_{\text{Cer}}}$ from the best-fit regression line. Our value of 1.62 was in close agreement with the 1.65 value obtained in the prior publication, even with different instruments and mitochondria-targeted fluorophores. Finally, we calculated donor FRET efficiency for each cell by using the equation

$$\langle E \rangle_{\text{Cer}} = \frac{\text{Ven}_{\text{FRET}}}{\text{Ven}_{\text{FRET}} + \frac{f_{\text{Ven}}}{f_{\text{Cer}}} \times \text{Cer}_{\text{direct}}} \tag{5}$$

and plotting against the mVenus fluorescence of that cell. Analysis was carried out in FlowJo and OriginPro. Constructs showing no FRET have a flat distribution. Spurious concentration-dependent FRET is seen as a

linear increase in FRET efficiency with mVenus concentration. True FRET is visible as a rapid rise in efficiency to a plateau. For display, we fit a straight line to those constructs that demonstrated no FRET or concentration-dependent spurious FRET, or an exponential of the form $y = a(1 - e^{-bx})$ for the constructs that demonstrated strong FRET. For the summary graph in Fig. 3b, we used the FRET efficiencies in the fluorescence window of 150,000–200,000 for mVenus expression.

For acceptor photobleaching assays, imaging was carried out via confocal imaging (see section above). Briefly, cells transfected with the respective constructs were grown on poly-lysine coated coverslips. Cells were fixed for 10–15 min in 4% paraformaldehyde in phosphate-buffered saline, then imaged in phosphate-buffered saline. Fluorescence for mVenus was imaged at 488 nm excitation and $550 \pm 30$ nm emission and fluorescence for mCerulean was imaged at 405 nm excitation and $475 \pm 25$ nm emission. Acquisition settings were kept identical for the entire dataset. Typically, 1–5 cells were imaged at high magnification. The photobleaching step used 80–85% laser power for ~10 s. A correction factor for mCerulean photobleaching was determined by transfecting cells with mito-mCerulean alone and measuring its intensity before and after photobleaching. For analysis, pre- and post-photobleaching images were aligned by using the ImageJ Rigid Registration tool with default parameters, as there was sometimes minor motion or change in mitochondrial shape during photobleaching. Regions of interest were picked from identical locations within each cell in the mCerulean channel in pre- and post-photobleaching images. After applying the correction for photobleaching, these were turned into effective FRET efficiency ($E_{Eff}$) by

$$E_{Eff} = 1 - \frac{Cer_{pre,corr}}{Cer_{post,corr}} \tag{6}$$

where $\frac{Cer_{pre,corr}}{Cer_{post,corr}}$ is the corrected ratio of mCerulean fluorescence pre- and post-photobleaching[74].

**Duolink proximity ligation assay**. Duolink proximity ligation (DUO92102, Sigma) assays were performed per the manufacturer's protocol. We used mono-clonal mouse anti-NDUFS2, and monoclonal rabbit anti-MCU, both of which have been knockout validated. Mitochondrial counterstain was with anti-COX IV-Alexa Fluor 488. Cells were imaged on a confocal microscope as described above. Areas for imaging were selected based on uniform distribution of DAPI-stained nuclei with about 20–50 cells per field. Separate images were taken of each field for DAPI (nuclei, 405 nm laser), COX IV-Alexa Fluor 488 (mitochondria, 488 nm laser), and Duolink spots (561 laser). Analysis was performed on CellProfiler (v3.1)[75]. After median filtering DAPI-stained nuclei, the ExpandObjects module was used to dilate these until touching adjacent objects to define cell borders. Mitochondrial objects were identified in median-filtered COX IV stained images, and Duolink spots were identified in the Duolink images. To detect Duolink spots corresponding to mitochondria the RelateObjects module was used, and any non-mitochondrial Duolink spots filtered out. Spots were assigned to the cell object they were in to generate an estimate of spots per cell. For display purposes only, the size and contrast of the Duolink spots in the manuscript figures have been increased uniformly to allow visualization with the small figure size.

**Doxycycline repression for determining MCU stability**. Stable lines expressing pCW57.1-MCU-Flag (derived from pCW57.1-MAT2A[48]) allowed repression of MCU transcription by doxycycline. The cells were grown using media supplemented with 3% fetal bovine serum, to minimize cell division. Day 0 cells were collected immediately prior to doxycycline addition, and the remaining cells were incubated with 1 μg/mL doxycycline for the indicated number of days. For Complex I inhibition, cells were incubated in 1 μM rotenone for 48 h prior to doxycycline addition.

**Rapamycin-induced dimerization**. Stable cell lines were created using lentiviral transduction, expressing (1) FRB-MCU-Flag and mito-FKBP-HA, or (2) FRB-MCU-Flag and NDUFA10-FKBP-HA. Co-immunoprecipitation was performed as described above, except lysates were incubated with either DMSO or 100 nM rapamycin during affinity purification overnight at 4 °C. For stability studies, cultured cells were incubated with DMSO or 100 nM rapamycin for one day prior to adding 1 μg/mL doxycycline, and cells were collected and processed for Western blotting at the indicated timepoints as described above.

**Oxygen consumption**. Oxygen consumption was measured using an Oxytherm +R system (Hansatech instruments, Norfolk, UK)[12]. The system was calibrated daily with air saturated deionized water (high $O_2$) and glucose oxidase + glucose. $10^6$ cells were washed in PBS and permeabilized in solution containing (in mM) 125 KCl, 10 $K_2HPO_4$, 1 $MgCl_2$, 0.01 EGTA, 20 HEPES, pH 7.2 supplemented with 50 μg/mL digitonin, for 15 min on ice. Oxygen consumption was monitored after the serial addition of 5 mM L-glutamic acid + 5 mM L-malic acid, 0.5 mM ADP, and 5 mM succinic acid + 1 μM rotenone.

**Mitochondrial $Ca^{2+}$ uptake**. Imaging was performed in 96-well plates on a Cytation 5 microplate reader (Biotek, Winooski, VT). For mice, 100 μg of cardiac

mitochondria were used per trial. For HEK293T, $10^5$ cells were permeabilized with 0.005% digitonin and used per trial. Samples were incubated in 100 μL of solution containing (in mM): 125 KCL, 20 HEPES, 5 $K_2HPO_4$, 1 $MgCl_2$, 5 succinic acid, 0.01 EGTA, 0.1% BSA, and 1 μM Oregon Green BAPTA-6F (O23990, Thermo-Fisher). pH was adjusted to 7.3 with KOH, and osmolality to 290–300 mOsm/L). Excitation and emission wavelengths were 485/510 nm. 10 μM $CaCl_2$ was injected per trial. For rate calculations, traces were transformed so that baseline corresponded to 0, and peak fluorescence after $Ca^{2+}$ pulse was 1, to allow comparison of experiments performed on different days. The uptake rate was calculated for the first 30 s after the pulse, and values were normalized to the average of the controls.

**Histology and nuclei counting**. Extracted hearts were incubated in a fixative solution containing 4% paraformaldehyde in PBS for 48 h at 4 °C, and then placed in a 70% ethanol solution. The samples were embedded in paraffin, cut, and stained with Masson's trichrome by the Research Histology core at the Huntsman Cancer Institute (University of Utah). Slides were imaged using a BX51WI microscope (Olympus, Center Valley, PA).

Nuclei density was calculated via automated analysis performed on CellProfiler (v3.1)[75]. The UnmixColors module was used to separate each image into two, one that highlighted nuclei (setting: hematoxylin), and one for cellular staining (setting: eosin). The eosin image was analyzed to determine how much of the image was occupied by cells, to correct for tissue processing and any blank spaces. The hematoxylin image was analyzed to count nuclei. Nuclei density was calculated by dividing the number of nuclei by the area of each image covered by cells. For the eosin image, we used IdentifyPrimaryObjects, MeasureImageAreaOccupied, and the image scale to calculate the area occupied by cells. For the hematoxylin image, to smooth over heterogeneity of staining within each nuclei, we sequentially applied the GaussianFilter, Threshold, and Opening modules. Subsequently, the IdentifyPrimaryObjects module was used to count nuclei. All images were processed identically.

**Echocardiography**. Echocardiography was performed at the University of Utah Small Animal Ultrasound Core[12]. Mice were sedated with inhaled isoflurane, restrained in the supine position, and cleared of chest fur. M-mode images were recorded in short-axis at the level of the papillary muscles, using a Vevo 2100 ultrasound machine equipped with a 55-MHz probe (Visual Sonics, Toronto, Ontario, Canada).

**Island assay**. Flight capabilities were determined using a modified island assay[76]. We used a rectangular ice pan that held a flypad in the middle. The bottom of the container was filled with cold soap water that reached the top of the flypad without covering it. Clear plastic wrap was placed over half of the container in order to prevent flies from escaping. Female flies were collected within 48 h of eclosion, and placed on normal fly food in groups of 12. Two days later, flies were tapped to the bottom of the vial, the vial was immediately inverted and flies were tapped onto the flypad in the island. Vials were inspected for any flies remaining after inversion to determine the starting number of flies on the island. Videos were taken of each group, and were manually scored to determine how many flies remained on the flypad every 2 s until the 20 s mark was reached.

***Drosophila* viability assay**. To determine viability, separate groups of male and female flies were counted from at least 3 crosses. Viability was calculated by dividing the number of flies without balancers (experimental group) by the number of flies with balancers (control group) and multiplying by 100 to obtain a percentage.

**Cartoons**. Cartoon drawings were based on PDB structures as follows: Complex I, 5LNK[77] and 5LC5[78]; respirasome, 5J4Z[79]; MCU, 6DNF[80]; FRB-Rapamycin-FKBP, 1FAP[49]; Lon protease, 4YPL[81].

**Statistics**. For flow cytometric assays and confocal imaging, N refers to individual cells. For electrophysiological assays, N refers to individual mitoplasts. For Duolink imaging, N refers to cells calculated from nuclei staining. For HEK293T cell culture qPCR, N refers to technical replicates. For western blots, N refers to independent replicates or individual animals. For $Ca^{2+}$ uptake imaging, N refers to individual replicates. For animal studies, N refers to individual animals. We considered $p < 0.05$ statistically significant. Analysis was performed in OriginPro 9, GraphPad Prism 8, Microsoft Excel 2016, or R v3.5. Individual tests are described in the figure legends.

**Reporting summary**. Further information on research design is available in the Nature Research Reporting Summary linked to this article.

## Data availability

The mass spectrometry proteomics data have been deposited to the ProteomeXchange Consortium via the PRIDE partner repository with the dataset identifier "PXD018742".

All other relevant data supporting the key findings of this study are available within the article and its Supplementary Information files or from the corresponding author upon reasonable request. Source data are provided with this paper.

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

## Acknowledgements

We thank Edward Owusu-Ansah, Alex Whitworth, Ronald Davis, and the Bloomington *Drosophila* Stock Center at Indiana University (supported by NIH P40OD018537) for fly stocks, Vamsi Mootha for multiple plasmids and MCU knockout cell lines, Michael Ryan for the NDUFB10$^{KO}$, NDUFS4$^{KO}$, and control cell lines, David Clapham for other HEK293T cell lines, John Elrod, Ronald Kahn, and Nils-Göran Larsson for mouse lines, James Marvin and Flow Cytometry Core staff at the University of Utah (supported by NIH P30CA042014), Small Animal Ultrasound Core staff at the University of Utah, and Derek Warner and DNA Sequencing Core staff at the University of Utah. Support is from the National Institutes of Health (DK110358 [JMP, ARR], UL1TR002538 [C.T.M.], HL124070 [D.C.], HL141353 [D.C.], HL007576 [S.S.]), the Nora Eccles Treadwell Foundation (D.C., M.T.F., S.G.D., S.F.), the Gilead Sciences Research Scholars Program (D.C.), the American Heart Association Postdoctoral Fellowship Award (834544 [D.R.E.]), and the University of Utah Driving Out Diabetes, a Larry H. Miller Wellness Initiative (D.C.). The content is solely the responsibility of the authors and does not necessarily represent the official views of the National Institutes of Health.

## Author contributions

Conception: D.C., E.B., D.R.E., S.S. Design: D.C., E.B., S.S., J.M.P., D.R.E., M.W.S., C.T.M., S.F., A.R.R. Acquisition: E.B., S.S., J.M.P., D.R.E., A.M.B., X.Y., M.W.S., M.C.P., A.K.H., K.F., C.T.M., S.C., A.B., R.D.B., D.C. Analysis and interpretation: all authors. E.B. and D.C. wrote the manuscript in consultation with all authors.

## Competing interests

The authors declare no competing interests.

## Additional information



