## [Peer Review File · Nature Communications]

Mitochondrial calcium uniporter stabilization preserves energetic homeostasis during Complex I impairmentREVIEWER COMMENTS

Reviewer #1 (Remarks to the Author):

Thankyou for asking me to comment on this intriguing paper. At the heart of the paper is the observation that the current carried through the mitochondrial calcium uniporter is increased following – but not in direct response to – inhibition of complex I of the respiratory chain. The rest of the paper is an exploration of the mechanism by which this happens and an attempt to give the phenomenon a 'story' as a mechanism that compensates for the mitochondrial deficiency cause by the complex I defect.

There are many very nice things in this paper – the phenomenon is described in multiple cell models and across species, and appears to involve the regulated processing of the MCU protein rather than a change in gene expression. The paper is quite complex and dense and there are some aspects where clarity could certainly be improved. My biggest problem is the way the phenomenon seems to have been shoehorned into a bigger story about compensation of bioenergetic capacity. If the authors simply presented this as the regulation of the MCU protein, it would be straightforward. The attempt to present this as a compensation for the energetic defect caused by a complex I deficiency raises a number of questions which muddy the story. This is also complicated by the fact that in essence the group have already published this conceptually – the basic principles of this concept are there in the Sommakia paper (ref 12), and the advance here is the generalisation to other models and the mechanism.

My problem is that the bioenergetic compensation is not demonstrated at all – one expects to see a rise in intramitochondrial calcium concentration, increased activity of the TCA cycle, perhaps decreased phosphorylation of PDH, increased ATP generation or capacity, an increase in calcium stimulated respiratory rate, perhaps an increase in mitochondrial membrane potential driven by increased calcium – none of these things have been explored, and yet this would surely present the basis for bioenergetic compensation? I'd expect to see at least some of these addressed?

The first sentence of the discussion illustrates clearly the way I feel the story has been overplayed: 'In this report, we identify an essential functional, biochemical, and genetic interaction between Complex I and the uniporter, necessary for survival when Complex I becomes impaired'. It isn't an essential function and neither is it 'necessary for survival' – the mice die at 10 weeks even with the MCU expression, so they don't survive – they just survive a bit longer!

As the paper stands, I have a number of comments which I'll present in the order they appear in the paper.

Figure 1:

- i) The MCU blots need to be quantified.
- ii) For each graph, it looks like you've chosen the smallest control current and the biggest increased current. As the range is very big, it would be better to show the range and errors in a composite plot to demonstrate the range.
- iii) You 'obtained fibroblasts from an infant with fatal lactic acidosis and cardiomyopathy due to NDUF10 deficiency (NDUF10-/C107S)' – mostly out of curiosity – was this a homoplasmic mutation?
- iv) On Page 4: you state: 'MCU levels increased in these different lines despite the absence of a corresponding mRNA upregulation (MCU expression versus control: 0.95, NDUF10KO; 0.68, FOXRED1KO; 0.88, NDUF10-/C107S, n = 2)'. I assume that the numbers refer to MCU mRNA levels? The protein levels should also be quantified.
- v) You use the genetically-encoded SoNar sensor in Fig. 2b, c: – these data are really unconconvincing, and it is very hard to imagine that these are statistically significantly different! It is also not clear what the fluorescence ratio means – you need to explain how the ratio relates to NADH/NAD.

- vi) The significance and mechanisms of these data are puzzling and unclear: Increased NADH is consistent with inhibition of complex I. You then argue (I think) that this increase in NADH is also driven by increased calcium uptake – why not simply ask what happens to NADH:NAD in the absence of calcium or in MCU ko?
- vii) There is an apparent complexity in this story that needs to be resolved: the experiments with the mito-LbNOX reduces the NADH:NAD ratio and this prevents the increase in iMCU, suggesting that the increase in iMCU is driven partly by NADH driven ROS generation. However, the (proposed) increase in mitochondrial matrix calcium should be driving the increase in NADH? Doesn't this constitute the basis for a positive feedback cycle here? NADH accumulates with complex I deficiency and drives ROS generation, This leads to upregulation of MCU which should increase mitochondrial calcium which should drive increased NADH and that should further increase ROS generation?
- viii) the legend could usefully give more information?
- ix) I really struggled with figure 3: in a, the WB ought to be quantified. In Fig 3b what are A5/A7 etc? how do these relate to the different proteins? It is very hard for a reader to interpret this figure! I don't understand what it is we're looking at.
- x) Similarly, the duolink data are startlingly unimpressive and hard to read. All I can see is a few green spots.
- xi) I struggle to understand why the authors make so much of a close interaction between complex I and MCU given that the expressed mitominiSOG – which is not localised had the same effect, and the SOD overexpression, which is also not localized, quenched the effect. These imply that localization is not really critical? This begs the question of whether other causes of mitochondrial oxidative stress also increase MCU levels?
- xii) On Page 7 you say 'Finally, Complex I impairment decouples MCU..' – I'm not sure what this means? What are you referring to? Decouples from what?
- xiii) 'Because TFAM deletion alters the driving force for Ca²⁺ uptake, we quantified the steadystate ability of mitochondria to take up Ca²⁺ rather than uptake rates (Fig. 5d).' why not energise with succinate and this way you might avoid problems with potential in your models?
- xiv) Fig 5 – quantify westerns. 5B doesn't show MCU levels in the TFAM ko? Have I misunderstood something?
- xv) The calcium uptake data in Fig 5C seem very odd. I initially asked how does mito calcium uptake happen in the TFAM-MCU double ko, but the blots show substantial MCU expression in the 'TFAM-MCU double knockout' which is confusing?
- xvi) Why are there no measurements of mitochondrial calcium uptake to match the current measurements – ie to show that with rotenone treatment mitochondrial calcium uptake is increased? Surely this lies at the heart of the whole story? You imply that calcium uptake might be decreased because of a decreased potential which clearly gets complicated but you can get around that by energizing mitochondrial with succinate to bypass the complex I block.
- xvii) 'In summary, in a disease model of cardiac ETC dysfunction, activity of the uniporter was essential for survival'. This seems an odd statement given that the mice are all dead at 10 weeks?!

Reviewer #2 (Remarks to the Author):

Cardiac depletion of Tfam has been used as a model for cardiomyopathy induced by mitochondrial OXPHOS deficiencies and was previously shown to be accompanied by accumulation of Ca²⁺ in mitochondria. Here, the authors pick up on this observation and reveal an intriguing functional link between complex I activity and mitochondrial Ca²⁺ influx via MCU, which appears to be conserved from Drosophila to mouse and human cells. They demonstrate that MCU binds complex I but dissociates from it upon complex I inhibition. Whereas MCU bound to complex I is unstable and rapidly degraded, dissociation from non-functional complex I results in stabilization of MCU and increased mitochondrial Ca²⁺ uptake.

This is an interesting study which is likely of relevance for complex I diseases. It reveals an unexpected mechanism regulating the turnover of MCU via the activity of complex I. Whereas assembly into protein complexes is generally observed to stabilize proteins, here the authors suggest the opposite and find that increased MCU turnover correlates with binding to complex I. While the

functional link between complex I inhibition and MCU stability is well documented, some questions remain whether or not this is indeed regulated via assembly. Although the authors use various approaches to assess MCU-complex I interaction and use an elegant approach to show that tethering of MCU to a complex I subunit induces degradation, I am still wondering about quantitative aspects of complex formation. Which fraction of MCU is bound to complex I and is this the fraction which is degraded? Many experiments are done after over-expression further hampering a quantitative assessment. It is shown that disturbances in the NADH/NAD⁺ ratio or increased ROS can induce MCU stabilization but how MCU binding to complex I is regulated remains speculative. Another concern is the conclusion of the authors that increasing MCU stability affects survival of heart specific Tfam-knockout mice, as the authors do not compare the TFAM-MCU DKO with heart-specific MCU knockout (Fig. 5). In this context, I find the title somehow misleading as MCU does not compensate for the loss of complex I activity but rather is likely part of metabolic rewiring allowing cells to better cope with complex I deficiencies.

Specific points:

1. What is the efficiency of the immunoprecipitation of MCU and complex I in Fig. 3a and for endogenous proteins? Exposures of different immunoblots cannot be compared. Previous studies on MCU interactors did not detect complex I.
2. Does MCU bind to fully assembled complex I? Only three complex I subunits were found to precipitate with MCU raising the possibility that a complex I intermediate rather than assembled complex I binds to MCU. Moreover, the FRET analysis identifies different complex I subunits than detected in the immunoprecipitate. Therefore, conclusions on the binding interface appear not appropriate.
3. In Fig. 4a, the authors use DOX to inhibit transcription of (overexpressed) MCU to assess MCU turnover. Cell proliferation should be monitored, which could contribute to the decline in MCU steady state levels. Moreover, it remains unclear why the authors did not use CHX to examine the stability of endogenous MCU as they did for (endogenous) ROMO1 in Figure S4A. What is the stability of other subunits of the MCU complex?
4. In general, the description of experiments in the figure legends should be improved.

Reviewer #3 (Remarks to the Author):

This paper by Chaudhuri and co-workers focuses on the activity regulation of the mitochondrial calcium uniporter (MCU) channel. Evidence is provided that complex I (CI) impairment stimulates MCU activity. It is proposed that this occurs via CI-mediated MCU degradation mediated by physical contacts between CI and the MCU pore-forming subunit (occurring under normal conditions), and that this MCU degradation mechanism is inhibited by CI dysfunction. Overall, the manuscript is nicely presented but not always easy to follow (i.e. it is not sufficiently explained how or why a certain experiment is performed). This likely has also to do with the large amount of data presented. In this context, I have several points that require clarification.

MAJOR POINTS

1. General remark: Regarding the use of chronic rotenone treatment in cell models. Although its primary effect might be limited to CI inhibition, cell metabolism and redox homeostasis also change (over time) as a consequence of this treatment. This makes the "rotenone-model" difficult to interpret in a CI-specific manner. Although less studied, this also holds true for the chronic use of other ETC inhibitors. To limit the potential impact of these effects, it is advisable to use the lowest inhibitor concentration that fully inhibits mitochondrial oxygen consumption in the cell model of choice.
2. Results section: How was the conclusion reached that "disruption" of the NDUF54 subunit induces a milder CI deficiency? It is referred to Fig. S1F, but this shows currents?
3. Results section: For making statements regarding the degree of CI deficiency it is necessary to obtain information on CI enzymatic activities. This also holds true for the other inhibitors and ETC

complexes that are studied. For instance, are the studies ETC activities fully or partially inhibited in the used inhibitor-treated and other models?

4.Results section: It is puzzling that matrix free calcium levels were higher in CI deficient cells (Fig. S1I). Why would there be mitochondrial calcium uptake? Does this suggest that calcium release from the ER is activated in CI deficient cells? Why? How does the magnitude of the sensor ratio signal (Fig. S1I) relate to the “real” matrix calcium concentration. Is the K_m value for calcium of this sensor known?

5.Results section: Please provide experimental evidence demonstrating that deltaPSI is depolarized under CI-deficient conditions in the used (cell) models. This cannot be assumed since deltaPSI depends on many other factors including CV and ANT activity, as well as metabolic state. This is evidenced by previous studies in the literature.

6.Results section: In case of Fig 1C (and for proper interpretation of many of the other results), it is crucial to demonstrate whether the levels of the CI holo-complex have changed. This can be done with BN-PAGE experiments (using VDAC1 or CII as a control). Previous studies suggest that chronic rotenone treatment increases the levels of this complex, which would suggest that increased CI levels correlate with increased MCU levels. This could mean that CI-MCU interaction plays a stabilizing role. This possibility needs to be ruled out.

7.Results section: I find it difficult to understand how dysfunctional CI would induce its own ROS-mediated degradation. In such a mechanism, there would never be any CI activity detectable in cells carrying CI mutations (i.e. there would be no CI holo-complex; see my remark above). I guess it is overlooked here that cells adapt to increased ROS levels by activation of ROS detox systems, as well as metabolic adaptation mechanisms, to prevent (full) CI loss and energy dysbalance.

8.Results section: MitoSOX analysis does not measure ROS production but ROS levels. Moreover, MitoSOX can be oxidized during its travel to mitochondria (it reacts at very high rates with ROS), its fluorescence analysis is not specific to superoxide, and it accumulates in mitochondria in a deltaPSI dependent manner. Therefore MitoSOX analysis is not ROS- or mitochondria-specific.

9.Results section: I have difficulty understanding the results with LbNOX. How does the fact that rotenone increases mitochondrial NADH levels links to the data in figure 1?

10.Results section: Singlet oxygen is not superoxide. How does this work in the mt-miniSOG cell model? Does MitoSOX detect singlet oxygen?

11.Results section: I have trouble understanding why first rotenone is used and then piericidin A? Long-term cell treatment with rotenone can induce off-target effects which are often not observed for pieridicin A. How was the concentration of piericidin A determined? The latter question can also be asked for all other inhibitors (see also my first comment).

12.Results section: As stated above, in various (cell) models the CI-deficient state is associated with increased NADH and reduced NAD⁺ levels. So why would “uniporter boosting of NADH/NAD⁺ ratio” be required for survival during CI dysfunction?

13.Results section: Does the length of the linker between the fluorescent protein (FPs) and CI subunit affect the FRET signal? I.e. is it expected to increase the length scale across which FRET will occur? To demonstrate interaction also acceptor-bleaching analysis needs to be performed.

14.Results section: Importantly, not all CI subunits can be tagged with an FP without disturbing their incorporation in the CI holocomplex. This can be investigated using BN-PAGE and/or fluo- and Western-blot analyses. Are the tagged subunits also close to other ETC complexes in the supercomplex(es)? This can be deduced from the available structures. If so, is it expected that they still can interact with MCU?

15.Results section: Please demonstrate that doxycycline treatment does not inhibit mitochondrial parameters (e.g. oxygen consumption. Such inhibition has been observed in various systems and might affect the signalling involved in CI-MCU interaction.

16.Results section: Why is it more likely that ROS generated by mito-miniSOG will non-specifically damage CI? What is the range of action of ROS (singlet oxygen?) generated by mini-SOG?

MINOR POINTS

1.Results section: What is exactly meant with “typical Ca²⁺ fluorescence assays”?

2.Results section: Can the authors rule out that the composition of their pipette medium affects their results in whole-mitoplast voltage-clamp recordings?

3.Results section: how does pyruvate-induced NAD⁺ regeneration exactly work?

4.Results section: What is meant by MCU acting as a “ROS buffer”? Is this an assumption or a conclusion?

We thank the reviewers for their comments and suggestions. We are glad they have found our manuscript to be impactful and of interest. We have reworked the manuscript and added substantial new data, including more than 20 new experiments, to bolster our conclusions.

GENERAL

- We have renumbered all points raised by the reviewers for ease of reference (R1-R50).
- Where changes to the manuscript have been made, these are numbered and **highlighted in red**, with line numbers in the main text referenced.
- Several reviewers commented on the density of the manuscript with a large number of experiments performed. Moreover, many individual comments requested additional experiments or clarification. Because of the 5,000 word limit for the main manuscript and the additional experiments/clarifications requested by the reviewers, our changes to the main manuscript text have by necessity been quite brief. We have expanded the supplementary information to add a lot of the additional materials requested.
- It was difficult to add all the new data figures into the supplementary information within the prior grouping, so we have broken up the Supplementary Figures into 44 separately referenced items.

Reviewer #1 (Remarks to the Author):

Thank you for asking me to comment on this intriguing paper. At the heart of the paper is the observation that the current carried through the mitochondrial calcium uniporter is increased following – but not in direct response to – inhibition of complex I of the respiratory chain. The rest of the paper is an exploration of the mechanism by which this happens and an attempt to give the phenomenon a ‘story’ as a mechanism that compensates for the mitochondrial deficiency caused by the complex I defect.

There are many very nice things in this paper – the phenomenon is described in multiple cell models and across species, and appears to involve the regulated processing of the MCU protein rather than a change in gene expression. The paper is quite complex and dense and there are some aspects where clarity could certainly be improved. My biggest problem is the way the phenomenon seems to have been shoehorned into a bigger story about compensation of bioenergetic capacity. If the authors simply presented this as the regulation of the MCU protein, it would be straightforward. The attempt to present this as a compensation for the energetic defect caused by a complex I deficiency raises a number of questions which muddy the story. This is also complicated by the fact that in essence the group have already published this conceptually – the basic principles of this concept are there in the Somrakia paper (ref 12), and the advance here is the generalisation to other models and the mechanism.

R1. My problem is that the bioenergetic compensation is not demonstrated at all – one expects to see a rise in intramitochondrial calcium concentration, increased activity of the TCA cycle, perhaps decreased phosphorylation of PDH, increased ATP generation or capacity, an increase in calcium stimulated respiratory rate, perhaps an increase in mitochondrial membrane potential driven by increased calcium –

none of these things have been explored, and yet this would surely present the basis for bioenergetic compensation? I'd expect to see at least some of these addressed?

We apologize for the lack of clarity on this issue. We had previously described the phenomenon of an increase in mitochondrial calcium noted during electron transport chain (ETC) dysfunction in *Tfam* KO mice¹. In that manuscript we had already performed several of the functional measurements listed by the reviewer, and had demonstrated Ca²⁺-induced increases in respiration and ATP generation (Fig. 6I, 6J, S1H, S2B in PMC8319341).

It was the finding of this phenotype, the increase in calcium levels and consequent bioenergetic compensation, that prompted the search for a mechanism and an analysis of organism-level consequences that forms the core of this manuscript. Thus, we had already established the ability of increased Ca²⁺ influx to maintain mitochondrial homeostasis during cardiac ETC dysfunction, though the mechanism driving calcium increase and whether this affected survival were unknown. We initially felt that examining cultured cell lines for these more indirect bioenergetic parameters (respiration, membrane potential, etc.) would be of lesser value, compared to the prior data collected in heart tissue. Nevertheless, we see that a reader may have the same questions as posed by the reviewer. Therefore, we attempted to examine the effect of Ca²⁺ on respiration and ATP synthesis. However, we were stymied in these efforts by rapid deterioration of Complex I deficient (whether via rotenone or genetic manipulation) samples after ADP addition. Essentially, after adding ADP to stimulate state 3 respiration, we would see a brief respiratory burst followed by rapid cessation of respiration/ATP synthesis. This effect persisted whether using Complex I deficient permeabilized cells or isolated mitochondria; whether measuring respiration (Oroboros or Oxytherm equipment) or ATP synthesis; or whether using different substrates/buffers/additives noted in the literature. It is unclear why ADP addition causes such rapid deterioration in these permeabilized/isolated mitochondria, and, notably, prior use of these cultured Complex I deficient lines have largely used these for assessment of Complex I assembly, not for respiratory measurements²⁻⁵. Since we couldn't successfully measure the effects of Ca²⁺ on ADP-stimulated function, as we had for cardiac mitochondria in our prior publication, we proceeded with alternative strategy of measuring PDH phosphorylation, as suggested by the reviewer:

1. We analyzed PDH phosphorylation in Fig. S12. As expected, increased mitochondrial Ca²⁺ led to diminished PDH phosphorylation in rotenone-treated cells, NDUFB10 KO cells, and FOXRED1 KO cells, compared to controls. We have added the following text (Line 124): "One downstream effect of mitochondrial Ca²⁺ is to stimulate pyruvate dehydrogenase phosphatases that reduce PDH phosphorylation and increase its activity. Consistent with enhanced Ca²⁺ uptake, we found reduced PDH phosphorylation in Complex I inhibited cells (Fig. S12)."
2. We have clarified in the introduction that our prior data showed that Ca²⁺ helped preserve bioenergetic homeostasis in the *Tfam* deficient heart (line 58): "These

increases in mitochondrial Ca²⁺ help rescue respiration and ATP synthesis to near wild-type levels.”

R2. The first sentence of the discussion illustrates clearly the way I feel the story has been overlaid: ‘In this report, we identify an essential functional, biochemical, and genetic interaction between Complex I and the uniporter, necessary for survival when Complex I becomes impaired’. It isn’t an essential function and neither is it ‘necessary for survival’ – the mice die at 10 weeks even with the MCU expression, so they don’t survive – they just survive a bit longer!

3. As requested by the reviewer, we have removed “essential” from the initial sentence of the discussion, and changed “necessary for survival” to “that helps maintain bioenergetic homeostasis”. (line 430)

As the paper stands, I have a number of comments which I’ll present in the order they appear in the paper.

R3. Figure 1: The MCU blots need to be quantified.

4. We have quantified the blots. These are included in Fig. S5. They show increases in MCU and EMRE in both pharmacological and genetic inhibition of Complex I.

R4. For each graph, it looks like you’ve chosen the smallest control current and the biggest increased current. As the range is very big, it would be better to show the range and errors in a composite plot to demonstrate the range.

In showing exemplars, we typically display traces near the average for each condition, not the smallest control current and biggest increased current, as suggested by the reviewer. We apologize for the confusion, as this impression about exemplar selection may be due to incomplete labeling of the figures on our part. In the original submission, we had not specified that each point in the summary graph represented the current density at -140 mV, and this may have led to the confusion. We have now relabeled graphs as documented below. In addition, to show the range, all data points and error values are shown in the summary graphs for each assay. This method of presentation is the standard for ion channel electrophysiology data, and is how traces have been presented for MCU by all groups, including the Anderson (Fig. S8, PMC3471377), Clapham (Fig. 1, PMC3818247), Elrod (Fig. 2, PMC4517182), Foskett (Fig. 1, PMC5384258), Kirichok (Fig. 1, PMC3818247), Madesh (Fig. 2, PMC5357178), Mootha (Fig. 3, PMC4091629), and Perocchi (Fig. 4, PMC5825229) groups. In Fig R1, we have indicated which value in the summary graph each exemplar represents (red circle with black outline) for data in the main figures. In the original submission, for the NDUFB10 KO and NDUFS4 KO graphs, the exemplars were further from the mean, and now these have been replaced in the current submission with exemplars closer to mean values (Figs 1E and S8). For Fig. 2E, where no difference is seen between SOD2-expressing cells with or without rotenone, we used exemplars further from the mean as these were the exemplars that were most comparable in shape. For *Drosophila* flight muscle uniporter currents in Fig. S38 of this submission, because the currents were tiny, we used exemplars further from the mean to clearly show the

presence of uniporter current (a finding independently confirmed by the Kirichok group, see Fig. 7 in ref ⁶).

We have made the following modifications:

5. In the original submission, we had not specified that each point in the summary graph represented the current density at -140 mV, and have updated the y-axis labels of all panels with electrophysiology data (e.g. see Fig. 1). We have also updated the Fig. 1 legend as follows (Line 475): “Each point in the summary graphs in panels 1b, 1d-h represents the current density measured at -140 mV for an individual mitoplast (see arrowheads in Fig. 1a).”

Figure R1: Each panel shows which data point (red circle) corresponds to each exemplar. The exemplars are traces close to the mean value of the data for that condition.

R5. You ‘obtained fibroblasts from an infant with fatal lactic acidosis and cardiomyopathy due to NDUFB10 deficiency (NDUFB10-/C107S’ – mostly out of curiosity – was this a homoplasmic mutation?

NDUFB10 is a nuclear gene, and the mutation is compound heterozygote with one allele encoding a paternally-inherited nonsense mutation (c.206_207insT, pE70*) and a maternally inherited missense mutation (c.319T>C, p.C107S)⁵. No viable protein is made from the nonsense mutation, and the missense mutation impairs mitochondrial import of NDUFB10. We could not detect NDUFB10 in the mutant cells (Fig. S9), so it appears all mitochondria in this line are deficient in NDUFB10.

R6. On Page 4: you state: ‘MCU levels increased in these different lines despite the absence of a corresponding mRNA upregulation (MCU expression versus control: 0.95, NDUFB10KO; 0.68, FOXRED1KO; 0.88, NDUFB10-/C107S, n = 2)’. I assume that the numbers refer to MCU mRNA levels? The protein levels should also be quantified.

Yes, these refer to mRNA. We have made the following changes:

6. Line 140: Modified the text to read: “(MCU mRNA expression versus control ...)”
7. Western blots have been quantified in Fig. S5.

R7. You use the genetically-encoded SoNar sensor in Fig. 2b, c: – these data are really unconconvincing, and it is very hard to imagine that these are statistically significantly different! It is also not clear what the fluorescence ratio means – you need to explain how the ratio relates to NADH/NAD.

8. We apologize for the lack of clarity. An increasing F405/F488 value corresponds to larger NADH/NAD⁺ ratio. We have amended the legend to Fig. 2B as follows: “Violin plots of MitoSoNar fluorescence ratio (B, increased F405/F488 value corresponds to increased NADH:NAD⁺ ratio).” (line 483)
9. The differences between the groups are actually very highly statistically significant ($p < 10^{-10}$). There is a ~20% increase in mean NADH:NAD⁺ ratio for FOXRED1 and ~60% increase with NDUFB10 compared to controls, and we have updated the Fig. 2B legend with these values (line 486). These are similar to changes seen with SoNar in other publications (Fig. 1C, PMID 27362337; Fig. 2B, PMC7393259; Fig. 2D, PMC7320160; Fig. 3A, PMC4850741; Fig. 3A, PMC3253140), and similar to the 10-20% increase seen in rotenone-treated cells using an older generation mito-targeted biosensor, mito-Frex (PMC3253140). Note that the exact value of the ratio depends on the imaging method and settings, but relative changes are similar to prior publications. Thus, we feel our data is consistent with similar assays done by other groups. We believe our approach measuring SoNar fluorescence via flow cytometry is quite rigorous, as it reveals the underlying distribution of NADH/NAD⁺ ratios across an entire cellular population. In addition, before using SoNar, we had tried several commercially-available NADH/NAD⁺ quantification kits. We found these to be very unreliable on HEK293T mitochondria, as ratios varied substantially from one replicate to another despite care to process samples identically and rapidly. As mito-SoNar measures mitochondrial NADH:NAD⁺ *in situ* in live cells, this approach seems far more reproducible.

R8. The significance and mechanisms of these data are puzzling and unclear: Increased NADH is consistent with inhibition of complex I. You then argue (I think) that this increase in NADH is also driven by increased calcium uptake – why not simply ask what happens to NADH:NAD in the absence of calcium or in MCU ko?

10. As requested, we have also measured the NADH/NAD⁺ ratio in MCU KO cells using the mito-SoNar reporter, and show that this ratio is reduced by 20% in MCU KO cells as expected (Fig. S14). We have amended the text as follows (line 156): “Conversely, NADH:NAD⁺ ratio was reduced in MCUKO cells (Fig. S14).” Please also see our response to inquiry R42 by reviewer 3 below.

R9. There is an apparent complexity in this story that needs to be resolved: the experiments with the mito-LbNOX reduces the NADH:NAD ratio and this prevents the increase in iMCU, suggesting that the increase in iMCU is driven partly by NADH driven ROS generation. However, the (proposed) increase in mitochondrial matrix calcium should be driving the increase in NADH? Doesn't this constitute the basis for a positive feedback cycle here? NADH accumulates with complex I deficiency and drives ROS generation, This leads to upregulation of MCU which should increase mitochondrial calcium which should drive increased NADH and that should further increase ROS generation?

We do not think that the model constitutes positive feedback since, as Complex I impairment worsens, it becomes inactive and/or degraded, preventing further ROS production and breaking any cycle^{7,8}. Regarding mito-*LbNOX*, we believe it is actually increasing MCU at baseline by blunting Complex I-dependent MCU degradation, which is why it doesn't increase further with rotenone. Consistent with this, though it is hard to make a direct comparison in stable lines expressing different sets of proteins, it appears that the uniporter current in the baseline (no rotenone) condition is higher when mt-*LbNOX* is expressed (~140 pA/pF, Fig. 2D) compared to when it is absent (~70 pA/pF, Fig. 1B).

R10. The legend could usefully give more information?

We have added the information about mito-SoNar ratio as indicated in the response to R9 above.

R11. I really struggled with figure 3: in a, the WB ought to be quantified.

11. We have quantified the western of the co-immunoprecipitation in Fig. S20. Additionally, as noted in response R14, the association between Complex I and MCU has been experimentally verified in unbiased screens by at least two other groups.

R12. In Fig 3b what are A5/A7 etc?

12. These refer to the different mVenus-tagged NDUF subunits tested in each panel. We have relabeled the Fig. 3B and expanded the legend as described in R13 below.

R13. How do these relate to the different proteins? It is very hard for a reader to interpret this figure! I don't understand what it is we're looking at.

In Fig. 3B, we display the FRET efficiency between MCU-mCerulean and mVenus-tagged Complex I subunits for a range of different subunits. Above each plot is a

micrograph showing localization for mVenus compared to MitoTracker, revealing colocalization for every subunit except for NDUFA2 and NDUFA5 (which then serve as negative controls). Within a graph, each point in the density plot corresponds to flow cytometric data for a single cell, and displays the FRET efficiency between MCU-mCerulean and the mVenus-tagged subunit (y-axis) against the mVenus fluorescence (x-axis) for that individual cell. This allows a plot of FRET efficiency across a very wide range of construct expression and reveals the underlying distribution, making it easy to separate FRET categories. Constructs that failed to target mitochondria when fused to mVenus (NDUFA2 and NDUFA5) showed no FRET, seen as a flat line near the baseline. Spurious concentration-dependent FRET, due to the small confines of the mitochondria, is seen as a linear increase in FRET efficiency with mVenus concentration (mito-mVenus, NDUFA7, NDUF6, NDUF6, and perhaps NDUFV2). True FRET is visible as a rapid rise in efficiency to a plateau (NDUFA10, NDUF3), suggesting these are closely interacting with MCU. The summary figure displays a subset of the data from each graph at moderate mVenus expression. We have made the following changes:

13. We have expanded the figure legend (line 493): “Images above each graph show mVenus-tagged constructs and MitoTracker Orange (MTO). Within graphs, each point in the density plot is an individual cell expressing MCU-mCerulean and the corresponding mVenus-tagged construct. For each cell, the FRET efficiency between its mCerulean and mVenus is displayed in the y-axis, while the degree of expression of the mVenus-tagged construct is revealed by the mVenus fluorescence in the x-axis. Right, summary of FRET efficiency between MCU-mCerulean and the corresponding mVenus-tagged constructs at moderate mVenus expression.”
14. We have added the FRET criteria to the methods section (line 864): Constructs showing no FRET have a flat distribution. Spurious concentration-dependent FRET is seen as a linear increase in FRET efficiency with mVenus concentration. True FRET is visible as a rapid rise in efficiency to a plateau.

R14. Similarly, the duolink data are startlingly unimpressive and hard to read. All I can see is a few green spots.

We are uncertain as to the concern with seeing “a few green spots”, or why this should be unimpressive. The Duolink PLA system creates a focal spot of fluorescence when antibodies for two different proteins are within ~40 nm, so a pattern of green spots is the expected output. Since the process is stochastic and labels only a fraction of every such pair of nearby antibodies, it is used for comparison between conditions rather than as an absolute measure. Several points bear emphasis. First, our data clearly shows that, whereas endogenous MCU and the Complex I subunit NDUF2 interact in wild-type or untreated cells, disruption of Complex I due to rotenone or NDUF10 mutation drastically reduces the number of cells containing Duolink spots, consistent with our hypothesis. Second, the overall appearance of our figure is very similar to the expected output of the Duolink system, as seen by its application by other investigators

using a variety of antibodies and targets, which also produce a pattern of spots (Fig. 3, PMC5334148; Fig. 3, PMC6137027; Fig. 1, PMC6613789; Fig. 3, PMC29257951; Fig. 5, PMC6123654; Fig. 4, PMC7111589; Fig. 6, PMC6102827; Fig. 3, PMC5675595). Third, since we don't expect the interaction between MCU and Complex I to be constitutive, and since this interaction appears to lead to MCU turnover (Fig. 4C-F), we would not expect a huge number of spots per cell, consistent with our results here. Fourth, because we agree with the reviewer that no one assay will be perfect, we have demonstrated the Complex I-MCU interaction using not just Duolink PLA, but also orthogonal approaches including mass spectrometry, co-immunoprecipitation, and FRET, suggesting that it is robustly detectable. Finally, there is **independent confirmation by other groups of MCU-Complex I interaction in unbiased proteomic analyses**, as we mentioned briefly in the manuscript (line 204-208)^{9,10}. In a proteomic screen for novel Complex I assembly factors, performed by the Harper/Gygi labs at Harvard, MCU (listed by its former name, CCDC109A, Fig. 1 in PMC4023825) was identified as interacting with NDUFA9. In another proteomic screen using BioID proximity labeling, performed by the Shoubridge lab at McGill, interactions were found between MCU and a substantial number of Complex I matrix-arm components (Table S4 in PMID: 32877691). Taken together, we think there is quite robust data for a MCU-Complex I interaction.

R15. I struggle to understand why the authors make so much of a close interaction between complex I and MCU given that the expressed mitominiSOG – which is not localised had the same effect, and the SOD overexpression, which is also not localized, quenched the effect. These imply that localization is not really critical? This begs the question of whether other causes of mitochondrial oxidative stress also increase MCU levels?

The reviewer raises a very interesting point. There are indeed other mechanisms by which ROS may modulate MCU, explored by the Madesh group¹¹, which appear to involve channel clustering via glutathionylation of an N-terminal cysteine residue. However, this mechanism does not appear responsible for the increases we see here, as a cysteine-free version of MCU still enhances following Complex I inhibition (Fig. S18). Moreover, it is clear that the interaction between MCU and Complex I is critical for controlling MCU levels. This is best shown in Fig. 4C-F), where replacing the N-terminal domain and abolishing the interaction with Complex I leads to stabilized channels, whereas enforcing a MCU-Complex I interaction with the FKBP-FRB-rapamycin system leads to almost complete intramitochondrial MCU degradation (Fig. 4F). These experiments were done in the absence of rotenone or any other form of Complex I impairment, emphasizing the importance of the MCU-Complex I interaction in controlling MCU turnover at baseline. Thus, our explanation for the SOD and miniSOG experiments is that, in the long term, these are affecting MCU much less than Complex I, because Complex I is much larger and concentrated in cristae. Since the ROS treatment is performed briefly, direct changes to MCU caused by ROS have likely dissipated by the time of our electrophysiological assay several days later. Thus, these manipulations are altering Complex I-MCU interactions leading indirectly to changes in MCU levels. We have made the following changes:

15. Retitled Fig. 7 to emphasize that it is our current hypothetical model best integrating the mechanistic evidence presented (line 541): **Current hypothetical model for Complex I-induced protein turnover of MCU.**

R16. On Page 7 you say ‘Finally, Complex I impairment decouples MCU.’ – I’m not sure what this means? What are you referring to? Decouples from what?

16. We have rephrased (line 285): **“Finally, impaired Complex I can no longer bind MCU ...”**

xiii) ‘Because TFAM deletion alters the driving force for Ca²⁺ uptake, we quantified the steady state ability of mitochondria to take up Ca²⁺ rather than uptake rates (Fig. 5d).’ why not energise with succinate and this way you might avoid problems with potential in your models?

To clarify, our main concern was to show persistent MCU activity despite embryonic deletion of the *Tfam* gene, so our approach seemed sufficient for such a binary yes-or-no assessment. Nevertheless, we have repeated the experiment energizing with succinate and see similar results as before. MCU activity is persistent in the double KO despite gene deletion, suggesting that the uniporter protein is being degraded at a much slower rate.

17. We have replaced the prior experiments with succinate-energized assays (Fig. 5C-D) and amended the methods (line 935) and legend (line 525): **C. Ca²⁺ uptake in isolated cardiac mitochondria incubated in Oregon Green BAPTA 6F (OGB6F). Arrow indicates 10 μM Ca²⁺ pulse. D. Summary of the normalized rate of Ca²⁺ uptake after a 10 μM Ca²⁺ pulse.**

R17. Fig 5 – quantify westerns. 5B doesn’t show MCU levels in the TFAM ko? Have I misunderstood something?

18. We have quantified the westerns in Fig. S35.

19. We have added the TFAM KO data to Fig. 5B.

R18. The calcium uptake data in Fig 5C seem very odd. I initially asked how does mito calcium uptake happen in the TFAM-MCU double ko, but the blots show substantial MCU expression in the ‘TFAM-MCU double knockout’ which is confusing?

The reviewer is entirely correct. Despite the *Mcu* gene being deleted in the *Tfam-Mcu* double KO, the protein appears to be stabilized and its activity is persistent, but declining, over the course of the first postnatal week. This is consistent with the hypothesis that impairment in Complex I stabilizes MCU.

R19. Why are there no measurements of mitochondrial calcium uptake to match the current measurements – ie to show that with rotenone treatment mitochondrial calcium uptake is increased? Surely this lies at the heart of the whole story? You imply that calcium uptake might be decreased because of a decreased potential which clearly gets complicated but you can get around that by energizing mitochondrial with succinate to bypass the complex I block.

The reviewer is correct that disabling Complex I will lead to marked decreases in membrane potential, and potentially alter other factors such as mitochondrial pH, shape, and Ca²⁺ buffering capacity. For these reasons the gold standard for measuring uniporter activity is the whole-mitoplast approach, because it allows us to control these other factors and measure in isolation the amount of channel activity. Nevertheless, the reviewer raises the important point of the functional consequence for global mitochondrial Ca²⁺ uptake of such enhanced uniporter activity. We have already shown the increase in mitochondrial Ca²⁺ uptake in *Tfam* KO animals (our prior publication¹ as well as in Fig. 5C). We now pursued the succinate-based assay requested by the reviewer for cell lines as well. Despite Complex I dysfunction, we found that Ca²⁺ uptake was increased in Complex-I deficient mitochondria from these cells (Fig. S10), consistent with the data obtained via electrophysiology as well as in *Tfam* KO animals.

20. We have added the figure legend (line 1071) and updated the results (line 121): “The increase in I_{MiCa} corresponded to increases in mitochondrial Ca²⁺ uptake measured by imaging (Fig. S10)”.

21. We have updated the methods (lines 933)

R20. ‘In summary, in a disease model of cardiac ETC dysfunction, activity of the uniporter was essential for survival’. This seems an odd statement given that the mice are all dead at 10 weeks?!

22. We have rephrased (line 388): “In summary, in a disease model of cardiac ETC dysfunction, loss of the uniporter further impaired survival.”

Reviewer #2 (Remarks to the Author):

Cardiac depletion of *Tfam* has been used as a model for cardiomyopathy induced by mitochondrial OXPHOS deficiencies and was previously shown to be accompanied by accumulation of Ca²⁺ in mitochondria. Here, the authors pick up on this observation and reveal an intriguing functional link between complex I activity and mitochondrial Ca²⁺ influx via MCU, which appears to be conserved from *Drosophila* to mouse and human cells. They demonstrate that MCU binds complex I but dissociates from it upon complex I inhibition. Whereas MCU bound to complex I is unstable and rapidly degraded, dissociation from non-functional complex I results in stabilization of MCU and increased mitochondrial Ca²⁺ uptake.

This is an interesting study which is likely of relevance for complex I diseases. It reveals an unexpected mechanism regulating the turnover of MCU via the activity of complex I. Whereas assembly into protein complexes is generally observed to stabilize proteins, here the authors suggest the opposite and find that increased MCU turnover correlates with binding to complex I. While the functional link between complex I inhibition and MCU stability is well documented, some questions remain whether or not this is indeed regulated via assembly. Although the authors use various approaches to assess MCU-complex I interaction and use an elegant approach to show that tethering of MCU to a complex I subunit induces degradation, I am still wondering about quantitative aspects of complex formation.

R21. Which fraction of MCU is bound to complex I and is this the fraction which is degraded?

Yes, MCU interaction with Complex I leads to its degradation. This is directly tested in Fig. 4C-F: when forced to interact with Complex I, MCU gets rapidly degraded. Since the fraction of MCU interacting with Complex I will depend on the abundance of both, the activity of Complex I, the abundance and activity of the quality control proteases, and perhaps other factors, this number will vary between tissues, developmental stages, and conditions. In addition, since Complex I is localized within cristae, whereas MCU has a more variable distribution, spatial constraints will also regulate the MCU-Complex I interaction^{12,13}. Based on super-resolution imaging data of MCU in cardiomyocytes, we estimate that up to 1/3 of total MCU may interact with Complex I at any time point, given this abundance of MCU within cristae¹³.

23. We have added the following to the discussion (line 443): **The fraction of MCU binding Complex I likely varies between tissues and cell types based on their abundances, localization, activity of quality-control proteases, and possibly other factors. In cardiomyocytes, approximately 1/3 of MCU protein is located within cristae, suggesting this fraction may be interacting with Complex I.**

R22. Many experiments are done after over-expression further hampering a quantitative assessment. It is shown that disturbances in the NADH/NAD⁺ ratio or increased ROS can induce MCU stabilization but how MCU binding to complex I is regulated remains speculative.

This is correct. There are potentially a range of potential mechanisms to be discovered that regulate MCU-Complex I interactions, but identifying such further mechanisms seem well beyond the scope of the current study, and would likely require extensive further experiments. Our goal here is to establish that this interaction has important functional consequences for MCU stability and organismal survival. Though some experiments rely on overexpression, note that the Duolink experiments examine endogenous interactions (Fig 3C-D, see also response R25), and changes in ion channel activity and survival are of endogenous MCU in Complex I inhibited states in cells, mice, and *Drosophila* (Fig. 1, 5, 6). Our future goals include understanding what regulates such interactions, including the factors listed above (abundance of MCU, Complex I, proteases; spatial localization within mitochondrial subdomains; possibly other regulatory factors such as cristae remodeling). We have specified that this is likely variable and potentially regulated in the discussion (see response R21 above).

R23 Another concern is the conclusion of the authors that increasing MCU stability affects survival of heart specific Tfam-knockout mice, as the authors do not compare the TFAM-MCU DKO with heart-specific MCU knockout (Fig. 5).

We apologize for not including this data. As shown by the Elrod and Molkentin labs, cardiac specific MCU KO does not cause any substantial lethality^{14,15}. Similarly, we have not seen any substantial mortality associated with cardiac-specific *Mcu* KO. Thus, the loss of MCU by itself is not lethal, but even partial MCU depletion in the *Tfam-Mcu* DKO leads to faster mortality.

24. We have updated Fig. 5F to include the survival data for MCU KO.

R24. In this context, I find the title somehow misleading as MCU does not compensate for the loss of complex I activity but rather is likely part of metabolic rewiring allowing cells to better cope with complex I deficiencies.

We agree completely that enhanced MCU activity helps cells better cope with complex I deficiencies.

25. To avoid confusion, we have altered the title to make this more explicit:
Mitochondrial calcium uniporter stabilization preserves energetic homeostasis during Complex I impairment.

Specific points:

R25. What is the efficiency of the immunoprecipitation of MCU and complex I in Fig. 3a and for endogenous proteins? Exposures of different immunoblots cannot be compared. Previous studies on MCU interactors did not detect complex I.

We respectfully disagree with the statement that prior studies have not shown MCU-Complex I interactions. There is **independent confirmation of the MCU-Complex I interaction in unbiased proteomic analyses**, as we mention briefly in the manuscript (line 204-208)^{9,10}. In a proteomic screen for novel Complex I assembly factors, performed by the Harper/Gygi labs at Harvard, MCU (identified by its former name, CCDC109A) was identified as interacting with NDUFA9. In another proteomic screen using proximity labeling, performed by the Shoubridge lab at McGill, interactions were found between MCU and a substantial number of Complex I matrix-arm components. Taken together, these publications and our data provide multiple independent sources of evidence for a MCU-Complex I interaction. We did not pursue the CoIP approach for endogenous MCU because the best antibody against MCU (Cell Signaling) works poorly for immunoprecipitation. Instead, we used the Duolink approach to label instances of endogenous interaction in a stochastic manner (Fig. 5C, See also response R14), and show that such interactions are lost when Complex I is impaired. This has the distinct advantage over CoIP of labeling endogenous interactions *in situ* in cells rather in lysed solutions.

26. We have repeated the CoIP on a single membrane, and replaced the prior blot with the new results (Fig. 3A)

R26. Does MCU bind to fully assembled complex I? Only three complex I subunits were found to precipitate with MCU raising the possibility that a complex I intermediate rather than assembled complex I binds to MCU. Moreover, the FRET analysis identifies different complex I subunits than detected in the immunoprecipitate. Therefore, conclusions on the binding interface appear not appropriate.

As mentioned previously, in data from the Shoubridge lab, MCU was found in close proximity to a variety of Complex I subunits across different modules, suggesting that it interacts with the holocomplex. Moreover, in the Complex I-deficient cells, subcomplexes are still present (Fig. 1 in ref 4), yet in these cases we no longer see interactions between MCU and Complex I (Fig. 5D), further supporting an interaction with the holocomplex. Regarding the interactions with different subunits seen in

different assays, this reflects limitations of the assays. For example, for the FRET assays we cannot label every protein (such as NDUFA13 or NDUFA8) due to inaccessible C-termini. Similarly, antibodies against some of these subunits either do not exist, or work poorly for immunoblotting or immunocytochemistry/Duolink. In addition, space constraints vary between the assays. FRET detects colocalization *in situ* on a shorter spatial scale than Duolink, which is shorter still than the Shoubridge BioID approach, so we expect to detect fewer interactions with FRET, more with Duolink, and more still with BioID. For Duolink, we chose the most robust and well-validated antibodies for labelling endogenous Complex I and MCU in cells (Abcam NDUFS2, Cell Signaling MCU), while the FRET approach allowed us to tag several Complex I subunits with accessible C termini. Thus, though we do not claim to have isolated a Complex I subunit that directly binds MCU, we have used a variety of different assays, and our results show that all the interacting Complex I proteins cluster on a surface near the interface between the Q module and transmembrane domains (Fig. S26). This was entirely unexpected and we believe an extremely strong result, as several different methodologies consistently show interaction along the same surface of Complex I.

R27. In Fig. 4a, the authors use DOX to inhibit transcription of (overexpressed) MCU to assess MCU turnover. Cell proliferation should be monitored, which could contribute to the decline in MCU steady state levels. Moreover, it remains unclear why the authors did not use CHX to examine the stability of endogenous MCU as they did for (endogenous) ROMO1 in Figure S4A. What is the stability of other subunits of the MCU complex?

Excess cell proliferation would definitely reduce the apparent lifetime of MCU, as the reviewer points out. To counteract this, we reduced FBS in the media to 3%, which effectively minimized proliferation and allowed robust comparisons. We have made the following changes:

27. We have now included our quantification of proliferation with this protocol (Fig. S28), which showed slightly higher (though not statistically significant) rates in the control over the rotenone-treated cells. We have also quantified the western blots showing the changes in MCU lifetime, correcting for proliferation, in Fig. S30, which show that the prolongation in MCU lifetime during Complex I inhibition is not an artifact of proliferation. We amended the text (line 268) as follows: “Cells were grown in 3% fetal bovine serum-supplemented media, to minimize proliferation (Fig. S28).”

Our initial pilot experiments several years ago did, in fact, use a 24-hour treatment of CHX for assays of MCU and other uniporter subunits. As requested by the reviewer, we now include these in Fig. S27. We saw a very subtle change in MCU, and no change for EMRE and MICU1, between rotenone and control. Because the effect on MCU was very subtle and difficult to assess in short-term CHX treatments, we realized we would have to monitor degradation over a longer timespan. However, because CHX application is toxic to cells over longer periods, causing cell-wide disruption of protein synthesis, we were unable to use it for the 2-4 day treatments necessary to show robust

changes in MCU turnover. For these, we had to exploit the doxycycline-repressible system. We did not pursue turnover assays for EMRE and MICU1 as we saw no changes in the CHX experiments. Moreover, protease regulation of EMRE lifetime has been studied extensively before^{16,17}. It turns over very rapidly when not in complex with MCU and is dependent on membrane-bound proteases. For MICU1, we found that though its levels increased with cardiac-specific TFAM knockout, inhibition of Complex I in HEK cells did not affect its levels (Fig. 1C, Fig. S5). It is also in the intermembrane space, and thus unlikely to interact substantially with Complex I.

28. We now include Fig. S27, and have amended the text as follows (line 264): “To evaluate MCU degradation, initial experiments with short-term cycloheximide treatment suggested a subtle change in MCU, but not MICU1 or EMRE lifetimes (Fig. S27).”

R28. In general, the description of experiments in the figure legends should be improved.

We have made multiple changes as detailed in the remainder of this document.

Reviewer #3 (Remarks to the Author):

This paper by Chaudhuri and co-workers focuses on the activity regulation of the mitochondrial calcium uniporter (MCU) channel. Evidence is provided that complex I (CI) impairment stimulates MCU activity. It is proposed that this occurs via CI-mediated MCU degradation mediated by physical contacts between CI and the MCU pore-forming subunit (occurring under normal conditions), and that this MCU degradation mechanism is inhibited by CI dysfunction. Overall, the manuscript is nicely presented but not always easy to follow (i.e. it is not sufficiently explained how or why a certain experiment is performed). This likely has also to do with the large amount of data presented. In this context, I have several points that require clarification.

MAJOR POINTS

R29. General remark: Regarding the use of chronic rotenone treatment in cell models. Although its primary effect might be limited to CI inhibition, cell metabolism and redox homeostasis also change (over time) as a consequence of this treatment. This makes the “rotenone-model” difficult to interpret in a CI-specific manner. Although less studied, this also holds true for the chronic use of other ETC inhibitors. To limit the potential impact of these effects, it is advisable to use the lowest inhibitor concentration that fully inhibits mitochondrial oxygen consumption in the cell model of choice.

The reviewer is correct that chronic inhibitor application may have off-target effects. Because of this, and as mentioned by the reviewer, we had performed a dose-response analysis for rotenone (Fig. 1B), and used the lowest concentration producing full effect (1 μ M), though a weaker effect could also be seen at lower doses. Because the effect is indirect, it is possible that lower doses could be used, but would require much longer incubations, during which further problems with cell proliferation and off-target effects could arise. In preliminary experiments, we found that, although we could see effects as early as 24h after incubation, the combination of 1 μ M rotenone for 2-3 days produced the increase in I_{MiCa} most robustly. Precisely because we were concerned about off-target effects, we have confirmed the effect of Complex I inhibition using

multiple different approaches: gene knockout (NDUFB10, FOXRED1, NDUFS4), mutation (NDUFB10 iPSCs), inhibition (*Drosophila*), as well as two different animal models of Complex I deficiency (*Drosophila*, mice). In addition, there are no off-target effects of rotenone on the uniporter when applied acutely to mitoplasts (Fig. S3), and rotenone treatment does not alter other mitochondrial ionic currents (Fig. S13), only the uniporter. Finally, we confirm that Complex I controls MCU turnover directly, via the experiments in Fig. 4C-F and via persistent MCU activity in *Tfam-Mcu* double KO mice in Fig 5A-D, and none of these assays use rotenone. We did not perform dose-response analyses for the other Complex inhibitors, as the chosen doses effectively inhibit the respective complexes in our (Fig. S1) and others' data (some examples include ¹⁸⁻²⁰, though there are many more), yet they produced no effect on I_{MiCa} , further confirming that the effect is specific to Complex I, and not due to general ETC dysfunction or a mitochondrial stress response (since ETC inhibition via these other inhibitors would be expected to produce downstream consequences that overlap with rotenone effects). Thus, we have performed extensive validation across assays and model systems to show that our findings are not due to non-specific rotenone effects.

R30. Results section: How was the conclusion reached that “disruption” of the NDUFS4 subunit induces a milder CI deficiency? It is referred to Fig. S1F, but this shows currents?

This is based on prior published data from the Ryan lab at Monash showing much less disruption of Complex I in NDUFS4 KO compared to NDUFB10 KO in these HEK293T cells (Fig. 1 in ref 4)

29. We have rephrased the sentence, moving the reference to Fig. S1F away from the references to the Stroud paper (line 109): “We saw a similar increase in I_{MiCa} after disruption of accessory subunit NDUFS4 (Fig. S6, S8), which produces much milder Complex I deficiency²⁴,...”

R31. Results section: For making statements regarding the degree of CI deficiency it is necessary to obtain information on CI enzymatic activities. This also holds true for the other inhibitors and ETC complexes that are studied. For instance, are the studies ETC activities fully or partially inhibited in the used inhibitor-treated and other models?

The reviewer raises an important point. We had performed such assays but had not included them in our initial submission. We now include these enzymatic assays revealing loss of ETC activity now in Fig. S1. These results show that incubation with the respective compound is effective at inhibiting the activity of Complex I, III, and IV.

30. We have added the relevant methods (line 924), and updated the results with the figure (line 91): “Individual ETC Complexes were inhibited pharmacologically for 2-3 days in HEK293T cells (Fig. S1).”

R34. Results section: It is puzzling that matrix free calcium levels were higher in CI deficient cells (Fig. S1I). Why would there be mitochondrial calcium uptake? Does this suggest that calcium release from the ER is activated in CI deficient cells? Why? How does the magnitude of the sensor ratio signal (Fig. S1I) relate to the “real” matrix calcium concentration. Is the K_m value for calcium of this sensor known?

Mitochondrial Ca^{2+} levels are higher in the $\text{NDUFB10}^{\text{KO}}$ compared to WT, and slightly higher in the $\text{FOXRED1}^{\text{KO}}$ compared to WT, but note there is substantial overlap with WT levels in both cases. Thus, in a hypothetical situation where MCU activity was unchanged during Complex I dysfunction, loss of $\Delta\Psi$ would lead to reduced Ca^{2+} accumulation and possibly reduced activity of the TCA cycle. In contrast, with the enhancement in MCU we actually show here, the increased uniporter levels will enhance Ca^{2+} accumulation (independent of any changes in ER stores or other sources of Ca^{2+}), and the main effect may be to preserve or restore mitochondrial Ca^{2+} levels to normal/slightly above normal. Hence the substantial overlap with WT levels. We did not pursue further analyses of ER stores, as (1) these results are not directly relevant to the mechanism of MCU enhancement discovered here; (2) it would be difficult to attribute any changes (up or down) in ER levels to changes in mito Ca^{2+} stores since long-term ETC inhibition is likely to have substantial other effects, and assessing these by depleting MCU in Complex I deficient cells is difficult due to substantial loss of viability; and finally, (3) the behavior of Ca^{2+} ER stores is likely different between the cell types considered here (HEK and iPSC cell lines, Drosophila flight muscle, cardiac mitochondria), whereas the MCU enhancement we describe is quite robustly conserved amongst these cell types.

Regarding the mitochondrial Ca^{2+} levels, the K_D for GCamp6m is 167 nM²¹. We calibrated the ratio using ionomycin in the presence of high EGTA or high Ca^{2+} , using a ratiometric Ca^{2+} binding equation²², and obtained average values of 206 nM, 370 nM, and 221 nM for WT, $\text{NDUFB10}^{\text{KO}}$, $\text{FOXRED1}^{\text{KO}}$. Although our levels are consistent with recent publications^{23,24}, note that quantification of resting mitochondrial Ca^{2+} levels in normal cells has revealed values ranging from 10 nM-200 μM ²⁵. Thus, to be conservative and not over-interpret, we have kept the ratios in the figure, as this is the actual data and our main goal is to compare levels between the different conditions.

31. We have added the estimated mitochondrial Ca^{2+} levels to the figure legend (line 1078).

R35. Results section: Please provide experimental evidence demonstrating that $\Delta\Psi$ is depolarized under CI-deficient conditions in the used (cell) models. This cannot be assumed since $\Delta\Psi$ depends on many other factors including CV and ANT activity, as well as metabolic state. This is evidenced by previous studies in the literature.

The reviewer raises an important point. We have now quantified TMRM staining as a surrogate for $\Delta\Psi$ and show that inhibition of Complex I reduces $\Delta\Psi$ by ~50% in $\Delta\Psi$ after rotenone treatment, and by ~15-20% in $\text{FOXRED1}^{\text{KO}}$ and $\text{NDUFB10}^{\text{KO}}$ cells (Fig. S7).

32. We have added methods (lines 724), the figure with TMRM data (Fig. S7), and amended the results as follows (line 108): “As with rotenone-treated cells, genetic inhibition of Complex I reduced $\Delta\Psi$ levels (Fig. S7)”.

R36. Results section: In case of Fig 1C (and for proper interpretation of many of the other results), it is crucial to demonstrate whether the levels of the CI holo-complex have changed. This can be done with BN-

PAGE experiments (using VDAC1 or CII as a control). Previous studies suggest that chronic rotenone treatment increases the levels of this complex, which would suggest that increased CI levels correlate with increased MCU levels. This could mean that CI-MCU interaction plays a stabilizing role. This possibility needs to be ruled out.

The reviewer raises an excellent point regarding the effects of rotenone on Complex I stability. As requested, we have performed BN-PAGE of rotenone-treated cells and find a decrease in holo-Complex I (Fig. S4). This is consistent with prior publications using cardiomyocytes and SH-SY5Y cells (Fig. 3D from PMC5836091 in SH-SY5Y cells; Fig. 1A,C from PMC6286307 in rat cardiac mitochondria), where chronic rotenone treatment produced decreases in holo-Complex I and respirasome formation^{26,27}. However, rotenone has been shown to stabilize holo-Complex I in fibroblasts, though this was studied mostly in the context of enhanced Complex I degradation caused by defects in other parts of the ETC^{18,28,29}. Of note, even in normal fibroblasts investigators have found that rotenone or piericidin produced a decrease in holo-Complex I (Fig. 3C from PMID 27052170 in fibroblasts). Thus, it appears the effect of rotenone on holo-Complex I may be partly context (e.g. length of incubation) or cell-line dependent, and under conditions used in our experiments it reduces these levels. Likely because the reductions in these levels occur slowly, we need >48h of rotenone treatment to robustly reveal the uniporter phenotype. For the gene-deleted cell lines, reductions in holo-Complex I formation were documented previously^{2,4}. Most importantly, in all these contexts, there is a profound reduction in physiological Complex I activity (see response to R27 below), which is key for MCU turnover, and loss of such activity would thus lead to MCU stabilization.

33. We have added Fig. S4, updated the methods (line 738), and added to the results (line 99): “This suggests that rotenone did not enhance I_{MICA} directly, but through its effect on Complex I, inhibiting electron transfer to ubiquinone, and leading to substantial reductions in Complex I activity and levels (Fig. S1, S4).”

R37. Results section: I find it difficult to understand how dysfunctional CI would induce its own ROS-mediated degradation. In such a mechanism, there would never be any CI activity detectable in cells carrying CI mutations (i.e. there would be no CI holo-complex; see my remark above). I guess it is overlooked here that cells adapt to increased ROS levels by activation of ROS detox systems, as well as metabolic adaptation mechanisms, to prevent (full) CI loss and energy dysbalance.

We agree with the reviewer that ROS detoxification may help these cells adapt, and that the degree of Complex I self-inactivation and subsequent degradation exists on a continuum. For example, some of the relatively severe Complex I deletions we studied (e.g. $NDUFB10^{KO}$) do produce almost complete loss of the holocomplex⁴. With rotenone treatment, or in $NDUFS4^{KO}$ cells, there is much less holocomplex loss. Notably, however, the amount of holocomplex left does not necessarily predict the amount of Complex I activity, as with both chronic rotenone incubation and in $NDUFS4^{KO}$ mice, there is loss of Complex I activity out of proportion to the more modest changes in holocomplex levels^{18,30}. Thus, what seems to be true in all of these

conditions is that, with prolonged loss of physiological Complex I activity, levels of MCU build up as uniporter degradation is reduced.

R38. Results section: MitoSOX analysis does not measure ROS production but ROS levels. Moreover, MitoSOX can be oxidized during its travel to mitochondria (it reacts at very high rates with ROS), its fluorescence analysis is not specific to superoxide, and it accumulates in mitochondria in a $\Delta\Psi$ dependent manner. Therefore MitoSOX analysis is not ROS- or mitochondria-specific.

Measurement of ROS is complicated, as the reviewer points out, due to various sources and species. In our sample, direct imaging of cells showed that the vast majority of mitoSOX fluorescent signal comes from mitochondria (Fig. 2C). Since Complex I inhibited cells had lower $\Delta\Psi$ (Fig. S7), if accumulation rates were the main determinant of signal strength, we would predict much less intense mitoSOX staining in Complex I deficient cells due to poorer accumulation. Nevertheless, mitoSOX staining was much higher in the Complex I impaired cells (Fig. 2C is on a logarithmic scale), and thus, if anything, it is possible we are actually underestimating the degree of excess ROS (superoxide and others) in the Complex I inhibited cells. In addition, it is well established that the main reactive oxygen species produced by Complex I is superoxide, and that measurement of other ROS species (e.g. H₂O₂) often reflects dismutation of superoxide, allowing these indirect measures to also reflect Complex I ROS levels. In our case, the data we produce here using mitoSOX is entirely consistent with a large amount of data using a variety of methods in both cells or isolated mitochondria from multiple other investigators, showing increases in mitochondrial ROS production during Complex I impairment, whether due to rotenone or mutations in Complex I subunits (though there are many more references, for a set of examples, see^{7,18,27,31-37}). Moreover, our interpretation is also supported by the independent assays using overexpression of SOD2 and miniSOG.

34. Because, as the reviewer points out, mitoSOX is not entirely specific for superoxide, and can, to a variable degree, also detect other ROS species, we have amended the statement in our results to replace “superoxide” with “ROS” (line 153): “**We found evidence for increases in ROS levels using the mitochondrially-targeted sensor MitoSOX**”

R39. Results section: I have difficulty understanding the results with LbNOX. How does the fact that rotenone increases mitochondrial NADH levels links to the data in figure 1?

The purpose of this experiment was to determine if the maintenance of NADH production by mitochondrial Ca²⁺ was necessary during Complex I dysfunction. Mito-*LbNOX* is a water-forming NADH oxidase used by the Mootha group to show that restoring the NADH/NAD⁺ ratio during Complex I dysfunction helped rescue cell viability and proliferation. We found that, whereas mito-*LbNOX* could partially rescue Complex I deficiency caused by piericidin in wild-type cells, it was lethal in MCU^{KO} cells treated with piericidin. These results were further supported by both the mouse and *Drosophila* data, where loss of MCU proved more lethal during Complex I impairment. In particular, it is notable that the mechanism for increasing the

NADH/NAD⁺ ratio is different for Complex I impairment versus MCU enhancement. Complex I impairment increases NADH by inhibiting its oxidation to NAD⁺, whereas MCU enhancement increases NADH by increasing its synthesis from NAD⁺. It may be that NADH synthesis may still need to be supported during Complex I dysfunction (via MCU), to support reactions beyond Complex I, such as for redox balance via NADPH, one-carbon metabolism, or other pathways.

35. We have edited the sentence in the results to emphasize NADH production (line 179): “...suggesting that the uniporter **maintenance of NADH production** is required for survival during Complex I dysfunction.”

R40. Results section: Singlet oxygen is not superoxide. How does this work in the mt-miniSOG cell model? Does MitoSOX detect singlet oxygen?

We thank the reviewer for this comment, as it led us to discover a minor misstatement (which actually improves the relevance of our approach). First, MitoSOX does indeed detect singlet oxygen, as shown by the documentation for this product from ThermoFisher, and has been used for this purpose³⁸. Second, and more importantly, though initially developed as a singlet oxygen generator, miniSOG also produces a substantial amount of superoxide as well^{39,40}. In addition, singlet oxygen itself reacts with several protein side chains, including Trp, Cys, His, Tyr, and Met, with a lifetime similar to superoxide⁴¹. Thus, we expect miniSOG to be a ROS sensitizer.

36. We have edited our miniSOG description (line 169): “This fluorescent flavoprotein generates both singlet oxygen **and superoxide** when excited by blue light”

R41. Results section: I have trouble understanding why first rotenone is used and then piericidin A? Long-term cell treatment with rotenone can induce off-target effects which are often not observed for piericidin A. How was the concentration of piericidin A determined? The latter question can also be asked for all other inhibitors (see also my first comment).

To maintain consistency for the *LbNOX* assay, we used same protocol as in the Mootha paper describing *LbNOX*¹⁹, which used piericidin. We decided to perform this experiment after all the experiments with rotenone had already been completed. We think the use of different methods to impair Complex I significantly strengthens the paper, as it shows that uniporter enhancement occurs robustly regardless of the treatment producing Complex I impairment or the assay used to measure the uniporter. Moreover, we have shown that MCU enhancement is not an off-target effect by replicating the results in multiple gene-deleted lines, *Drosophila*, and mice. See also the response to R29.

R42. Results section: As stated above, in various (cell) models the CI-deficient state is associated with increased NADH and reduced NAD⁺ levels. So why would “uniporter boosting of NADH/NAD⁺ ratio” be required for survival during CI dysfunction?

We appreciate this insightful comment. This is one of the most intriguing consequences of our results, and is part of what we hope to accomplish in future work on this project, once this manuscript describing Complex I-dependent MCU enhancement is published. As noted in response R39, it is likely that uniporter-dependent NADH production is needed for bioenergetic compensation, as, without MCU, mito-*LbNOX* is ineffective at rescuing Complex I dysfunction. Beyond establishing this, we have not had the chance to explore how mitochondrial Ca^{2+} alters metabolism during Complex I dysfunction. Since mitochondrial Ca^{2+} has well-documented effects on TCA cycle and fuel substrate utilization⁴²⁻⁴⁵, there are a range of possibilities. As mentioned in response R39, these include NADH or TCA cycle contribution to redox balance, one-carbon metabolism, epigenetic changes, or potentially other pathways⁴⁶⁻⁴⁹. It is also possible that, though Complex I dysfunction leads to MCU enhancement in a highly conserved manner across several different tissues and species, as shown here, the downstream consequences of such uniporter enhancement may vary depending on cell type or disease state, as relevant pathways may be activated to a variable degree. Sorting this out will require substantial further effort and does not directly affect the major conclusions, novelty, or impact of this report (detailing the mechanistic link between MCU and Complex I, and showing that MCU is needed during Complex I dysfunction in animal models). Thus, we feel that elucidating the mechanisms by which MCU-Complex I interactions maintain downstream mitochondrial metabolic homeostasis is well beyond the scope of the current work, especially as the reviewers have already mentioned concerns regarding the length and complexity of this manuscript. Here, we limit ourselves to showing that NADH/NAD⁺ biology is likely a key component of MCU enhancement, via the mito-*LbNOX* experiment. As listed in item 35 above, we have amended the statement regarding “boosting” to emphasize NADH production.

R43. Results section: Does the length of the linker between the fluorescent protein (FPs) and CI subunit affect the FRET signal? I.e. is it expected to increase the length scale across which FRET will occur? To demonstrate interaction also acceptor-bleaching analysis needs to be performed.

We used a relatively short linker length of 13 amino acids. At this length, spurious FRET is very low and only occurs at very high expression levels (see panels for mt-mV, NDUFA7, NDUF6, and NDUF6 in **Fig. 3B**), confirming that our linker length is adequately sized. Changing the linker length would potentially change the FRET efficiency, but given that longer linkers would allow the fluorophore to occupy a greater volume and more orientations, causing increased separation of donor and acceptor, it is more likely that FRET efficiency would decrease for proteins that are co-localized. This is seen when the fluorophores are joined by long linkers⁵⁰.

Please note that we do not need acceptor bleaching analysis to demonstrate FRET interaction (though we have performed this analysis as requested for a subset of our constructs, see below). Intensity-based methods to quantify FRET rely on two general approaches^{51,52}: (1) quenching of donor fluorescence (which can be assessed by the acceptor photobleaching method, as mentioned by the reviewer), and (2) enhanced

acceptor fluorescence due to sensitized emission (which underlies the flow cytometric method we use in **Fig. 3B**). These are entirely independent methods, both quantifying FRET. The flow cytometric method has been validated for FRET and used by multiple groups in the past^{50,51,53-58}. Conversely, although acceptor photobleaching is popular for cytoplasmic FRET studies, we have found that it is a much poorer alternative for mitochondrial FRET analyses. There are several reasons for poor performance of acceptor photobleaching. First, because the same cells need to be imaged at high magnification before and after photobleaching, FRET needs to be measured via microscope imaging and flow cytometry cannot be used, severely limiting the number of cells that can be used for analysis. Most acceptor photobleaching assays are limited to tens of cells, whereas with the flow cytometry approach we can measure >10,000 within a few minutes, allowing us to display an entire FRET distribution. Second, mitochondria are motile organelles and extremely sensitive to light-induced oxidative damage. The photobleaching step is exceedingly toxic to mitochondria and rapidly leads to cessation of mitochondrial motion and frequently causes mitochondrial fission, potentially leading to spurious results. For this reason, acceptor photobleaching for mitochondrial FRET assays can only be performed in fixed cells (e.g. with paraformaldehyde), not live ones. The flow cytometric approach, in contrast, can be used easily with live cells, since each cell is only illuminated once briefly at non-toxic laser powers. Third, the principle of acceptor photobleaching assumes that this step selectively destroys the acceptor, leaving the FRET donor intact. In practice however, donors typically have non-trivial absorbance at the laser wavelengths used for acceptor photobleaching, especially because of the high intensity used. This requires donor intensity measurements to be corrected for loss of donor fluorescence during photobleaching. We assessed for such correction factors by expressing only mito-targeted mCerulean (FRET donor) and performing photobleaching at the acceptor wavelength. We found the degree of donor photobleaching to be highly nonlinear at low mCerulean expression, meaning that such corrections could only be performed accurately in cells expressing high levels of mCerulean-tagged MCU. This reduces the number of cells that can be used for analysis to only those that have high expression levels, and can potentially make it more difficult to separate spurious from true FRET within the narrow confines of the mitochondria. In the flow cytometric approach, correction factors for channel crosstalk are linear across a vast range of fluorescence intensities, allowing us to obtain a broad distribution of FRET efficiency and easily discriminate true versus spurious FRET. In summary, the acceptor photobleaching method can only be used in relatively few, fixed cells with high fluorescent protein expression, whereas the flow cytometric approach can be used for thousands of live cells across a wide range of fluorescent protein expression. Nevertheless, subject to these limitations, we performed acceptor photobleaching analysis for the control (mito-mVenus) and two constructs (NDUFA10-mVenus and NDUFS3-mVenus) showing positive FRET (**Fig. S22**), and found results confirming the flow cytometric analysis.

37. We have added the new Fig. S22, updated methods (line 872), and modified the results as follows (line 230): “NDUFA10 and NDUF3, however, demonstrated robust interaction with MCU-mCerulean, confirmed via acceptor photobleaching FRET assays as well (Fig. S22), implying close physical proximity.”

R44. Results section: Importantly, not all CI subunits can be tagged with an FP without disturbing their incorporation in the CI holocomplex. This can be investigated using BN-PAGE and/or fluo- and Western-blot analyses. Are the tagged subunits also close to other ETC complexes in the supercomplex(es)? This can be deduced from the available structures. If so, is it expected that they still can interact with MCU?

We discussed the limitations of tagging Complex I subunits with fluorescent proteins in line 216, particularly the possibility of tagging only those with C-termini exposed on the surface of the holocomplex. For the FRET experiment, we tagged subunits that had previously been shown to successfully incorporate into Complex I with fluorescent protein tags (Fig. 1D in PMC3516733)^{59,60}, along with 3 additional subunits (NDUFA5, NDUFA7, and NDUFA10) that had exposed C-termini on Complex I structures. Two of these additional fusions also targeted mitochondria (insets in Fig. 3B). As requested by the reviewer, we performed BN-PAGE on the constructs that were mitochondria-targeted and show clear incorporation into holo-Complex I and higher-order supercomplexes (Fig. S21). We had some difficulty with this experiment because the three commercially-available anti-GFP antibodies we tested all produced weak staining in BN-PAGE. Such reduced performance of many antibodies when used for BN-PAGE is often noted, presumably due to differences in epitope exposure between SDS- and BN-PAGE⁶¹. (Note that a custom-made GFP antibody was used in ref ⁵⁹.) Nevertheless, there is clear incorporation of mVenus-tagged NDUF subunits into holo-Complex I.

The subunits we tagged reside in several Complex I sub-modules, and interestingly, data across all the various interaction assays (CoIP, FRET, Duolink, as well as those published by independent groups) revealed that the uniporter interaction surface clusters at the junction between the Q module and transmembrane domains, far away from Complex III, and IV in supercomplex structures (see Fig. S26)

38. We have added Fig. S21, updated methods (lines 739), and added the following to the results (lines 219): “To sample various portions of Complex I, we tagged eight NDUF subunits with mVenus, while MCU was fused to mCerulean (Fig. 3b, Fig. S21).”

R45. Results section: Please demonstrate that doxycycline treatment does not inhibit mitochondrial parameters (e.g. oxygen consumption). Such inhibition has been observed in various systems and might affect the signalling involved in CI-MCU interaction.

As requested, we have measured the effects of doxycycline on oxygen consumption and find no effect (Fig. S29). In prior studies, doxycycline effects on mitochondrial function become evident when incubated for a prolonged period of time (>7 days), at concentrations >5 µg/mL, and/or in special media^{62,63}, whereas we used lower concentrations in normal media for shorter durations.

39. We have added Fig. S29, updated methods (lines 909), and added a reference to the figure on line 269: “Addition of 1 $\mu\text{g}/\text{mL}$ doxycycline had no effect on respiration (Fig. S29),”

R46. Results section: Why is it more likely that ROS generated by mito-miniSOG will non-specifically damage CI? What is the range of action of ROS (singlet oxygen?) generated by mini-SOG?

We state this because Complex I is much larger than MCU (both in transmembrane and matrix-facing components) and enriched in cristae, and thus a much bigger target. Moreover, if miniSOG were preferentially non-specifically damaging MCU, we would expect to see a reduction in I_{MiCa} current rather than an increase. There is likely some degree of damage to all mitochondrial proteins, with Complex I $>$ MCU because of the reasons listed above, and a subsequent increase in MCU levels over 2-3 days due to impairment in Complex I. We found the effect only with very short periods of miniSOG light activation, while longer miniSOG light activation clearly led to toxicity and cell death. Thus, our explanation for the miniSOG experiments is that, in the long term, these are affecting MCU much less than Complex I, because Complex I is much larger and concentrated in cristae. Since the ROS treatment is performed briefly, direct changes to MCU caused by ROS have likely dissipated by the time of our electrophysiological assay several days later. Thus, these manipulations are altering Complex I-MCU interactions leading indirectly to changes in MCU levels. This also fits with the data in Fig. 4C-F, where forcing MCU and Complex I to interact at baseline (without changes in ROS) leads to rapid degradation of MCU. Note that miniSOG produces superoxide as well as singlet oxygen (see item R40 above), which are both highly reactive, with diffusion lifetimes $< 5 \mu\text{s}$ and distances $< 0.5 \mu\text{m}$ ^{64,65}.

MINOR POINTS

R47. Results section: What is exactly meant with “typical Ca^{2+} fluorescence assays”?

These refer to the Ca^{2+} uptake assays routinely used as indirect surrogates for uniporter activity (e.g. Fig. S10 and Fig. 5C in our paper, and in most other publications examining uniporter function).

R48. Results section: Can the authors rule out that the composition of their pipette medium affects their results in whole-mitoplast voltage-clamp recordings?

We are unclear as to what is meant by this question, and apologize for any confusion. The whole-mitoplast electrophysiology approach mimics whole-cell voltage clamp in accessing the interior of the mitochondria (the matrix space) and replacing the contents with the pipette/internal solution, then recording I_{MiCa} in a defined bath/external solution. This is done purposely, to allow full control over the transmembrane voltage gradient, internal and external pH, Ca^{2+} concentrations/buffering, ionic composition, osmolarity, etc. Under these conditions, we can measure uniporter activity directly by quantifying I_{MiCa} , eliminating these other factors ($\Delta\Psi$, pH, Ca^{2+} buffering, osmolarity, etc.) that typically confound indirect measurements of uniporter activity, such as in fluorescence assays. In this regard, our

solutions definitely “affect the results”, but this is done purposely to allow precise measurement of total isolated uniporter activity. In fact, this is the gold standard for quantifying ion channel activity, and such electrophysiology has been routinely used in a vast number of publications examining plasma-membrane ion channel behavior in neurons, cardiomyocytes, and other cell types. Quoting Bertil Hille’s definitive textbook on ion channels⁶⁶: “The procedure, known as voltage clamping, has been the best biophysical technique for the study of ion channels for over 50 years.” As far as we can tell, the main reason whole-mitoplast electrophysiology has been deployed less frequently for investigating the uniporter or other mitochondrial ion channels has been the difficulty of the protocol, though in the past decade there has been increased recognition by multiple groups for the need for this technique to quantify uniporter activity (see response R4 above for citations for recordings by other groups performed in mitochondria from different cell types using solutions similar to ours).

R49. Results section: how does pyruvate-induced NAD⁺ regeneration exactly work?

Pyruvate-to-lactate conversion by cytoplasmic lactate dehydrogenase regenerates NAD⁺ from NADH, which can then alter mitochondrial NADH/NAD⁺ directly (mitochondrial NAD⁺ transporter) or indirectly (e.g. malate-aspartate shuttle). This was originally suggested by King and Attardi due to the pyruvate requirement of mtDNA-depleted cell lines^{67,68}, and subsequently tested by the Mootha group in their publication on *LbNOX*¹⁹.

R50. Results section: What is meant by MCU acting as a “ROS buffer”? Is this an assumption or a conclusion?

40. This was meant as a simile rather than a mechanism. To avoid confusion, we have removed the phrase.

- 1 Sommakia, S. *et al.* Mitochondrial cardiomyopathies feature increased uptake and diminished efflux of mitochondrial calcium. *Journal of molecular and cellular cardiology* **113**, 22-32, doi:10.1016/j.yjmcc.2017.09.009 (2017).
- 2 Formosa, L. E. *et al.* Characterization of mitochondrial FOXRED1 in the assembly of respiratory chain complex I. *Human molecular genetics* **24**, 2952-2965, doi:10.1093/hmg/ddv058 (2015).
- 3 Stroud, D. A. *et al.* COA6 is a mitochondrial complex IV assembly factor critical for biogenesis of mtDNA-encoded COX2. *Human molecular genetics* **24**, 5404-5415, doi:10.1093/hmg/ddv265 (2015).
- 4 Stroud, D. A. *et al.* Accessory subunits are integral for assembly and function of human mitochondrial complex I. *Nature* **538**, 123-126, doi:10.1038/nature19754 (2016).
- 5 Friederich, M. W. *et al.* Mutations in the accessory subunit NDUFB10 result in isolated complex I deficiency and illustrate the critical role of intermembrane space import for complex I holoenzyme assembly. *Human molecular genetics* **26**, 702-716, doi:10.1093/hmg/ddw431 (2017).
- 6 Fieni, F., Bae Lee, S., Jan, Y. N. & Kirichok, Y. Activity of the mitochondrial calcium uniporter varies greatly between tissues. *Nature communications* **3**, 1317, doi:10.1038/ncomms2325 (2012).
- 7 Chen, Y. R., Chen, C. L., Zhang, L., Green-Church, K. B. & Zweier, J. L. Superoxide generation from mitochondrial NADH dehydrogenase induces self-inactivation with specific protein radical

- formation. *The Journal of biological chemistry* **280**, 37339-37348, doi:10.1074/jbc.M503936200 (2005).
- 8 Pryde, K. R., Taanman, J. W. & Schapira, A. H. A LON-ClpP Proteolytic Axis Degrades Complex I to Extinguish ROS Production in Depolarized Mitochondria. *Cell reports* **17**, 2522-2531, doi:10.1016/j.celrep.2016.11.027 (2016).
- 9 Guarani, V. *et al.* TIMMDC1/C3orf1 functions as a membrane-embedded mitochondrial complex I assembly factor through association with the MCIA complex. *Mol Cell Biol* **34**, 847-861, doi:10.1128/MCB.01551-13 (2014).
- 10 Antonicka, H. *et al.* A High-Density Human Mitochondrial Proximity Interaction Network. *Cell metabolism* **32**, 479-497 e479, doi:10.1016/j.cmet.2020.07.017 (2020).
- 11 Dong, Z. *et al.* Mitochondrial Ca²⁺ Uniporter Is a Mitochondrial Luminal Redox Sensor that Augments MCU Channel Activity. *Molecular cell* **65**, 1014-1028 e1017, doi:10.1016/j.molcel.2017.01.032 (2017).
- 12 Gottschalk, B. *et al.* MICU1 controls cristae junction and spatially anchors mitochondrial Ca(2+) uniporter complex. *Nature communications* **10**, 3732, doi:10.1038/s41467-019-11692-x (2019).
- 13 De La Fuente, S. *et al.* Strategic Positioning and Biased Activity of the Mitochondrial Calcium Uniporter in Cardiac Muscle. *The Journal of biological chemistry* **291**, 23343-23362, doi:10.1074/jbc.M116.755496 (2016).
- 14 Kwong, J. Q. *et al.* The Mitochondrial Calcium Uniporter Selectively Matches Metabolic Output to Acute Contractile Stress in the Heart. *Cell reports* **12**, 15-22, doi:10.1016/j.celrep.2015.06.002 (2015).
- 15 Luongo, T. S. *et al.* The Mitochondrial Calcium Uniporter Matches Energetic Supply with Cardiac Workload during Stress and Modulates Permeability Transition. *Cell reports* **12**, 23-34, doi:10.1016/j.celrep.2015.06.017 (2015).
- 16 Tsai, C. W. *et al.* Proteolytic control of the mitochondrial calcium uniporter complex. *Proceedings of the National Academy of Sciences of the United States of America* **114**, 4388-4393, doi:10.1073/pnas.1702938114 (2017).
- 17 Konig, T. *et al.* The m-AAA Protease Associated with Neurodegeneration Limits MCU Activity in Mitochondria. *Molecular cell* **64**, 148-162, doi:10.1016/j.molcel.2016.08.020 (2016).
- 18 Verkaart, S. *et al.* Superoxide production is inversely related to complex I activity in inherited complex I deficiency. *Biochimica et biophysica acta* **1772**, 373-381, doi:10.1016/j.bbadis.2006.12.009 (2007).
- 19 Titov, D. V. *et al.* Complementation of mitochondrial electron transport chain by manipulation of the NAD⁺/NADH ratio. *Science* **352**, 231-235, doi:10.1126/science.aad4017 (2016).
- 20 Ji, D., Kamalden, T. A., del Olmo-Aguado, S. & Osborne, N. N. Light- and sodium azide-induced death of RGC-5 cells in culture occurs via different mechanisms. *Apoptosis* **16**, 425-437, doi:10.1007/s10495-011-0574-4 (2011).
- 21 Chen, T. W. *et al.* Ultrasensitive fluorescent proteins for imaging neuronal activity. *Nature* **499**, 295-300, doi:10.1038/nature12354 (2013).
- 22 Grynkiewicz, G., Poenie, M. & Tsien, R. Y. A new generation of Ca²⁺ indicators with greatly improved fluorescence properties. *The Journal of biological chemistry* **260**, 3440-3450 (1985).
- 23 Lu, X. *et al.* Measuring local gradients of intramitochondrial [Ca(2+)] in cardiac myocytes during sarcoplasmic reticulum Ca(2+) release. *Circulation research* **112**, 424-431, doi:10.1161/circresaha.111.300501 (2013).
- 24 Wescott, A. P., Kao, J. P. Y., Lederer, W. J. & Boyman, L. Voltage-energized Calcium-sensitive ATP Production by Mitochondria. *Nat Metab* **1**, 975-984, doi:10.1038/s42255-019-0126-8 (2019).
- 25 Fernandez-Sanz, C., De la Fuente, S. & Sheu, S.-S. Mitochondrial Ca²⁺ concentrations in live cells: quantification methods and discrepancies. *FEBS letters* **593**, 1528-1541, doi:<https://doi.org/10.1002/1873-3468.13427> (2019).

- 26 Inoue, N. *et al.* Knockdown of the mitochondria-localized protein p13 protects against
experimental parkinsonism. *EMBO reports* **19**, e44860, doi:10.15252/embr.201744860 (2018).
- 27 Jang, S. & Javadov, S. Elucidating the contribution of ETC complexes I and II to the respirasome
formation in cardiac mitochondria. *Scientific reports* **8**, doi:10.1038/s41598-018-36040-9 (2018).
- 28 Guarás, A. *et al.* The CoQH2/CoQ Ratio Serves as a Sensor of Respiratory Chain Efficiency. *Cell*
reports **15**, 197-209, doi:10.1016/j.celrep.2016.03.009 (2016).
- 29 Tropeano, C. V. *et al.* Fine-tuning of the respiratory complexes stability and supercomplexes
assembly in cells defective of complex III. *Biochim Biophys Acta Bioenerg* **1861**, 148133,
doi:10.1016/j.bbabi.2019.148133 (2020).
- 30 Kruse, S. E. *et al.* Mice with mitochondrial complex I deficiency develop a fatal
encephalomyopathy. *Cell metabolism* **7**, 312-320, doi:10.1016/j.cmet.2008.02.004 (2008).
- 31 Kayser, E.-B., Sedensky, M. M. & Morgan, P. G. Region-Specific Defects of Respiratory
Capacities in the Ndufs4(KO) Mouse Brain. *PLoS one* **11**, e0148219,
doi:10.1371/journal.pone.0148219 (2016).
- 32 Robb, E. L. *et al.* Control of mitochondrial superoxide production by reverse electron transport at
complex I. *The Journal of biological chemistry* **293**, 9869-9879, doi:10.1074/jbc.RA118.003647
(2018).
- 33 Wang, L. *et al.* Mitochondrial Respiratory Chain Inhibitors Involved in ROS Production Induced
by Acute High Concentrations of Iodide and the Effects of SOD as a Protective Factor. *Oxid Med*
Cell Longev **2015**, 217670, doi:10.1155/2015/217670 (2015).
- 34 Schlehe, J. S., Journel, M. S., Taylor, K. P., Amodeo, K. D. & LaVoie, M. J. The mitochondrial
disease associated protein Ndufaf2 is dispensable for Complex-I assembly but critical for the
regulation of oxidative stress. *Neurobiol Dis* **58**, 57-67, doi:10.1016/j.nbd.2013.05.007 (2013).
- 35 Dlaskova, A., Hlavata, L., Jezek, J. & Jezek, P. Mitochondrial Complex I superoxide production
is attenuated by uncoupling. *The international journal of biochemistry & cell biology* **40**, 2098-
2109, doi:10.1016/j.biocel.2008.02.007 (2008).
- 36 Pryde, K. R. & Hirst, J. Superoxide is produced by the reduced flavin in mitochondrial complex I:
a single, unified mechanism that applies during both forward and reverse electron transfer. *The*
Journal of biological chemistry **286**, 18056-18065, doi:10.1074/jbc.M110.186841 (2011).
- 37 Wong, H. S., Dighe, P. A., Mezera, V., Monternier, P. A. & Brand, M. D. Production of
superoxide and hydrogen peroxide from specific mitochondrial sites under different bioenergetic
conditions. *The Journal of biological chemistry* **292**, 16804-16809, doi:10.1074/jbc.R117.789271
(2017).
- 38 Zhou, X. *et al.* Laser controlled singlet oxygen generation in mitochondria to promote
mitochondrial DNA replication in vitro. *Scientific reports* **5**, 16925, doi:10.1038/srep16925
(2015).
- 39 Pimenta, F. M., Jensen, R. L., Breitenbach, T., Etzerodt, M. & Ogilby, P. R. Oxygen-dependent
photochemistry and photophysics of "miniSOG," a protein-encased flavin. *Photochem Photobiol*
89, 1116-1126, doi:10.1111/php.12111 (2013).
- 40 Barnett, M. E., Baran, T. M., Foster, T. H. & Wojtovich, A. P. Quantification of light-induced
miniSOG superoxide production using the selective marker, 2-hydroxyethidium. *Free Radic Biol*
Med **116**, 134-140, doi:10.1016/j.freeradbiomed.2018.01.014 (2018).
- 41 Davies, M. J. Reactive species formed on proteins exposed to singlet oxygen. *Photochem*
Photobiol Sci **3**, 17-25, doi:10.1039/b307576c (2004).
- 42 Kwong, J. Q. *et al.* The mitochondrial calcium uniporter underlies metabolic fuel preference in
skeletal muscle. *JCI Insight* **3**, doi:10.1172/jci.insight.121689 (2018).
- 43 Glancy, B. & Balaban, R. S. Role of mitochondrial Ca²⁺ in the regulation of cellular energetics.
Biochemistry **51**, 2959-2973, doi:10.1021/bi2018909 (2012).
- 44 Altamimi, T. R. *et al.* Cardiac-specific deficiency of the mitochondrial calcium uniporter
augments fatty acid oxidation and functional reserve. *Journal of molecular and cellular*
cardiology **127**, 223-231, doi:10.1016/j.yjmcc.2018.12.019 (2019).

- 45 Gherardi, G. *et al.* Loss of mitochondrial calcium uniporter rewires skeletal muscle metabolism and substrate preference. *Cell death and differentiation* **26**, 362-381, doi:10.1038/s41418-018-0191-7 (2019).
- 46 Maynard, A. G. & Kanarek, N. NADH Ties One-Carbon Metabolism to Cellular Respiration. *Cell metabolism* **31**, 660-662, doi:10.1016/j.cmet.2020.03.012 (2020).
- 47 Roberti, A., Fernández, A. F. & Fraga, M. F. Nicotinamide N-methyltransferase: At the crossroads between cellular metabolism and epigenetic regulation. *Mol Metab* **45**, 101165, doi:10.1016/j.molmet.2021.101165 (2021).
- 48 Yang, L. *et al.* Serine Catabolism Feeds NADH when Respiration Is Impaired. *Cell metabolism* **31**, 809-821.e806, doi:10.1016/j.cmet.2020.02.017 (2020).
- 49 O'Rourke, B., Ashok, D. & Liu, T. Mitochondrial Ca(2+) in heart failure: Not enough or too much? *Journal of molecular and cellular cardiology* **151**, 126-134, doi:10.1016/j.yjmcc.2020.11.014 (2020).
- 50 Lee, S. R., Sang, L. & Yue, D. T. Uncovering Aberrant Mutant PKA Function with Flow Cytometric FRET. *Cell reports* **14**, 3019-3029, doi:10.1016/j.celrep.2016.02.077 (2016).
- 51 Rivas, S., Hanif, K., Chakouri, N. & Ben-Johny, M. Probing ion channel macromolecular interactions using fluorescence resonance energy transfer. *Methods in enzymology* **653**, 319-347, doi:10.1016/bs.mie.2021.01.047 (2021).
- 52 Erickson, M. G., Alseikhan, B. A., Peterson, B. Z. & Yue, D. T. Preassociation of calmodulin with voltage-gated Ca(2+) channels revealed by FRET in single living cells. *Neuron* **31**, 973-985, doi:10.1016/s0896-6273(01)00438-x (2001).
- 53 Kanner, S. A., Shuja, Z., Choudhury, P., Jain, A. & Colecraft, H. M. Targeted deubiquitination rescues distinct trafficking-deficient ion channelopathies. *Nature Methods* **17**, 1245-1253, doi:10.1038/s41592-020-00992-6 (2020).
- 54 Kang, P. W. *et al.* Elementary mechanisms of calmodulin regulation of Na(V)1.5 producing divergent arrhythmogenic phenotypes. *Proceedings of the National Academy of Sciences of the United States of America* **118**, doi:10.1073/pnas.2025085118 (2021).
- 55 Papa, A. *et al.* Adrenergic Ca(V)1.2 Activation via Rad Phosphorylation Converges at $\alpha(1C)$ I-II Loop. *Circulation research* **128**, 76-88, doi:10.1161/circresaha.120.317839 (2021).
- 56 Choi, H. Y. *et al.* p53 destabilizing protein skews asymmetric division and enhances NOTCH activation to direct self-renewal of TICs. *Nature communications* **11**, 3084, doi:10.1038/s41467-020-16616-8 (2020).
- 57 Niu, J., Yang, W., Yue, D. T., Inoue, T. & Ben-Johny, M. Duplex signaling by CaM and Stac3 enhances Ca(V)1.1 function and provides insights into congenital myopathy. *The Journal of general physiology* **150**, 1145-1161, doi:10.1085/jgp.201812005 (2018).
- 58 Liu, G. *et al.* Mechanism of adrenergic Ca(V)1.2 stimulation revealed by proximity proteomics. *Nature* **577**, 695-700, doi:10.1038/s41586-020-1947-z (2020).
- 59 Dieteren, C. E. *et al.* Subunit-specific incorporation efficiency and kinetics in mitochondrial complex I homeostasis. *The Journal of biological chemistry* **287**, 41851-41860, doi:10.1074/jbc.M112.391151 (2012).
- 60 Dieteren, C. E. *et al.* Subunits of mitochondrial complex I exist as part of matrix- and membrane-associated subcomplexes in living cells. *The Journal of biological chemistry* **283**, 34753-34761, doi:10.1074/jbc.M807323200 (2008).
- 61 Calvaruso, M. A., Smeitink, J. & Nijtmans, L. Electrophoresis techniques to investigate defects in oxidative phosphorylation. *Methods* **46**, 281-287, doi:10.1016/j.ymeth.2008.09.023 (2008).
- 62 Perry, E. A. *et al.* Tetracyclines promote survival and fitness in mitochondrial disease models. *Nat Metab* **3**, 33-42, doi:10.1038/s42255-020-00334-y (2021).
- 63 Moullan, N. *et al.* Tetracyclines Disturb Mitochondrial Function across Eukaryotic Models: A Call for Caution in Biomedical Research. *Cell reports* **10**, 1681-1691, doi:10.1016/j.celrep.2015.02.034 (2015).

- 64 Mikkelsen, R. B. & Wardman, P. Biological chemistry of reactive oxygen and nitrogen and radiation-induced signal transduction mechanisms. *Oncogene* **22**, 5734-5754, doi:10.1038/sj.onc.1206663 (2003).
- 65 Baier, J., Maier, M., Engl, R., Landthaler, M. & Bäuml, W. Time-resolved investigations of singlet oxygen luminescence in water, in phosphatidylcholine, and in aqueous suspensions of phosphatidylcholine or HT29 cells. *J Phys Chem B* **109**, 3041-3046, doi:10.1021/jp0455531 (2005).
- 66 Hille, B. *Ion channels of excitable membranes*. 3rd edn, (Sinauer, 2001).
- 67 King, M. P. & Attardi, G. Human cells lacking mtDNA: repopulation with exogenous mitochondria by complementation. *Science* **246**, 500-503, doi:10.1126/science.2814477 (1989).
- 68 King, M. P. & Attardi, G. Isolation of human cell lines lacking mitochondrial DNA. *Methods in enzymology* **264**, 304-313, doi:10.1016/s0076-6879(96)64029-4 (1996).

REVIEWER COMMENTS

Reviewer #2 (Remarks to the Author):

The authors have carefully and satisfactorily addressed the concerns and questions on the original version of the manuscript and added a number of experiments supporting their conclusions. This is an intriguing and in part complex manuscript describing a conserved compensatory stabilization of MCU upon complex I deficiency which counteracts mitochondrial cardiomyopathy. Some questions remain, such as quantitative aspects, the precise interaction surface between MCU and complex I or why MCU bound to complex I is destabilized (increased ROS damage in proximity to complex I?), they do not affect the main finding of this interesting manuscript and can be clarified in future studies.

Reviewer #3 (Remarks to the Author):

Original comment 2: Results section: How was the conclusion reached that “disruption” of the NDUFS4 subunit (is knockout meant here?) induces a milder CI deficiency? It is referred to Fig. S1F, but this shows currents?

I'm still not happy with the formulations here. The authors use the presence of CI complexes as a measure of CI activity. This is not correct. If such statements need to be made, especially in the case of NDUFS4 knockout (is this what the authors mean with “disruption”), the use of BN-PAGE is not a good approach since the absence of NDUFS4 protein destabilizes CI, leading to the complex falling apart during the BN-PAGE procedure. However, in cellulo, there is still considerable rotenone-sensitive oxygen consumption in NDUFS4 knockout cells. If the authors want to use statements regarding milder or severe “CI deficiency”, I propose measuring rotenone-sensitive respiration in these cells.

Original comment 4: Results section: It is puzzling that matrix free calcium levels were higher in CI deficient cells (Fig. S1I). Why would there be mitochondrial calcium uptake? Does this not suggest that calcium release from the ER is activated in CI deficient cells? Why? How does the magnitude of the sensor ratio signal (Fig. S1I) relate to the “real” matrix calcium concentration. Is the K_m value for calcium of this sensor known?

This still does not answer my question why the calcium in the mitochondrial matrix is higher in CI deficient cells. Does this mean that mitochondria take up calcium (which is a low-affinity process) in resting cells? Why? Or was the data obtained under conditions of stimulated ER calcium release?

Original comment 5: Results section: Please provide experimental evidence demonstrating that deltaPSI is depolarized under CI-deficient conditions in the used models. This cannot be assumed since deltaPSI depends on many other factors including CV and ANT activity, as well as metabolic state. This is evidence by previous studies in the literature.

The new TMRM data appears convincing. However, can the authors rule out that the observed drop in mitochondrial TMRM fluorescence is to a less negative plasmamembrane potential? Moreover, please take care about how to formulate these results. A 50% drop in fluorescence is not a 50% depolarization in deltaPSI given the Nernstian behaviour of the TMRM dye. I propose using the word “depolarization” instead of “drop”.

Original comment 10: Results section: Singlet oxygen is not superoxide. How does this work in the mt-miniSOG cell model? Does MitoSOX detect singlet oxygen?

I am not totally convinced by this explanation (see the work of Robinson, PNAS, 2006 on MitoHE, which is actually MitoSOX).

Original comment 13: Results section: Does the length of the linker between the fluorescent protein (FPs) and CI subunit affect the FRET signal? I.e. does it increase the length scale across which FRET will occur? To demonstrate interaction also acceptor-bleaching analysis needs to be performed.

I don't understand how the expression levels of the FRET sensor would specifically boost the FRET signal and how this "confirms" that the linker is of adequate length. Unless there is forced FRET occurring between individual FRET sensor molecules (which is not desirable at all). However, I'm willing to accept the author's explanation and appreciate the acceptor photobleaching analysis that has been performed.

Original comment 16: Results section: Why is it more likely that ROS generated by mito-miniSOG will non-specifically damage CI? What is the range of action of ROS (singlet oxygen?) generated by mini-SOG?

I don't understand the author's statement that superoxide is "highly reactive". What is meant here: that superoxide is highly reactive with biomolecules or that it is very quickly removed by SODs?

Reviewer #3 (Remarks to the Author):

Original comment 2: Results section: How was the conclusion reached that “disruption” of the NDUFS4 subunit (is knockout meant here?) induces a milder CI deficiency? It is referred to Fig. S1F, but this shows currents?

I’m still not happy with the formulations here. The authors use the presence of CI complexes as a measure of CI activity. This is not correct. If such statements need to be made, especially in the case of NDUFS4 knockout (is this what the authors mean with “disruption”), the use of BN-PAGE is not a good approach since the absence of NDUFS4 protein destabilizes CI, leading to the complex falling apart during the BN-PAGE procedure. However, in cellulo, there is still considerable rotenone-sensitive oxygen consumption in NDUFS4 knockout cells. If the authors want to use statements regarding milder or severe “CI deficiency”, I propose measuring rotenone-sensitive respiration in these cells.

Given the strength of the reviewer’s concerns, we have opted to remove the discussion of the severity of Complex I dysfunction in the different KO lines, as it is only a single sentence in the manuscript (line 109). This data is tangential to main topic of the manuscript, and does not alter the main conclusions, impact, or novelty of our findings. The sentence now reads “We saw a similar increase in I_{MiCa} after disruption of accessory subunit NDUFS4 as well (Fig. S6, S8).”

Original comment 4: Results section: It is puzzling that matrix free calcium levels were higher in CI deficient cells (Fig. S1I). Why would there be mitochondrial calcium uptake? Does this not suggest that calcium release from the ER is activated in CI deficient cells? Why? How does the magnitude of the sensor ratio signal (Fig. S1I) relate to the “real” matrix calcium concentration. Is the K_m value for calcium of this sensor known?

This still does not answer my question why the calcium in the mitochondrial matrix is higher in CI deficient cells. Does this mean that mitochondria take up calcium (which is a low-affinity process) in resting cells? Why? Or was the data obtained under conditions of stimulated ER calcium release?

The increase in mitochondrial Ca^{2+} is due to increased uniporter levels. Increased amounts of uniporter allow greater mitochondrial Ca^{2+} entry (Fig. 5C, S10) even during transient Ca^{2+} signals that occur in proliferating cells¹ or in resting cardiomyocytes. Two points are of note. First, elegant recent studies conducted independently by the Boyman/Lederer, Hajnoczky, and Kirichok groups have shown **that mitochondrial Ca^{2+} uptake occurs even at sub-micromolar cytoplasmic Ca^{2+} concentrations**, albeit at slower rates than during larger Ca^{2+} signals²⁻⁵. Sub-micromolar Ca^{2+} uptake is particularly evident in the heart, where there does not appear to be any concentration threshold for Ca^{2+} uptake²⁻⁵. Thus, transient Ca^{2+} signals in “resting” cells, such as Ca^{2+} sparks in cardiomyocytes or those associated with cell division in proliferating or cultured cells¹, will be more easily transmitted to mitochondria when uniporter levels are enhanced, raising mitochondrial Ca^{2+} levels. As mentioned in our prior response, because these sources of Ca^{2+} signals are heterogeneous across different cell lines/tissues, whereas the effect of Complex I on MCU is uniformly conserved, we

do not interrogate ER Ca^{2+} sources in the current manuscript, as it does not impact the major findings. In future work, we are interested in pursuing how Complex I dysfunction may alter sarcoplasmic reticulum-mitochondrial Ca^{2+} transfer in the heart, but this is a major undertaking and we prefer to defer it to after this manuscript is published.

Second, it is also worth pointing out that the commonly-used description of the uniporter as being “low-affinity” is not ideal and somewhat inaccurate, especially given the recent results by the groups mentioned above. The “low affinity” description arose from experiments, performed several decades ago, measuring Ca^{2+} uptake in isolated mitochondria. In these, Ca^{2+} uptake would not *saturate* (V_{max}) except at high micromolar Ca^{2+} concentrations, leading to micromolar calculated $K_{1/2}$ values for uptake ($\sim 10 \mu\text{M}$, see, for example, the 1994 J Bioenerg Biomembr review by Gunter and Gunter⁶). These micromolar $K_{1/2}$ values led to the description of uniporter-mediated Ca^{2+} uptake as “low affinity”. However, such analyses rarely interrogated Ca^{2+} uptake at sub-micromolar concentrations and also accounted poorly for concurrent changes in $\Delta\Psi$ and matrix Ca^{2+} buffering during Ca^{2+} entry. When whole-mitoplast electrophysiology was performed to control these confounding factors and more accurately measure Ca^{2+} binding, it was found that **the uniporter actually has an extremely high affinity for Ca^{2+} (<10 nM)**⁷, confirmed in recent structural studies to be due to stacked rings of closely-spaced negatively-charged glutamate and aspartate residues that bind Ca^{2+} at the channel pore⁸⁻¹². What was labelled as “low-affinity” in the early studies actually reflects the substantial capacity of mitochondria to take up Ca^{2+} (i.e. the high $K_{1/2}$ values reflect capacity for uptake, not affinity). Thus, the more accurate description is that the uniporter has extremely high affinity for Ca^{2+} , and endows mitochondria with high-capacity Ca^{2+} uptake (high-affinity, high-capacity).

Original comment 5: Results section: Please provide experimental evidence demonstrating that $\Delta\Psi$ is depolarized under CI-deficient conditions in the used models. This cannot be assumed since $\Delta\Psi$ depends on many other factors including CV and ANT activity, as well as metabolic state. This is evidence by previous studies in the literature.

The new TMRM data appears convincing. However, can the authors rule out that the observed drop in mitochondrial TMRM fluorescence is to a less negative plasma membrane potential?

In our cells, plasma membrane and non-mitochondrial intracellular TMRM fluorescence was $\sim 1\%$ of values seen in mitochondria across conditions. This suggests that direct plasma membrane fluorescence contribution to the TMRM signal is trivial, and cannot account for changes in levels in Complex I deficient cells. To rule out indirect mechanisms, we have now repeated the experiment in cells bathed in high potassium (125 mM) to depolarize the plasma membrane and largely remove its contribution. Under these conditions as well, TMRM staining remained diminished in Complex I deficient cells. We have updated Figure S7.

Moreover, please take care about how to formulate these results. A 50% drop in fluorescence is not a 50% depolarization in $\Delta\psi$ given the Nernstian behaviour of the TMRM dye. I propose using the word “depolarization” instead of “drop”.

We have amended line 108 to say that “genetic inhibition of Complex I depolarized $\Delta\psi$ ”.

Original comment 10: Results section: Singlet oxygen is not superoxide. How does this work in the mt-miniSOG cell model? Does MitoSOX detect singlet oxygen?

I am not totally convinced by this explanation (see the work of Robinson, PNAS, 2006 on MitoHE, which is actually MitoSOX).

It is unclear to us what the reviewer is requesting here, as the referenced paper¹³ does not discuss singlet oxygen in relation to MitoSOX, nor methods to distinguish singlet oxygen from superoxide, which is the topic of the reviewer’s original comment. The PNAS paper suggests measurement of MitoSOX signal using ~400 nm excitation is more selective for superoxide over several other forms of ROS (Table 1 in that paper), but singlet oxygen was not tested. First, even at 400 nm excitation, low but non-trivial fluorescence was seen from other ROS sources. Second, MitoSOX appears to be more sensitive for singlet oxygen than most of the other forms of ROS listed in the paper, per the Thermo product literature, so it is unclear if the selectivity for superoxide at 400 nm will similarly hold. As mentioned previously, we fully agree that MitoSOX can detect a variety of different forms of ROS, as shown by the documentation for this product from ThermoFisher, and has been used for quantifying singlet oxygen¹⁴ as well as other forms of ROS. We have removed any mentions of MitoSOX being specific for superoxide, and instead refer only to ROS (lines 152, 483). As requested, we have also cited the Robinson reference (line 152).

Original comment 13: Results section: Does the length of the linker between the fluorescent protein (FPs) and CI subunit affect the FRET signal? I.e. does it increase the length scale across which FRET will occur? To demonstrate interaction also acceptor-bleaching analysis needs to be performed.

I don’t understand how the expression levels of the FRET sensor would specifically boost the FRET signal and how this “confirms” that the linker is of adequate length. Unless there is forced FRET occurring between individual FRET sensor molecules (which is not desirable at all). However, I’m willing to accept the author’s explanation and appreciate the acceptor photobleaching analysis that has been performed.

We appreciate the reviewer’s comments. We apologize for the confusion with our original answer. Longer linkers may allow interactions between targets across longer distances. However, this comes at the expense of a greater volume potentially occupied by the fluorophore, which is now able to sample more space and orientations relative to the protein it is tagged to, reducing the likelihood of interacting and producing FRET with distant targets. Since the volume potentially occupied by the fluorophore grows as the cube power of the linker length (e.g. imagine a sphere where the protein of interest is the center, the linker the radius, and the fluorophore the surface), whereas the distance between two interacting

proteins is linear, the volume growth predominates. Therefore, in most cases FRET signals to be measured by long linkers are quite weak.

Original comment 16: Results section: Why is it more likely that ROS generated by mito-miniSOG will non-specifically damage CI? What is the range of action of ROS (singlet oxygen?) generated by mini-SOG?

I don't understand the author's statement that superoxide is "highly reactive". What is meant here: that superoxide is highly reactive with biomolecules or that it is very quickly removed by SODs?

We apologize for the confusion. We meant that these have diffusion lifetimes < 5 μ s and distances < 0.5 μ m in lipid/protein or cell preparations^{15,16}, not specifically reactivity with SODs.

1. Zhao, H., T. Li, K. Wang, F. Zhao, J. Chen, G. Xu, J. Zhao, T. Li, L. Chen, L. Li, Q. Xia, T. Zhou, H.Y. Li, A.L. Li, T. Finkel, X.M. Zhang, and X. Pan, *AMPK-mediated activation of MCU stimulates mitochondrial Ca(2+) entry to promote mitotic progression*. *Nat Cell Biol*, 2019. **21**(4): p. 476-486, PMID 30858581.
2. Wescott, A.P., J.P.Y. Kao, W.J. Lederer, and L. Boyman, *Voltage-energized Calcium-sensitive ATP Production by Mitochondria*. *Nat Metab*, 2019. **1**(10): p. 975-984, PMC6964030, PMID 31950102.
3. Csordas, G., T. Golenar, E.L. Seifert, K.J. Kamer, Y. Sancak, F. Perocchi, C. Moffat, D. Weaver, S. de la Fuente Perez, R. Bogorad, V. Koteliensky, J. Adijanto, V.K. Mootha, and G. Hajnoczky, *MICU1 controls both the threshold and cooperative activation of the mitochondrial Ca(2+)(+) uniporter*. *Cell Metab*, 2013. **17**(6): p. 976-87, 3722067, PMID 23747253.
4. Garg, V., J. Suzuki, I. Paranjpe, T. Unsulangi, L. Boyman, L.S. Milescu, W.J. Lederer, and Y. Kirichok, *The mechanism of MICU-dependent gating of the mitochondrial Ca(2+)uniporter*. *Elife*, 2021. **10**, PMC8437439, PMID 34463251.
5. Paillard, M., G. Csordas, G. Szanda, T. Golenar, V. Debattisti, A. Bartok, N. Wang, C. Moffat, E.L. Seifert, A. Spat, and G. Hajnoczky, *Tissue-Specific Mitochondrial Decoding of Cytoplasmic Ca2+ Signals Is Controlled by the Stoichiometry of MICU1/2 and MCU*. *Cell Rep*, 2017. **18**(10): p. 2291-2300, PMID 28273446.
6. Gunter, K.K., and T.E. Gunter, *Transport of calcium by mitochondria*. *J Bioenerg Biomembr*, 1994. **26**(5): p. 471-85, PMID 7896763.
7. Kirichok, Y., G. Krapivinsky, and D.E. Clapham, *The mitochondrial calcium uniporter is a highly selective ion channel*. *Nature*, 2004. **427**(6972): p. 360-4, PMID 14737170.
8. Baradaran, R., C. Wang, A.F. Siliciano, and S.B. Long, *Cryo-EM structures of fungal and metazoan mitochondrial calcium uniporters*. *Nature*, 2018. **559**(7715): p. 580-584, PMC6336196, PMID 29995857.
9. Fan, C., M. Fan, B.J. Orlando, N.M. Fastman, J. Zhang, Y. Xu, M.G. Chambers, X. Xu, K. Perry, M. Liao, and L. Feng, *X-ray and cryo-EM structures of the mitochondrial calcium uniporter*. *Nature*, 2018. **559**(7715): p. 575-579, PMC6368340, PMID 29995856.
10. Nguyen, N.X., J.P. Armache, C. Lee, Y. Yang, W. Zeng, V.K. Mootha, Y. Cheng, X.C. Bai, and Y. Jiang, *Cryo-EM structure of a fungal mitochondrial calcium uniporter*. *Nature*, 2018. **559**(7715): p. 570-574, PMC6063787, PMID 29995855.

11. Wang, Y., N.X. Nguyen, J. She, W. Zeng, Y. Yang, X.C. Bai, and Y. Jiang, *Structural Mechanism of EMRE-Dependent Gating of the Human Mitochondrial Calcium Uniporter*. *Cell*, 2019. **177**(5): p. 1252-1261 e13, PMC6597010, PMID 31080062.
12. Yoo, J., M. Wu, Y. Yin, M.A. Herzik, Jr., G.C. Lander, and S.Y. Lee, *Cryo-EM structure of a mitochondrial calcium uniporter*. *Science*, 2018. **361**(6401): p. 506-511, PMC6155975, PMID 29954988.
13. Robinson, K.M., M.S. Janes, M. Pehar, J.S. Monette, M.F. Ross, T.M. Hagen, M.P. Murphy, and J.S. Beckman, *Selective fluorescent imaging of superoxide in vivo using ethidium-based probes*. *Proc Natl Acad Sci U S A*, 2006. **103**(41): p. 15038-43, PMC1586181, PMID 17015830.
14. Zhou, X., Y. Wang, J. Si, R. Zhou, L. Gan, C. Di, Y. Xie, and H. Zhang, *Laser controlled singlet oxygen generation in mitochondria to promote mitochondrial DNA replication in vitro*. *Sci Rep*, 2015. **5**: p. 16925, PMC4649627, PMID 26577055.
15. Mikkelsen, R.B., and P. Wardman, *Biological chemistry of reactive oxygen and nitrogen and radiation-induced signal transduction mechanisms*. *Oncogene*, 2003. **22**(37): p. 5734-54, PMID 12947383.
16. Baier, J., M. Maier, R. Engl, M. Landthaler, and W. Bäuml, *Time-resolved investigations of singlet oxygen luminescence in water, in phosphatidylcholine, and in aqueous suspensions of phosphatidylcholine or HT29 cells*. *J Phys Chem B*, 2005. **109**(7): p. 3041-6, PMID 16851318.

REVIEWER COMMENTS

Reviewer #3 (Remarks to the Author):

The authors have now adequately addressed my concerns and remarks.